# Enhanced inter-regional coupling of neural responses and repetition suppression provide separate contributions to long-term behavioral priming

Stephen J. Gotts [1✉], Shawn C. Milleville[1] & Alex Martin [1]

Stimulus identification commonly improves with repetition over long delays ("repetition priming"), whereas neural activity commonly decreases ("repetition suppression"). Multiple models have been proposed to explain this brain-behavior relationship, predicting alterations in functional and/or effective connectivity (*Synchrony* and *Predictive Coding* models), in the latency of neural responses (*Facilitation* model), and in the relative similarity of neural representations (*Sharpening* model). Here, we test these predictions with fMRI during overt and covert naming of repeated and novel objects. While we find partial support for predictions of the Facilitation and Sharpening models in the left fusiform gyrus and left frontal cortex, the data were most consistent with the Synchrony model, with increased coupling between right temporoparietal and anterior cingulate cortex for repeated objects that correlated with priming magnitude across participants. Increased coupling and repetition suppression varied independently, each explaining unique variance in priming and requiring modifications of all current models.

[1] Section on Cognitive Neuropsychology, Laboratory of Brain and Cognition, National Institute of Mental Health, National Institutes of Health, Bethesda, MD, USA. ✉email: gottss@mail.nih.gov

Repeated exposure to objects during the performance of a task leads to improved identification speed and accuracy, a phenomenon referred to as "repetition priming"[1,2]. Repetition priming is stimulus-specific, occurs in a wide range of tasks, and is extremely long-lasting, surviving delays of months and even years[3–5]. The relative sparing of repetition priming in patients with damage to medial temporal lobe structures, such as those with amnesia, also highlights the likely neocortical basis of this form of learning[6]. Indeed, neocortical brain regions commonly exhibit a companion phenomenon to repetition priming referred to as "repetition suppression", in which neural activity in task-engaged brain regions decreases with repetition[7–10]. Like repetition priming, repetition suppression is stimulus-specific, builds up across repetitions, occurs in a wide range of tasks, and can be long-lasting[11,12]. The joint occurrence of repetition priming and repetition suppression across a wide range of experimental contexts with different sensory and motor modalities has led to the notion that these phenomena reflect incremental neocortical learning mechanisms, serving to form and shape long-term perceptual, conceptual, and motor knowledge representations throughout the brain[13–15].

Multiple theoretical models have been proposed to account for the simultaneous observation of both neural repetition suppression and behavioral priming (reviewed in[16]; see Fig. 1). The *Synchrony* model[16–18] holds that as neural activity decreases, cells become more synchronized in their firing, leading to a larger impact on downstream targets and earlier, more reliable propagation of individual spikes, supporting earlier motor responses.

**Fig. 1 Neural models of repetition priming.** Four prominent models of repetition priming and repetition suppression are considered. The Synchrony model (upper left) holds that neural activity becomes more synchronized with repetition, permitting more coordinated propagation of activity at lower overall activity levels. The Predictive Coding model (upper right) holds that top-down causal influences are more strongly negative, leading to repetition suppression in the receiving region, along with a gain enhancement that leads to more rapid onset and offset of responses. The Facilitation model (lower left) claims that neural activity onset and offset are advanced in time, with earlier peak responses and a reduction in overall activity. The Sharpening model (lower right) holds that weakly tuned, poorly responsive cells are the ones driving repetition suppression, reducing downstream support for competing stimulus identities and speeding downstream stimulus-selective responses. Figure reproduced with permission from Gotts et al.[16].

The *Predictive Coding* model, as formulated by Friston and colleagues[8,19,20], views the cortex as a form of hierarchical generative Bayesian statistical model in which perceptual inference occurs as an interaction between bottom-up sensory input ("evidence") and top-down expectations ("prediction"). Top-down predictions improve with repetition, reducing prediction error by inhibiting or suppressing bottom-up sensory evidence, thereby producing repetition suppression[8,21]. Simultaneously, this can speed up evoked neural responses via an increase in synaptic gain due to enhanced encoding precision and confidence, producing behavioral priming. A related model, the *Facilitation* model[8,22,23], simply posits that evoked neural responses are resolved more quickly in time with repetition with earlier termination of activity, reflected as repetition suppression when measured with techniques such as BOLD fMRI. Finally, the *Sharpening* model[7,10] holds that while neural activity is decreasing overall with repetition, the task-engaged cells are becoming more selectively tuned, with the largest decreases occurring in cells that are poorly responsive and/or weakly tuned to the repeated stimuli. In contrast, cells that are the most responsive and selective to the repeated stimuli maintain their firing rates. When combined, bottom-up support would be removed for alternative or competing representations in downstream brain regions, allowing more rapid propagation of stimulus-selective activity throughout task-engaged neural pathways, as well as faster and more accurate behavioral responses.

While these models are not necessarily mutually exclusive, each makes unique predictions that can be tested. The Synchrony, Predictive Coding, and Sharpening models differ from one another with regard to their predictions about the effect of repetition on connectivity between brain regions. For example, the Synchrony model predicts increased positive inter-regional coupling with repetition. The Predictive Coding model, in contrast, would seem to predict that top-down connections should have stronger negative coupling with repetition and that this coupling should correlate with the magnitude of repetition suppression in the regions receiving this input (though see[24]). The Sharpening model, like the Synchrony model, predicts an increase in positive coupling, but it further requires the sending region to exhibit repetition suppression, accompanied by a reduction in the similarity of neural responses in the sending region due to reduced overlap among neural representations. The Facilitation model is less clear regarding predictions about connectivity but predicts that neural responses in regions exhibiting repetition suppression should peak and terminate earlier for repeated objects, with a similar prediction made by the Predictive Coding model under conditions of high model precision[21]. Finally, each model predicts that behavioral priming magnitude should be related to its core changes, with repetition suppression serving as an indirect marker of the operation of its critical underlying mechanisms. A strength of BOLD fMRI is its potential for examining changes in connectivity at the scale of the whole brain, as well as its ability to evaluate local changes in representational similarity, germane to predictions of the Synchrony, Predictive Coding, and Sharpening models. The ability of BOLD fMRI to adjudicate subtle changes in the timing of neural activity is less clear, but we nevertheless examine the predicted timing changes of the Facilitation and Predictive Coding models in the event that relevant timing information is detectable. What this means is that it will not be possible to test all four models on equal terms, but the results will have the potential to favor certain models over others, as well as to clarify the relationships between changes in connectivity, repetition suppression, and representational similarity.

We test each of these predictions in the context of an object identification task in fMRI. Participants were required to name

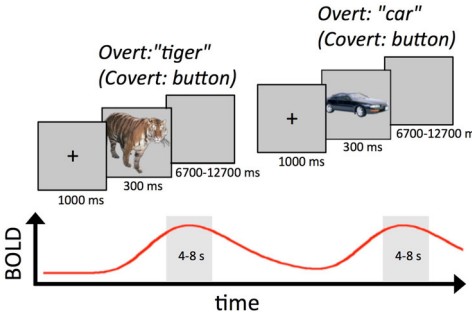

**Fig. 2 Overt and covert picture naming tasks.** Participants were instructed to name pictures out loud (Overt Naming) or silently to themselves, pressing a button to mark the naming time (Covert Naming). Trials were structured in a slow event-related design in order to separate the peak responses of individual trials, and jitter of 8–14 s occurred from trial onset to trial onset. Name responses were marked for correctness and the onset time of the voice/button response was recorded for each trial. For analyses of local activity, an empirical hemodynamic response function was estimated across trials with a separate regressor at each timepoint following stimulus onset. For connectivity analyses, the peak BOLD response was calculated for each trial and in each fMRI voxel by averaging the 2 timepoints (4–6 and 6–8 s) adjacent to the expected peak of the hemodynamic response function, with a single peak value saved for each trial in order to minimize the contribution of the temporal contour of the evoked responses from functional and effective connectivity estimates.

pictured common objects (Fig. 2). Overt picture naming has frequently been used in the repetition priming literature[1,5,25], with the advantages that the priming effects are large and correct responses are unlikely to be due to guessing (in contrast to alternative-forced-choice tasks). However, given the risks of motion-related artifacts when employing overt verbal responding, Covert Naming[18,26], in which participants named objects silently while pressing a response button to mark the naming onset, was also included as a form of artifact control. Features of neural activity that do not differ across Overt and Covert Naming cannot be due to overt speech artifacts. Given the importance of between-region connectivity to the predictions being tested, a novel form of fMRI task-based connectivity was designed to minimize contamination by the temporal contour of the task-evoked response to functional and effective connectivity measures. A slow event-related design was used, permitting relatively clean isolation of the peak response to each individual trial compared to rapid event-related designs. The response peak was then notched out of the time series, with connectivity calculated as the co-fluctuation (correlation) of peak responses across trials between pairs of regions/voxels.

## Results

### Regions engaged in object naming and showing repetition effects.
Task analyses first identified brain regions commonly engaged in Overt and Covert Naming (32 and 28 participants, respectively; see "Methods"). Voxels with above-baseline responses in both tasks are shown in Fig. 3a at two levels of significance, one with a minimum level of significance ($P < 0.05$, false discovery rate (FDR)-corrected to $q < 0.05$; shown in orange) and one with a more stringent level of significance in both tasks individually, for which responses can be said to replicate across tasks ($P < 0.0001$, $q < 0.00016$ in each task; shown in red). As in previous studies of picture naming responses[12,26–28], these regions included left and right lateral prefrontal cortex, bilateral occipitotemporal and ventral temporal cortex, bilateral

intraparietal cortex, anterior cingulate cortex (ACC), thalamus, striatum, and cerebellum.

Repetition effects were then calculated in all voxels showing above-baseline task responses, as all models being tested posit that the relevant changes occur among cell populations engaged by the task. Regions exhibiting repetition effects, either repetition enhancement or repetition suppression, are shown in Fig. 3b at two levels of significance. At a minimum level of significance ($P < 0.05$, $q < 0.05$), most of the voxels responding above baseline exhibited repetition suppression (light blue), with more restricted locations showing repetition enhancement, including cingulate cortex just dorsal to the posterior portion of the corpus callosum, bilateral cuneus, right posterior superior temporal gyrus (STG), and bilateral putamen. At a more stringent level of significance ($P < 0.00001$, $q < 0.00006$, in each task individually), only repetition suppression was observed across both tasks in four large regions (dark blue): left frontal, bilateral fusiform, and ACC, consistent with previous studies of repetition suppression in object naming[12,26,27]. Given the concordance of responses in these regions in both Overt and Covert Naming, these findings cannot be easily attributed to overt speech artifacts that would only be present in Overt Naming.

Correlations between repetition suppression and priming were conducted across all participants (Overt and Covert Naming participants combined; button-press response times used for Covert Naming) in the four large Repetition Suppression (RS) regions (dark blue in Fig. 3b). A significant correlation was observed in the left frontal region, with larger RS associated with larger priming in terms of effect size (Cohen's $d$) [Pearson's $r$ (58) = 0.3674, $P < 0.004$, corrected; Fig. 3c]. Non-significant trends were also observed for the left fusiform and ACC regions ($P < 0.065$ and $P < 0.095$, respectively). This result replicates previous studies showing robust correlations between RS and priming in left frontal cortex[29–31].

### Repetition priming and Primeability in object naming.
Most previous studies of correlations between neural activity and repetition priming have examined correlations across participants (as in Fig. 3c), rather than making use of single-trial responses to examine correlations within-participant. This is usually necessary in fMRI because single-trial neural responses are largely overlapping in rapid event-related experiments. The slow event-related design used in the current experiment allowed us to isolate single-trial neural responses, permitting examination of both across-participant and within-participant priming relationships (see[32] for related discussion).

Each object was named three times approximately 30 min prior to scanning (see "Methods"). In order to design an appropriate within-participant measure of priming magnitude, we first examined the reliability of these pre-fMRI naming response times. The single-participant response times to each object across these pre-fMRI presentations were highly variable, only becoming more reliable when either averaging across participants (Fig. 4a, b and Supplementary Fig. 1) or when pooling objects within-participant into larger groupings (Fig. 4c). This fact, combined with the need to pool trials into larger groups anyway in order to calculate connectivity estimates across trials (with only a single datapoint contributed by each trial), led us to adopt a median split of trials into strongly and weakly primed objects (see "Methods" for details). The selection of objects into strongly and weakly primed groupings was actually based on the group-average naming time to an object when NEW, as this variable was strongly and directly related to eventual priming magnitude (Fig. 4b). Since this distinction is defined on normative group measurements rather than on an individual participant's

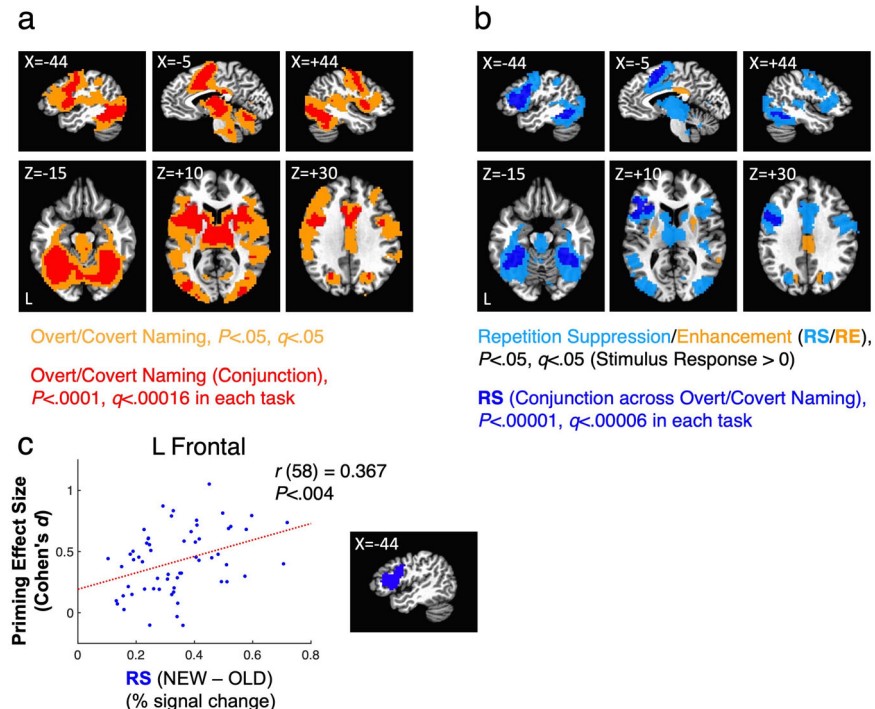

**Fig. 3 Regions engaged in object naming and showing repetition effects. a** Above-baseline responses to Overt and Covert Naming tasks are shown at two levels of significance, one at a minimum level of significance (pooling the tasks, $P < 0.05$, FDR $q < 0.05$; shown in orange) and another at a more restrictive level of significance ($P < 0.0001$, FDR $q < 0.00016$ in each task individually) for which responses can be said to replicate across tasks (shown in red). **b** Repetition effects, either repetition suppression (blue colors) or repetition enhancement (orange), are shown at two levels of significance, one at a minimum level of significance (pooling the tasks, $P < 0.05$, FDR $q < 0.05$; suppression shown in light blue, enhancement shown in orange) and another at a more restrictive level of significance ($P < 0.00001$, FDR $q < 0.00006$ in each task individually). Repetition effects were masked by the less restrictive threshold in (**a**), as all theories being evaluated make claims about regions engaged at above-baseline levels during the tasks. Statistics in (**a**) and (**b**) are based on $N = 32$ independent participant datasets for Overt Naming and $N = 28$ independent datasets for Covert Naming. **c** Repetition suppression was highly correlated (Pearson) with priming magnitude in left frontal cortex in terms of effect size (Cohen's $d$) across $N = 60$ independent participants, combining both Overt and Covert Naming (corrected for multiple comparisons by both Bonferroni and FDR). The left frontal region is the same as the dark blue region showing Repetition Suppression (RS) at the more stringent threshold in (**b**).

measured priming, we refer to it as "Primeability" rather than priming, in analogy to recent studies of "memorability" as an item property for explicit recollection[33]. Observed priming magnitude for each participant on strongly primeable and weakly primeable objects was then quantified by effect size (Cohen's $d$) in order to adjust for differences in raw response time per participant and per condition. As expected, based on the object grouping scheme, strongly primeable objects exhibited larger priming effect sizes than weakly primeable objects, both for Overt Naming times during fMRI and button-press response times for Covert Naming during fMRI (Fig. 4c). When repetition suppression is recalculated separately for Strong and Weak Primeable objects in the four large RS regions, significant effects of Primeability are observed in left frontal, left fusiform, and ACC regions, with larger RS observed for strongly primeable objects (Fig. 5). These results provide evidence that RS and priming magnitude are related in regions outside of left frontal cortex, including the left fusiform gyrus and the ACC (Fig. 5a, b).

**Functional connectivity changes related to Primeability.** Rather than restricting the evaluation of functional connectivity changes to those regions exhibiting repetition suppression or enhancement, we instead performed a data-driven, whole-brain search at the voxel level using "connectedness"[34]. This approach simplifies the bivariate map of correlations among all possible pairs of voxels into a univariate map of average correlations, with each

voxel's value representing the average functional connectivity level with a desired cohort of voxels, in this case the set of all task-responsive voxels (see "Methods"). Effects in whole-brain connectedness can then identify effective seeds for more traditional seed-based analyses, thereby detecting a more complete set of regions exhibiting effects in functional connectivity.

Voxelwise connectedness estimates were calculated for each participant in four conditions: Strong Primeable OLD, Strong Primeable NEW, Weak Primeable OLD, and Weak Primeable NEW (e.g., a Strong Primeable OLD object corresponded to an object named pre-fMRI and that was strongly primeable, i.e., had a slower than median response time in the normative picture naming data; a Strong Primeable NEW object corresponded to an object that was strongly primeable based on the normative data, but had not been previously seen). These voxelwise estimates were then entered into a linear mixed effects (LME) model with factors of Task (Overt, Covert), Primeability (Strong, Weak), and Repetition (OLD, NEW), covarying the global correlation level, GCOR[35], per condition and participant as a measure of residual whole-brain artifacts. There was no overall main effect of Repetition that survived whole-brain correction, but there was a significant interaction between Repetition and Primeability in the right temporoparietal (R TP) cortex ($P < 0.001$, corrected to $P < 0.025$; Fig. 6a). The 3-way interaction with Task failed to reach significance in this or any other location, which is to say that the interaction between Repetition and Primeability was not found to differ significantly in Overt versus Covert Naming (see also

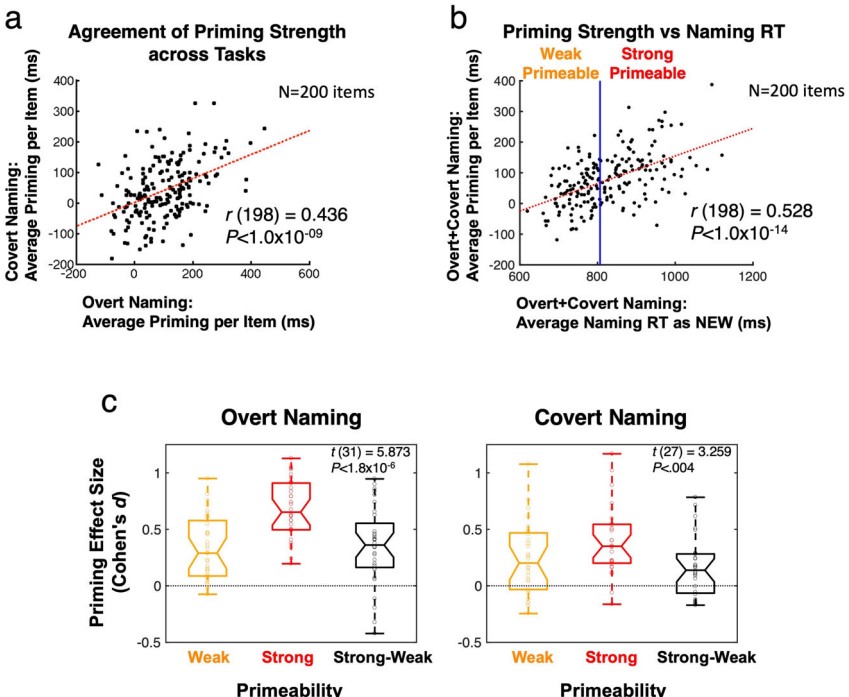

**Fig. 4 Repetition priming and Primeability. a** When averaged across participants within each task, priming magnitudes specific to each of the 200 stimuli are highly reliable across task. For the Covert Naming task, these priming magnitudes were assessed in a post-fMRI session during which participants overtly named all stimuli (either pre-exposed as OLD or novel to the fMRI session as NEW). **b** When responses are averaged across both participants and tasks, a strong relationship was observed between average response time (RT) to an object when NEW and the subsequent priming magnitude observed. Items were subjected to a median split based on the RT when NEW (using the group-average normative data) to classify objects as either Strong Primeable (slow RT) or Weak Primeable (fast RT), permitting a within-participant measure of "Primeability". Statistics in (**a**) and (**b**) are based on $N = 200$ independent stimuli. **c** When grouping objects by Primeability (either Strong or Weak), priming effect sizes for each participant were indeed greater for Strong compared to Weak Primeable objects in the Overt Naming task (left panel); this is expected since these responses contributed to the original calculation of Primeability. However, this same relationship was also found for the Covert Naming button-press response times during fMRI, which were not used in calculating Primeability (right panel). The middle horizontal line in each box plot represents the median (50th %ile), the horizontal lines just above and below the median represent the 25th and 75th %iles, the top and bottom horizontal lines represent the minimum and maximum values, and the boundaries of the horizontal notches inside the 25th and 75th %iles depict the 95% confidence limits of the median. Individual datapoints are plotted as open circles. Statistics in (**c**) are based on $N = 32$ independent participant datasets for Overt Naming and $N = 28$ independent datasets for Covert Naming. For related content in SI, see Supplementary Fig. 1.

Supplementary Fig. 2). Using this temporoparietal region as a seed and testing a corresponding seed-based LME model, significant Repetition × Primeability interactions ($P < 0.001$, corrected to $P < 0.025$) were observed with the right fusiform gyrus, the anterior cingulate, right STG, and right putamen. To these regions, we added all regions exhibiting significant repetition suppression that replicated across tasks, given their importance to all theories being tested (Fig. 6b). This led to a total of 7 regions of interest (ROIs), sampled as spheres centered on the peak statistic from each region: 3 ROIs identified as showing effects in functional connectivity (right temporoparietal, right STG, and right putamen), 2 ROIs showing repetition suppression effects (left frontal cortex and left fusiform gyrus), and 2 ROIs showing both functional connectivity and repetition suppression effects (ACC and right fusiform gyrus).

The ROI-level data were submitted to an additional LME analysis, affording the assessment of all region-by-region effects of repetition on functional connectivity, as well as interactions between Task, Primeability, and Repetition. As with the voxelwise data, there was no overall effect of Repetition among the ROIs. There was a significant interaction of Repetition and Primeability involving the same regions as detected in the whole-brain analysis (as expected), but there was also an additional interaction detected between the ACC and right fusiform gyrus (all $P < 0.0099$, $q < 0.05$; Fig. 6c). When evaluating the qualitative nature

of these interactions by separately assessing Repetition in the Strong and Weak Primeable conditions, greater functional connectivity was observed for OLD compared to NEW objects in the Strong Primeable condition (R TP ROI with ACC, right putamen, and right STG ROIs), whereas weaker functional connectivity was observed for OLD compared to NEW objects in the Weak Primeable condition (also R TP ROIs with ACC, right putamen, and right STG ROIs) (all $P < 0.0287$, $q < 0.05$). As with the whole-brain data, none of these effects exhibited a further interaction with Task (Overt versus Covert Naming).

**Effective connectivity changes related to Primeability and Repetition suppression.** The functional connectivity effects were then further probed with estimates of effective connectivity among the 7 ROIs. This is important because functional connectivity measures are ambiguous with respect to claims about underlying coupling between regions, as localized real changes in coupling can manifest indirectly at other locations through polysynaptic interactions (see[36] for discussion). We therefore applied structural equation modeling (SEM), a form of effective connectivity estimation that utilizes the pattern of correlation among ROIs in order to evaluate directional changes in underlying inter-regional coupling[37]. We first performed a data-driven search for the optimal SEM model while pooling all data conditions (Task, Primeability, Repetition) (see "Methods"). The

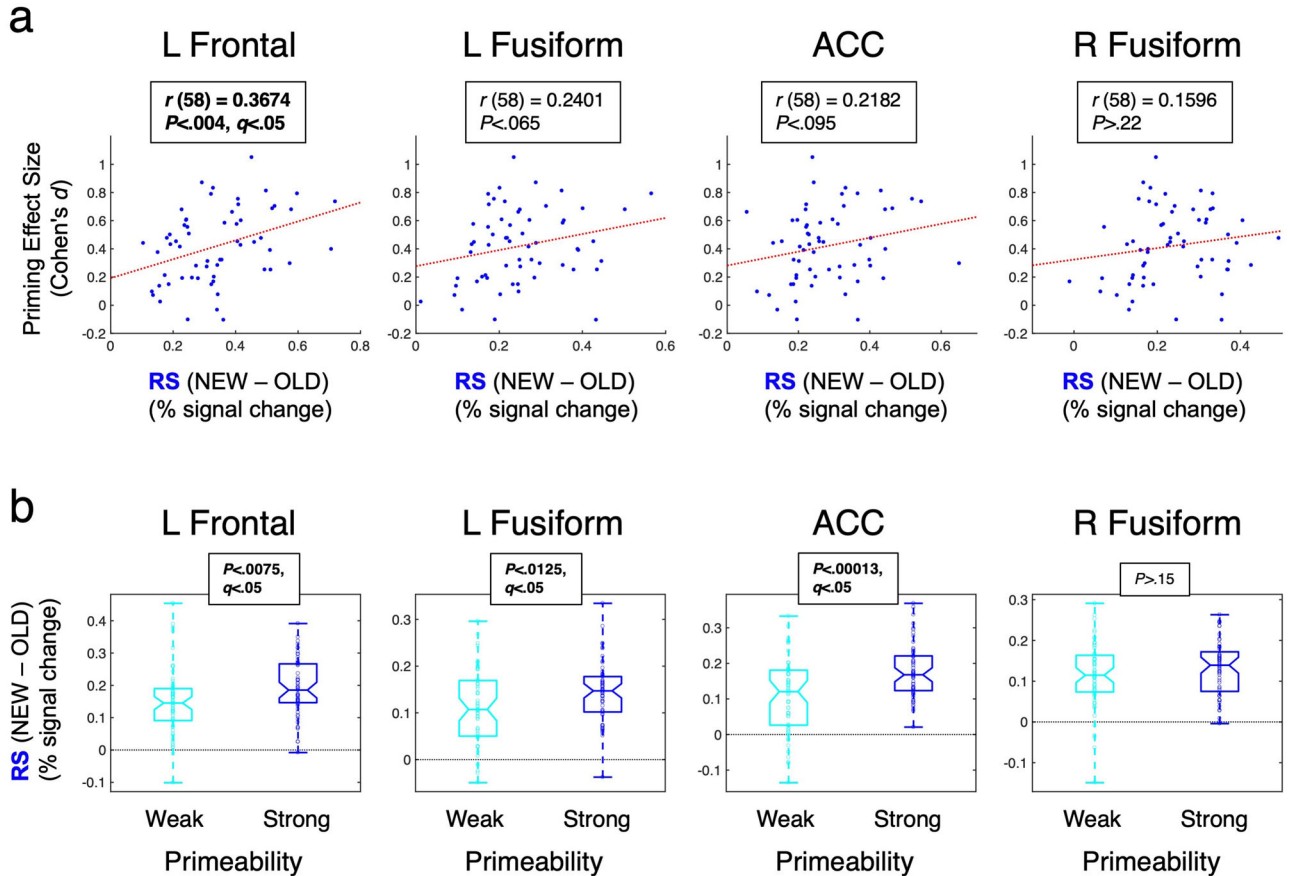

**Fig. 5 Across- and within-participant relationships between repetition suppression and priming. a** Correlations (Pearson) across participants between Repetition Supression (RS) magnitude (NEW–OLD) and priming effect size (Cohen's *d*) reveal a significant relationship in left frontal cortex and non-significant trends in left and right fusiform regions (left-most plot same as Fig. 3c). **b** Separating trials into Strong and Weak Primeable conditions show that significant within-participant relationships between RS and priming are found in left frontal, left fusiform, and ACC regions (greater RS in the Strong Primeable condition by paired *t*-tests). The middle horizontal line in each box plot represents the median (50th %ile), the horizontal lines just below and above the median represent the 25th and 75th %iles, the bottom and top horizontal lines represent the minimum and maximum values, and the boundaries of the horizontal notches inside the 25th and 75th %iles depict the 95% confidence limits of the median. Individual datapoints are plotted as open circles. Statistics (paired *t*-tests) are based on *N* = 60 independent participants (combining Overt and Covert Naming), and multiple comparisons were corrected by FDR (*q* < 0.05).

optimal model was then parameterized for each participant's data per condition, with model parameters tested using the same LME modeling approach as for tests of functional connectivity. The optimal model is shown in Fig. 7a, with arrows indicating non-zero parameters and the direction of the causal interactions among the ROIs. In all, 10 parameters were found to differ significantly from zero, with all connections being positive (above zero) and with no connections showing negative parameters that would be consistent with inhibition or some form of suppression in any of the conditions. Connections exhibiting a significant Repetition × Primeability interaction are shown as thick red arrows (*P* < 0.0063, *q* < 0.05 for all), with non-significant inter-actions shown in black. Figure 7b clarifies the nature of these interactions. The connection from the R TP ROI to the ACC ROI exhibited a significantly larger parameter for OLD than NEW objects for the Strong Primeable condition (*P* < 0.0035, *q* < 0.05), with no significant difference observed for the Weak Primeable condition. In contrast, the connections from the ACC to the right fusiform ROI and the right STG to the R TP ROI exhibited no significant OLD/NEW effect for the Strong Primeable condition and significantly smaller parameter values for OLD compared to NEW objects in the Weak Primeable condition (*P* < 0.0091, *q* < 0.05 for both). Of these connections, the one from the R TP to the

ACC ROI appeared to provide the best candidate for a correlate of priming magnitude, with the strongest effects observed in the Strong Primeable condition (compare parameter values in Fig. 7c to priming measures in Fig. 4c). We therefore examined the a-cross-participant correlation between priming effect sizes and the R TP to ACC ROI parameter values. There was a significant correlation between the contrast in Strong versus Weak Prime-able effect sizes in priming (behavioral data) and the SEM parameter values (fMRI data) across participants [Pearson's *r* (56) = 0.3303, *P* < 0.0114]. This correlation appeared to be driven by the data in the Strong Primeable condition [Pearson's *r* (56) = 0.3002, *P* < 0.0222; for Weak Primeable con-dition: *r* (56) = 0.0700, *P* > 0.6].

These data are most compatible with the Synchrony model, for which coupling is predicted to be greater (and positive) for the OLD compared to the NEW condition, particularly for Strongly Primeable objects. The increased coupling for the OLD condition also occurred despite no change in activity levels in the R TP ROI and significantly reduced activity in the ACC ROI (see SI, Supplementary Fig. 3). While the Sharpening model similarly predicts greater positive coupling for OLD items, the lack of any discernible repetition suppression in the putative sending region (R TP) fails to provide support for this model. The data are also

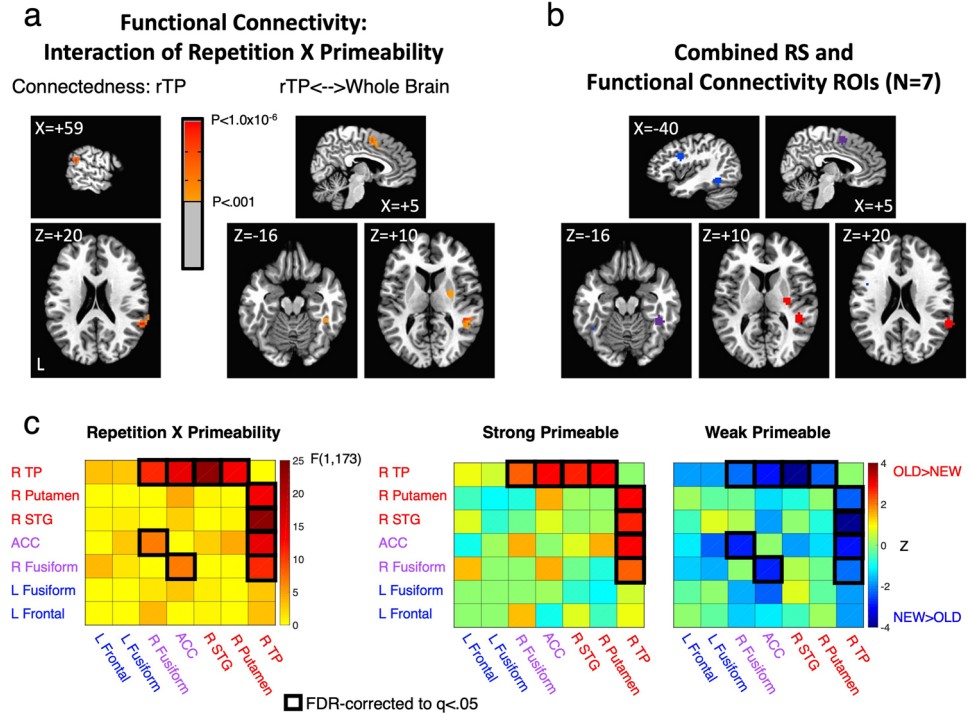

**Fig. 6 Functional connectivity shows interaction between Primeability and Repetition. a** A right temporoparietal (R TP) region exhibited an interaction in whole-brain connectedness between Repetition and Primeability, similar to that seen in the behavioral priming results ($P < 0.001$, corrected to $P < 0.025$). When used as a seed, R TP jointly exhibited this interaction with the anterior cingulate (ACC), right putamen, right STG, and the right fusiform gyrus. **b** Regions showing a Repetition × Primeability interaction were combined with regions showing repetition suppression in both Overt and Covert Naming Tasks. **c** Region-by-region functional connectivity interactions of Repetition × Primeability are shown for all 7 regions (left panel). FDR-corrected effects are indicated by black boxes ($P < 0.01$, $q < 0.05$). Region-by-region comparisons of OLD versus NEW functional connectivity are shown for Strong and Weak Primeable conditions separately in the right panels. Warm colors (red) indicate OLD > NEW and cool colors (blue) indicate NEW > OLD. FDR-corrected effects ($P < 0.0286$, $q < 0.05$) were calculated among all region-by-region combinations showing a significant Repetition × Primeability interaction, indicated by black squares. Statistics are based on $N = 60$ independent participants, organized into a factorial design with Task (Overt, Covert Naming) as a between-participant variable. For related content in SI, see Supplementary Fig. 2 and Supplementary Table 1.

less compatible with the predictions of the Predictive Coding model, in which top-down connections are claimed to be more strongly negative following repetition, leading to repetition suppression in the receiving area[16,19,21]. As noted, none of the model connections in any of the conditions were found to be negative. Nevertheless, two of the ROIs exhibiting significant repetition suppression (ACC and right fusiform gyrus) were found to have incoming SEM connections modulated by stimulus repetition (from R TP and ACC ROIs, respectively). We therefore examined the relationship between SEM parameter values at these connections and the magnitude of repetition suppression observed in the receiving ROIs. A contrast of repetition suppression in Strong minus Weak Primeable conditions failed to correlate with the corresponding contrast of the SEM parameters (OLD–NEW parameter values in the Strong Primeable condition minus the same difference in the Weak Primeable condition), either for the connection from R TP to ACC [$r$ (56) = 0.0897, $P > 0.5$] or for the connection from ACC to the right fusiform gyrus [$r$ (56) = $-0.0429$, $P > 0.74$]. These results, taken together with the lack of modulation by stimulus repetition of connections into other ROIs exhibiting repetition suppression (e.g., left frontal and left fusiform ROIs), suggest that changes in coupling due to repetition are largely independent of local repetition suppression magnitudes. In order to establish this relationship more clearly for the one connection correlated with behavioral priming magnitude, R TP to ACC, we used partial correlation to remove the magnitude of repetition suppression exhibited in the ACC from the correlation between priming effect

size and SEM parameter contrasts in the Strong versus Weak Primeable conditions. This partial correlation remained at approximately the same level as without the partialling [partial $r$ (55) = 0.334, $P < 0.0112$]. We further asked whether the connection from R TP to ACC remained correlated with priming after partialling overall RS observed in the left frontal ROI, which was previously shown to correlate with priming (Figs. 3c and 5a). The partial correlation again remained virtually unchanged [partial $r$ (55) = 0.329, $P < 0.0125$]. Taken together, these results establish that changes in coupling between R TP and ACC are correlated with priming magnitude in a manner largely independent of the magnitude of repetition suppression, with each accounting for unique portions of the priming variance.

**Testing predictions regarding changes in representational similarity.** In the previous analyses, we found that functional and effective connectivity effects due to repetition were consistent with predictions of the Synchrony model and failed to provide support for the Predictive Coding model. One aspect of these results also failed to support the Sharpening model, namely that regions showing repetition suppression (L Frontal, L Fusiform, R Fusiform, and ACC ROIs) failed to exhibit increased feedforward coupling during the processing of OLD relative to NEW objects.

The Sharpening model further predicts that the spatial similarity of neural responses should decrease with repetition as activity to OLD stimuli becomes "sharper" and overlaps less across objects. We evaluated this prediction using multi-voxel

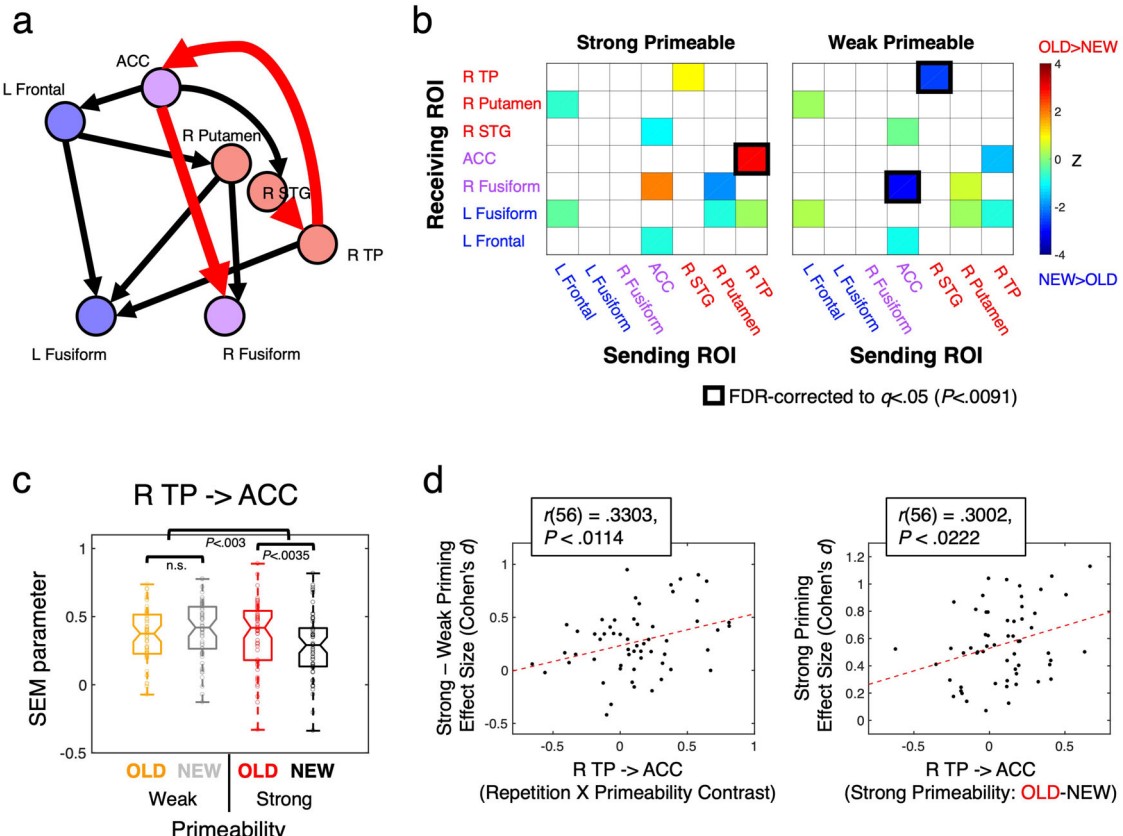

**Fig. 7 Effective connectivity increases between R TP cortex and ACC correlate with priming magnitude. a** Structural equation modeling (SEM) was used to estimate effective connectivity among the 7 regions. The optimal 10-parameter SEM model is shown, with arrows indicating causal directionality and connections exhibiting a significant Repetition × Primeability interaction ($P < 0.0063$, $q < 0.05$) shown with thick red arrows (non-significant interactions shown with black arrows). **b** Region-by-region comparisons of SEM parameters for OLD versus NEW objects are shown separately for the Strong and Weak Primeable conditions. For directionality, sending ROIs are listed along the $x$-axes and receiving ROIs are listed along the $y$-axes. FDR-corrected OLD/ NEW comparisons ($P < 0.0091$, $q < 0.05$) were assessed among connections exhibiting a Repetition × Primeability interaction, indicated with black squares. **c** The connection from the R TP ROI to ACC ROI shows a Repetition × Primeability interaction that is consistent with the Synchrony model (with increased coupling for OLD objects in the Strong Primeable condition). The middle horizontal line in each box plot represents the median (50th %ile), the horizontal lines just below and above the median represent the 25th and 75th %iles, the bottom and top horizontal lines represent the minimum and maximum values, and the boundaries of the horizontal notches inside the 25th and 75th %iles depict the 95% confidence limits of the median. Individual datapoints are plotted as open circles. **d** The R TP to ACC connection further exhibited a correlation across participants with observed priming magnitude, assessed by effect size (Cohen's $d$). This correlation appeared to be driven by the Strong Primeable condition (rightmost panel), with priming effect size in the Strong Primeable condition correlated with the difference between OLD and NEW SEM parameters in the Strong Primeable condition. Statistics are based on $N = 60$ independent participants, organized into a factorial design with Task (Overt, Covert Naming) as a between-participant variable (Primeability and Repetition are both within-participant variables). For related content in SI, see Supplementary Fig. 3.

pattern analysis[38] (MVPA) applied to the 4 repetition suppression ROIs. The pattern of peak BOLD responses across voxels in an ROI to each trial was correlated (Pearson) with all correct trials of the same type (e.g., Strong Primeable OLD trials) (see "Methods"). As shown in Fig. 8a, the inter-item correlation level was indeed lower for OLD compared to NEW trials ($Z > 4.352$, $P < 1.4 \times 10^{-5}$ for all 4 ROIs) with a significantly larger decrease for Strong compared to Weak Primeable conditions in the Left Frontal [Repetition × Primeability $F(1,174) = 23.259$, $P < 3.1 \times 10^{-6}$, FDR $q < 0.05$] and ACC ROIs [Repetition × Primeability $F(1,174) = 5.530$, $P < 0.0199$, FDR $q < 0.05$]. However, on further examination, these effects appeared to be related to average beta-weight levels and estimates of the signal-to-noise ratio (SNR) of the peak responses (see SI, Supplementary Fig. 4). When covarying these quantities, only the Repetition × Primeability interaction in the Left Frontal ROI remained significant [$F(1,172) = 16.906$, $P < 6.1 \times 10^{-5}$, FDR $q < 0.05$], driven by an OLD/ NEW difference in the Strong Primeable condition [$Z = 3.607$,

$P < 3.2 \times 10^{-4}$ vs Weak: $Z = -0.649$, $P > 0.5$] (Fig. 8b). However, even this last effect remains in doubt, given the strong dependence on beta-weight levels and SNR for this ROI [β: $F(1,172) = 16.117$, $P < 8.9 \times 10^{-5}$; SNR: $F(1,172) = 19.234$, $P < 2.1 \times 10^{-5}$]. In other words, the unadjusted MVPA results are consistent with the predictions of the Sharpening model, but the results may simply reflect the average amplitude of the BOLD response rather than the pattern of responses across voxels, per se, with response levels nearer to zero being more contaminated with noise (see also[39–41] for discussion). Consistent with this interpretation, there was no correlation between MVPA measures and behavioral priming magnitudes across participants in the Left Frontal and Right Fusiform ROIs either prior to or after adjustment ($| r | < 0.1$, $P > 0.56$ for all).

**Testing predictions regarding changes in timing of activity.** The Facilitation model predicts that the timing of neural activity, particularly the timing of the peak response[22,23], should track

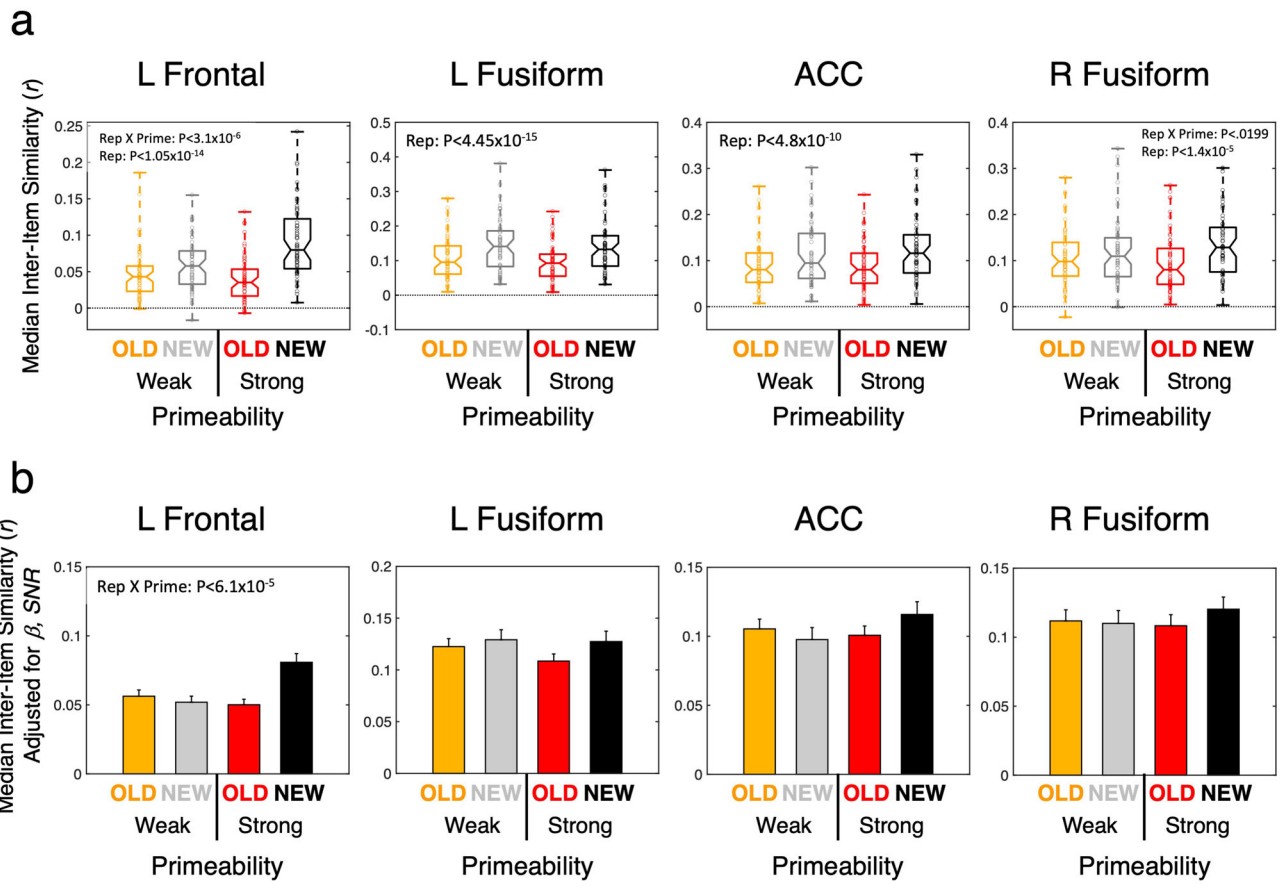

**Fig. 8 MVPA tests of the Sharpening model. a** Spatial correlations of peak responses on each individual trial were calculated across all trials per condition type using the full extent of the four Repetition Suppression (RS) clusters (see Supplementary Table 1), retaining the median inter-item correlation per participant. A significant Repetition × Primeability interaction was observed for the Left Frontal and ACC ROIs (corrected by FDR, $q < 0.05$). The middle horizontal line in each box plot represents the median (50th %ile), the horizontal lines just below and above the median represent the 25th and 75th %iles, the bottom and top horizontal lines represent the minimum and maximum values, and the boundaries of the horizontal notches inside the 25th and 75th % iles depict the 95% confidence limits of the median. Individual datapoints are plotted as open circles. **b** Strong covariation of the spatial correlations with average beta coefficients and estimated signal-to-noise ratio eliminated most of the differences between conditions after adjustment for these variables, leaving only a significant Repetition × Primeability interaction in the Left Frontal ROI. Bar plots are used along with standard error of the mean (SE) estimates, since these adjusted means and model-residual error are not defined on the individual participants and only on the full LME model. Statistics are based on $N = 60$ independent participants, organized into a factorial design with Task (Overt, Covert Naming) as a between-participant variable (Primeability and Repetition are within-participant variables). For related content in SI, see Supplementary Fig. 4.

repetition suppression and behavioral priming magnitudes. Similarly, under conditions of high model precision and enhanced processing gain, the Predictive Coding model explains priming through a speeding up of neural activity, with this speeding possibly generating an additional component of repetition suppression[21]. As discussed above, fMRI is not the optimal technique to evaluate this prediction, but we nevertheless examined changes in the timing of the BOLD response in the event that such changes were detectable. To evaluate this, we extracted the beta coefficients for each participant at each timepoint following stimulus onset in each of the ROIs exhibiting repetition suppression. A continuous response function was fit to these datapoints in each of the four conditions (OLD, NEW × Strong, Weak Primeable), affording separate estimates of the timing and amplitude of the peak BOLD response (see "Methods"). Figure 9 shows the estimated hemodynamic response functions for each of the 4 repetition suppression ROIs, along with estimates of the timing of the peak responses (shown as vertical dotted lines). There was no significant Repetition × Primeability interaction in the timing of the peak responses in any of the 4 ROIs ($P > 0.27$ for all), but there was a significant main effect of Repetition on the timing of the peak in the Left Fusiform ROI [mean (SE) OLD

$t_{peak} = 4.382$ (0.0967) s; NEW $t_{peak} = 5.004$ (0.091) s; $F(1,174) = 9.233$, $P < 0.0028$, FDR $q < 0.05$] and an uncorrected effect of Repetition in the Left Frontal ROI [OLD $t_{peak} = 4.647$ (0.168) s; NEW $t_{peak} = 4.979$ (0.1257) s; $F(1,174) = 4.849$, $P < 0.03$, FDR $q > 0.05$]. The earlier peak timing of OLD relative to NEW trials in the Left Fusiform ROI (with a similar uncorrected effect in the Left Frontal ROI) is consistent with predictions of the Facilitation and Predictive Coding models. However, there was no interaction with Primeability, failing to provide a correlation with priming magnitude.

## Discussion
In the current study, we have tested predictions from four prominent models of repetition suppression and priming (Synchrony, Predictive Coding, Sharpening, and Facilitation) regarding changes in connectivity, representational similarity, timing of the BOLD response, and the relationship of these quantities to repetition suppression and priming magnitude. We performed data-driven, whole-brain analyses of changes in functional and effective connectivity related to a within-participant measure of priming magnitude we refer to as

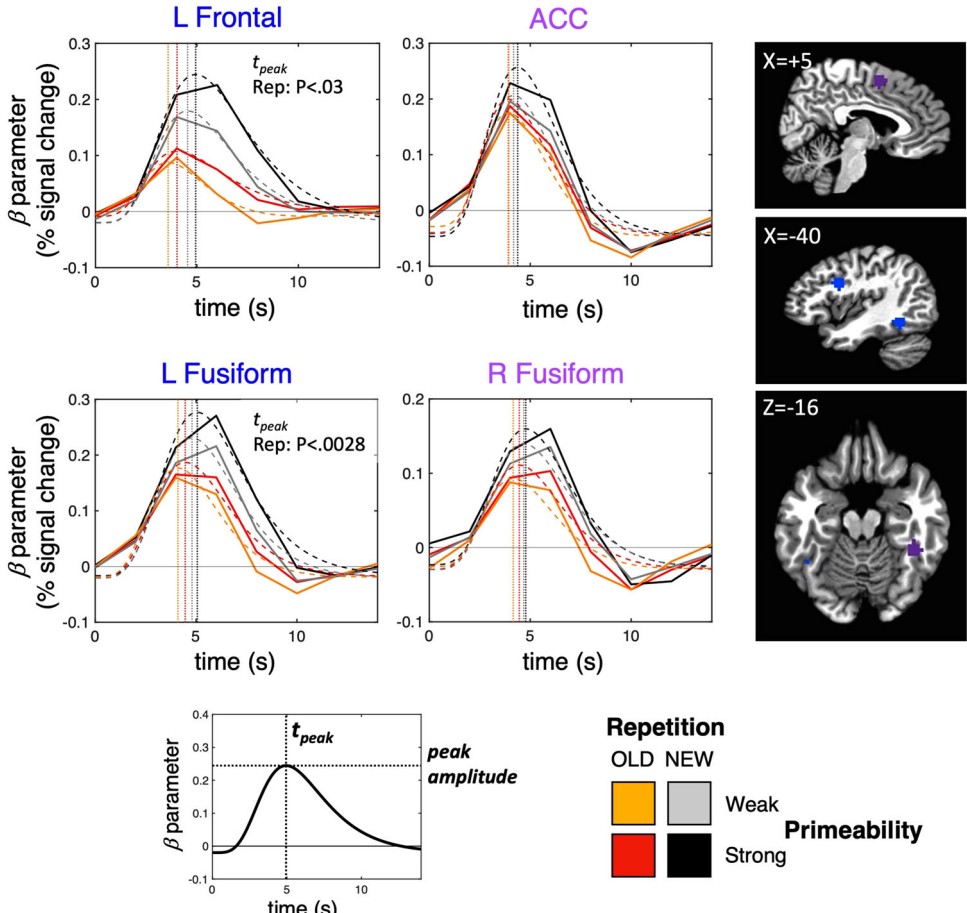

**Fig. 9 Assessing activity timing predictions of the Facilitation and Predictive Coding models.** A hemodynamic response function model (gamma variate) was fit to the beta coefficients at each timepoint (TR) for each participant and experimental condition, permitting estimates of the peak time ($t_{peak}$) (see graphic at the bottom). Group-average response functions are shown with dashed lines for each condition, and group-average estimates of peak times are shown with vertical dotted lines (mean data on the actual measured beta coefficients are shown with solid lines). Main effects of Repetition on peak time were observed in the Left Fusiform and Left Frontal ROIs (OLD peaks earlier than NEW peaks), but there were no significant interactions between Repetition and Primeability on peak time. Statistics are based on $N = 60$ independent participants, organized into a factorial design with Task (Overt, Covert Naming) as a between-participant variable. A color key for the experimental conditions is shown at the bottom right (OLD, Strong Primeable = red; OLD, Weak Primeable = orange; NEW, Strong Primeable = gray; NEW, Weak Primeable = black).

"Primeability", as these changes were central to three of the four models. We then tested predictions about the decreased similarity of neural responses derived from the Sharpening model, as well as the timing of neural responses derived from the Facilitation and Predictive Coding models. Changes in functional and effective connectivity were most consistent with predictions of the Synchrony model, with increased coupling from R TP cortex to the ACC for OLD relative to NEW objects, particularly for the Strong Primeable condition. This occurred despite no change in the activity levels in R TP cortex and decreased activity for OLD objects in the ACC, demonstrating increased impact of activity of one region on the other that is consistent with enhanced synchronization of activity. Furthermore, these changes were correlated with across-participant variability in behavioral priming. Top-down negative coupling that increased with repetition, a prediction of the Predictive Coding model[16,19,21], was not observed among any regions, with only positive coupling observed. Finally, changes in coupling were found to be largely independent of repetition suppression magnitudes.

With respect to the Sharpening model, the spatial similarity of neural responses assessed with MVPA interacted with Primeability in a manner qualitatively similar to behavioral priming effect sizes. However, these effects were strongly related to the

average response amplitude and estimates of the SNR in each region, largely eliminating differences in similarity after adjustment for these factors. With respect to predictions of the Facilitation and Predictive Coding models about timing, neural responses were indeed found to peak at earlier times for OLD compared to NEW objects in regions showing repetition suppression, although these effects failed to track priming magnitude. On this point, it is important to mention that none of the original proponents of these models have made explicit claims about differential timing for different types of items (e.g., greater change in the peak timing for more strongly primed items that are identified initially more slowly). Nevertheless, we believe that this prediction should follow if these models are to serve as a unifying explanation of repetition suppression and priming.

All of the models—Synchrony included—failed to account for the details of repetition suppression magnitude as it related to priming. This failure stemmed from the assumption that a unitary mechanism was responsible for both repetition suppression and priming. While we found that priming and repetition suppression were indeed correlated in left frontal cortex (within- and between-participant) and the left fusiform gyrus and ACC (within-participant), the portion of variance explained had little overlap with that explained by changes in coupling. Little to no

changes in functional or effective connectivity were observed among regions showing repetition suppression, despite quite large repetition suppression effects that replicated across tasks (approximately 40–50% decreases in left frontal and bilateral fusiform regions). Instead, coupling changes were observed in a partially overlapping, right-lateralized network of regions. Models of priming will therefore all require some substantive revision, and repetition suppression itself will require some rethinking. If repetition suppression does not reflect changes in coupling between regions, perhaps it reflects some form of local, regional change in processing efficiency that benefits object identification? In any case, a central finding of the current study is that repetition suppression and coupling each account for unique portions of the variance in priming and are expressed in only partially over-lapping brain regions. This is reminiscent of recent findings from another study, in which beta-weight correlations with face recognition memory were found locally to traditional face-processing regions, whereas connectivity-based correlations with face memory lay mainly in relationships with regions outside of the traditional face-processing network[42].

**Limitations of the current study**. The failure to observe robust support for the Predictive Coding, Facilitation, and Sharpening models may be at least partially due to methodological factors. The current study was conducted with fMRI, which afforded simultaneous testing of all four major theoretical models. However, fMRI utilizes the BOLD signal rather than the underlying electrophysiological signals, and BOLD fMRI may be blind to effects that occur at the more rapid time scale of milliseconds to hundreds of milliseconds. Ghuman et al.[17] previously observed increased phase locking (12–14 Hz, low beta band) between left frontal and left fusiform cortex for repeated objects, consistent with the Synchrony model, and this increased phase locking was correlated with priming. However, such band-limited effects may be washed out when effectively averaging across all frequency bands, as one would expect when using a single, average signal (as in fMRI). As a further example, in a recent study of short-term priming in picture naming using electrocorticography (ECoG), we found evidence of both earlier neural response onset consistent with the Facilitation model, as well as increased top-down effective connectivity from left frontal cortex to left ventral temporal cortex with repetition within the first 200 milliseconds of processing[43], potentially consistent with the Predictive Coding model. While this study examined only short-term effects (<20 s separating repetitions) and did not have the statistical power to evaluate the relationship between changes in coupling, repetition suppression magnitude, and behavioral priming magnitude across participants, the observation of effects over a much more rapid physiological time scale suggests the need for further studies with higher temporal resolution (such as ECoG, MEG, and joint fMRI/EEG studies). It is similarly unclear whether spatial correlations in fMRI are a sufficient test of the Sharpening model[39,40,44]. However, multiple other studies have tested the Sharpening model using a range of methods in both monkeys and humans, with little supportive evidence to date at the most common time scales used in priming studies (i.e., within a daily session[11,25,44–47]; cf.[48]; see[13] for discussion). In our view, it will take multiple studies across a range of methodologies to fully rule out any of these models, and the primary contribution of the current study is its whole-brain assessment of changes in connectivity and their relationship to repetition suppression and priming magnitude, with more moderate progress on model evaluation.

The current study utilized multiple data-driven, whole-brain analyses, with voxel-wise assessment of repetition suppression/enhancement and functional connectivity. While every effort was made to appropriately correct for multiple comparisons in these analyses, they are inherently exploratory. Future studies should complement this approach with explicit hypothesis testing. As a central example, the detection of increased functional coupling between R TP cortex and the anterior cingulate with repetition is novel to the current study. Rather than viewing this connection in isolation, it is useful to examine its context within the effective connectivity model that included interactions with portions of the striatum (right putamen), STG and the right fusiform gyrus (Fig. 7a). It is possible that this entire circuit functions to retrieve prior stimulus–response associations encoded through cortico-striatal loops[49,50] (see also[51]). Some evidence consistent with the Synchrony model that we previously observed in MEG in covert picture naming was also right-lateralized, with enhanced evoked power in the theta/alpha frequency ranges involving the right fusiform gyrus and right lateral prefrontal cortex[18]. Other studies have highlighted a role for the R TP cortex in both attention- and memory-related contexts as a portion of the ventral attention network[52,53]. Nevertheless, it will be important for future studies, both physiological and neuropsychological, to engage in more explicit hypothesis testing about the role of R TP cortex and the ventral attention network in repetition priming and object processing.

As an additional example, the effective connectivity analyses in the current study utilized exploratory SEM, in which a search was conducted for the optimal model that explained the data while minimizing out-of-sample error. We made efforts to include the most relevant brain regions in the current task, pooling both regions exhibiting repetition suppression and those identified as exhibiting any priming-related connectivity changes. However, there is a risk that effective connectivity relationships can be model- and method-dependent[54], and it is inherently challenging for a statistical model to induce the correct underlying feedforward and feedback influences among highly interconnected cortical regions. Future work should examine alternatives that employ different underlying statistical approaches using the full fMRI time series (e.g., dynamic causal modeling[55]), as well as different measurement strategies that can potentially isolate feedforward/feedback cortical signals, such as layer-specific fMRI (see[56] for one recent example).

In the current study, repetition of stimuli was implicit to the task being performed. However, participants were likely aware that repeated stimuli had been presented previously, and markers of explicit recollection could be present, particularly in the neural repetition effects. Future studies should extend the current work by measuring both priming and measures of explicit memory (e.g., recognition memory) simultaneously in the same participants, which may permit the partialing and/or isolation of the contributions of explicit recollection. In one recent study[27], we directly compared neural repetition effects in priming during object naming and during recognition memory, performed in alternating runs and on different subsets of stimuli but for the same participants. Repetition suppression was largely restricted to the priming condition in frontal and ventral temporal cortex, whereas repetition enhancement was observed across both task contexts in parietal regions that initially responded near or below baseline, with overall elevated responses during recognition memory. At an item level, behavioral measures of recognition memory performance in this previous study are largely unrelated to priming magnitudes measured in the current study (Supplementary Table 2), suggesting that the current brain–behavior correlations with priming are not strongly related to explicit recollection. Nevertheless, it will be important to examine this relationship further, and the lack of an explicit recollection condition in the current study limits our conclusions in this regard.

Finally, the four theoretical models themselves are specified in somewhat general terms that make joint, head-to-head testing difficult. For example, they do not make direct claims about the involvement and role of particular brain regions in priming, but rather apply generally to task-engaged regions and particularly those that exhibit repetition suppression. They also do not necessarily have equivalent levels of complexity or address the same exact family of possible experimental observations. As previous proponents of the Synchrony model[16], we feel that it is important to disclose that while the connectivity pattern observed between the R TP and ACC regions fits with the general expectation of this model, the separation of repetition suppression from connectivity in the current experiment is also problematic. The Synchrony model (along with the Predictive Coding and Sharpening models) has no clear way to explain the correlation between repetition suppression and priming that is not mediated by connectivity changes. Furthermore, we cannot measure spiking activity directly with fMRI, so even the aspects of the current results that provide supporting evidence are necessarily indirect. The Synchrony model, at least as previously articulated[16], is no less guilty of assuming that connectivity changes would be observed among regions showing repetition suppression, although it left open the possibility that regions not showing repetition suppression could also contribute. These failures will require further examination of the underlying mechanistic bases of repetition suppression using methods with higher temporal resolution (e.g., ECoG, simultaneous fMRI/EEG), and they will require rethinking of all of the models, the Synchrony model included.

## Conclusions
We have examined the relationships among behavioral repetition priming, neural repetition suppression, and brain connectivity in relation to the Synchrony, Predictive Coding, Sharpening, and Facilitation models. Repetition suppression and connectivity were found to be largely independent of one another in their contributions to repetition priming. While the connectivity changes are most compatible with the Synchrony model, all current models fail to explain the correlations between priming and repetition suppression that are not mediated by connectivity.

## Methods
**Ethics statement**. Ethics approval for this study was granted by the NIH Institutional Review Board (protocol 93-M-0170, clinical trials number NCT00001360).

**Participants**. Thirty-two participants performed the Overt Naming Task (18 females) with a mean (SD) age of 24.03 (3.58) years (range: 19–38), and 28 additional participants performed the Covert Naming Task (19 females) with a mean (SD) age of 23.43 (1.62) years (range: 21–28). Participants were right-handed, neurologically healthy native English speakers with normal or corrected-to-normal vision. All participants granted informed consent and were monetarily compensated for their participation.

**Experimental stimuli**. Participants completed either overt or covert picture naming, consisting of 200 colored photographic images of animals, plants, foods, and everyday objects. Images were presented across two lists of 100 pictures each, with the lists matched in conceptual category membership and in average lexical properties of picture names (omnibus $F$ statistics all <1), including lexical decision times on the names (mean response time = 632.3 ms, SD = 71.9 ms) and log HAL frequency determined by the English Lexicon Project database (mean = 8.57, SD = 1.54)[57]. Images were resized to 600 × 600 pixels and presented against a gray background (RGB value: 75, 75, 75). Outside of the scanner, pictures presented on a laptop subtended approximately the central 6° × 5° of visual angle (horizontal × vertical). Inside of the scanner, pictures subtended approximately the central 7.8° × 6.2° of visual angle (horizontal × vertical).

**Naming tasks**. Participants in both Overt and Covert Naming conditions initially named one set of 100 images aloud three times through in a pseudorandom order outside the scanner (in a quiet testing room). In each naming trial, the trial started with a central fixation cross for 500 ms, followed by the picture to be named for 200 ms. The picture offset was followed by a blank screen for 1300 ms, yielding a total trial duration of 2000 ms. Participants were instructed to name each image aloud as quickly and accurately as possible, with correct performance and error responses notated by the experimenter, and response time marked according to voice onset using a microphone on the display computer (Presentation software package Version 11.3, www.neurobs.com).

After a delay of approximately 30 min, participants performed either Overt or Covert Naming inside the MR scanner. In both tasks, a naming trial consisted of a central black fixation cross presented for 1000 ms, followed by the picture to be named for 300 ms, followed by a blank screen for a period ranging from 6700 to 12,700 ms at multiples of the TR (total trial lengths of 8000, 10,000, 12,000, and 14,000 ms) and sampled with a uniform distribution (Fig. 2; see[58] for discussion of optimal ISIs in slow event-related fMRI designs). For Overt Naming, participants spoke into an MR-compatible microphone placed next to the head coil approximately 3–5 cm from the participant's mouth. For Covert Naming, participants were instructed to name the pictures silently to themselves, pressing a response button to mark the beginning of their naming response. In both tasks, the 100 pictures named pre-fMRI (OLD) were randomly intermixed with 100 pictures that were novel for the fMRI session (NEW), with trials organized into 5 runs of 40 pictures each. For Covert Naming, participants completed an additional Overt Naming session of all 200 pictures immediately after fMRI, in order to confirm that their button-press response times during fMRI agreed with actual overt response times (same timing and methods used as for the pre-fMRI session). Experimental lists (OLD versus NEW) were counterbalanced across participants, and only correct trials were included in priming estimates and task analyses.

**Recording naming responses during MRI**. Spoken responses were captured with an Opto-Acoustics FOMRI-III NC MR-compatible microphone with built-in noise cancellation and routed into an M-Audio FastTrack Ultra 8-R USB audio interface. Responses were recorded with Adobe Audition. To calculate response times, the stimulus presentation computer emitted a square wave pulse at the onset of each trial and a custom Matlab program calculated the time difference between the square pulse onset and voice response onset for each trial.

**MRI methods**. Images were acquired with a General Electric Signa HDxt 3.0T scanner (GE Healthcare) using an 8-channel receive-only head coil. A high-resolution T1-weighted anatomical image (MPRAGE, magnetization-prepared rapid gradient-echo) was obtained for each participant (124 axial slices, 1.2 mm slice thickness, field of view = 24 cm, 224 × 224 acquisition matrix). Functional (T2*-weighted) images were acquired using a gradient-echo echo-planar imaging (EPI) sequence [Array Spatial Sensitivity Encoding Technique, ASSET, acceleration factor = 2, TR = 2000 ms, TE = 27 ms, flip angle = 60º, 40 sagittal slices (3.5 mm slice thickness), field of view = 216 mm, 72 × 72 acquisition matrix, voxel resolution = 3.5 × 3.0 × 3.0 mm³]. Each experimental task run lasted 7 min 40 s for a total of 230 consecutive whole-brain volumes, with each participant receiving a total of 5 runs. Foam earplugs were worn by participants to attenuate scanner noise and participants' head positions were stabilized using foam pillows. All EPI data were evaluated for transient head motion artifacts, with included scans required to be ≤0.3 mm/TR using AFNI's @1dDiffMag function (comparable to mean Framewise Displacement[59]). Independent measures of cardiac and respiration cycles were recorded during the task scans for later removal.

**fMRI data preprocessing**. Preprocessing utilized the AFNI software package[60], applying steps in the following order: (1) removal of the first 3 TRs to allow for T1 equilibration; (2) 3dDespike to bound outlying timepoints per voxel within 4 standard deviations of the time series mean; (3) 3dTshift to adjust for slice acquisition time within each volume (to $t = 0$); (4) 3dvolreg to align each volume of a run's scan series to the first retained volume of the first run; (5) each scan was then spatially blurred by a 6-mm Gaussian kernel (full-width at half-maximum) and divided by the voxelwise time series mean to yield units of percentage signal change. De-noising of each scan then utilized the ANATICOR nuisance regression approach[34,61]. White matter and large ventricle masks were created from the aligned MPRAGE scan using Freesurfer[62], and a large draining vein mask was created from a standard deviation map of the volume-registered EPI data (from step 4 above). All masks were resampled to EPI resolution and eroded by 1 voxel to prevent partial volume effects with gray matter voxels, and the related nuisance time series were calculated on the volume-registered data just prior to spatial blurring (after step 4 and prior to step 5 above). Nuisance regression for each voxel was performed on the spatially blurred volume-registered data (after step 5 above), and the regressors consisted of: 6 head-position parameter time series (3 translation, 3 rotation), 1 average eroded ventricle time series, 1 "localized" eroded white matter time series (averaging the time series of all white matter voxels within a 20-mm-radius sphere), 1 eroded draining vein time series, 8 Retroicor time series (4 cardiac, 4 respiration) calculated from the cardiac and respiratory measures taken during the scan[63], 5 respiration volume per time (RVT) time series to minimize end-tidal $CO_2$ effects following deep breaths[64], and the first 3 principal component time series calculated on a union mask of the nuisance tissues (white matter, ventricles, draining veins) (aCompCor regressors[65,66]). Prior to regression, all

nuisance time series were detrended by a 4th-order polynomial function to remove slower scanner drift and drift in head position, with the de-noised residuals detrended in the same manner during regression. After regression, de-noised residual time series were transformed to standardized anatomical space (Talairach-Tournoux) for task analyses at a resolution of 3 mm³ isotropic.

**Statistics and reproducibility**. Detailed information on statistical tests performed are provided in the specific subsections below. Tests of behavioral measures included paired $t$-tests for all within-participant measures and correlation tests (Pearson's $r$) for across-participant associations between brain and behavioral measures (or solely between behavioral measures). Normality of data for these tests was assessed using Lilliefors' composite goodness-of-fit test (in Matlab). Test–retest reliability of behavioral measures was assessed with Pearson's $r$. Group-level tests of fMRI beta parameters utilized one-sample $t$-tests (versus 0) for stimulus responses and paired $t$-tests for repetition effects. Analyses of functional and effective connectivity, representational similarity, and peak BOLD response timing utilized LME models with covariates. Replications were assessed for stimulus and repetition effects (betas) across Overt and Covert Naming experiments ($N = 32$ and $N = 28$, respectively), as well as for Weak/Strong Primeability effects in behavior.

**Normative analyses of naming response times and Primeability**. A within-participant estimation of "Primeability" (Strong versus Weak) in object naming was determined from normative analyses of response times across all 60 participants. Response times on correct naming trials were averaged across participants when encountered for the first time as NEW objects (i.e., the first presentation in the pre-fMRI session or the first presentation during the fMRI session), as well as for OLD objects (i.e., presentation during fMRI of objects seen during the pre-fMRI session). Only response times in Overt Naming sessions were used for this purpose, since button-press response times during Covert Naming were systematically faster than for Overt Naming sessions by as much as 200–300 ms per participant. Each item's resulting average NEW response time included the correct naming responses from approximately 40 participants. Test–retest reliability of mean NEW response times per object was assessed across Overt and Covert Naming experiments using Pearson correlation, as well as the test–retest reliability of priming magnitude per object, estimated as the difference in response time when an object was OLD versus NEW. Reliability of NEW responses and priming magnitudes across Overt and Covert Naming experiments was high for both measures, but was highest for NEW responses [$r$ (198) = 0.704 vs $r$ (198) = 0.436]. Given the high reliability of mean NEW response times across experiments, as well as the strong relationship between mean NEW responses and priming magnitude by item when combining experiments (larger priming for slower NEW responses; Fig. 4b), "Primeability" was calculated as a median split of mean NEW response times, with Strong Primeable objects defined as the 100 objects with the slowest mean NEW response times across participants and Weak Primeable objects defined as the 100 objects with the fastest mean NEW response times across participants. We fully expect item Primeability to be a complex function of a variety of factors including frequency of exposure, the level of visual detail and color present in the picture, as well as its "neighborhood" of visual and conceptual relationships with other objects[67]. The use of response time to NEW pictures here is a practical convenience.

Although the contribution of any particular participant's response times during fMRI was expected to be minimal to the overall Strong/Weak Primeability norms, we also conducted a set of follow-up analyses in which each participant's behavioral data was excluded from the calculation of the norms. This allowed selection of Strong/Weak Primeability to be completely independent of the analyses of the fMRI responses, with a unique selection of the Strong/Weak median split for each participant. As expected, there was no marked change in the pattern of results (see Supplementary Figs. 5–7).

**Behavioral priming analyses**. Response times to correct naming trials during fMRI were tabulated for each participant, separately for each experimental condition [Repetition (OLD, NEW) crossed with Primeability (Strong, Weak) for a total of four conditions]. Trials with responses slower than 2000 ms were excluded from analyses (modeled as incorrect trials), as these trials would likely contain speech artifacts affecting multiple TRs. Repetition priming magnitudes per participant were then calculated for Strong and Weak Primeable conditions using effect size (Cohen's $d$), the difference in means (NEW–OLD) divided by the pooled standard deviation. For purposes of correlating priming magnitudes with measures of neural activity derived from fMRI, effect sizes determined for Covert Naming participants were averaged from the fMRI and post-fMRI sessions (during which the same items were named overtly).

**Choice of statistical thresholds for fMRI analyses**. The overall analysis strategy was first to identify brain regions exhibiting repetition suppression using typical general linear model (GLM) contrasts of OLD versus NEW. The voxelwise alpha levels on these contrasts were chosen such that FDR[68] indicated that fewer than 1 voxel in each task (Overt Naming, Covert Naming) could be due to chance, with the interpretation that surviving voxels replicate across tasks (in this case, a voxelwise threshold of $P < 0.00001$, with FDR correction to $q < 0.00006$). This high thresholding also broke apart large clusters and afforded detection of isolated ROIs.

We also required that any such voxels exhibited above-baseline levels of activity at a minimum level of significance ($P < 0.05$, $q < 0.05$). Whole-brain searches for repetition-related changes in functional connectivity were conducted on the same data sample used for the identification of repetition suppression ROIs. For whole-brain searches of functional connectivity using cluster-size correction, we chose a voxelwise alpha of $P < 0.001$, correcting for multiple comparisons to a familywise Type-I error rate of $P < 0.05$, as this has been shown to control for false positive rates at 5% or less when using an empirical autocorrelation function estimation for cluster distribution[69,70]. All subsequent analyses utilized FDR to correct for multiple comparisons, with the FDR controlled at $q < 0.05$.

**fMRI task analyses**. Traditional task analyses were conducted at the voxel level using a GLM, in which the data at each timepoint are treated as the sum of all effects thought to be present at that timepoint, plus an error term. Responses associated with each condition were modeled using TENT basis functions in AFNI, with a separate regressor for each timepoint following the stimulus onset, permitting empirical estimation of the hemodynamic response function (HRF) shape. This approach assumes that all responses for a given condition share the same response shape but makes no assumption as to what the shape of that response might be. Responses for each participant were modeled over 8 timepoints from $t = 0$ s to $t = 14$ s in increments of the TR (2 s). One additional regressor of no-interest was coded for error trials in naming (either omissions or commissions). For the purposes of statistical testing, peak response magnitudes were estimated by averaging the 3rd and 4th timepoints of the TENT function corresponding to the peak of the typical BOLD response, reflecting activity 4–8 s post-stimulus onset; visual examination of the beta coefficients confirmed that this period did indeed represent the typical peak well (see Fig. 9). Single-participant contrasts for overall Stimulus Response (pooling conditions) and effect of Repetition (either repetition suppression or enhancement, pooling across Strong/Weak Primeability conditions) were tested using linear tests within the GLM. Beta coefficients from the regression for each participant were submitted to group-level analyses of the overall Stimulus Response and effects of Repetition.

A group-level effect of Stimulus Response relative to baseline was evaluated in each voxel separately for Overt and Covert Naming using a one-sample $t$-test across participants of the related beta coefficients (β) versus 0 (averaging the 3rd and 4th TR β's at the peak response), with positive β's indicating above-baseline and negative β's indicating below-baseline responses. Two statistical thresholds were employed: (1) a minimum level of significance at $P < 0.05$ when pooling participants across the two tasks, corrected for whole-brain comparisons using FDR to $q < 0.05$, and (2) a stringent level of significance in each task individually ($P < 0.0001$, $q < 0.00016$); given the number of voxels meeting the threshold in the brain volume (6322 voxels at $q = 0.00016$ in Covert Naming, 9287 voxel at $q = 0.00009$ in Overt Naming), this level of FDR corresponds roughly to the expectation of only 1 false positive voxel in each task. At the more stringent threshold (with the extremely low FDR), responses can be said to replicate across tasks.

A similar approach was taken in evaluating Repetition effects (OLD versus NEW objects) at the group level, collapsing across Strong/Weak Primeability conditions. Two statistical thresholds were employed using a paired $t$-test (OLD–NEW) across participants: (1) a minimum level of significance at $P < 0.05$, $q < 0.05$) when pooling participants across the two tasks, and (2) a stringent level of significance in each task individually ($P < 0.00001$, $q < 0.00006$), with an expectation of <1 false positive voxel in each task. This was combined with the minimum level of significance ($P < 0.05$, $q < 0.05$) for the Stimulus Response when identifying ROIs.

**fMRI task-based functional connectivity analyses**. In order to minimize contamination of task-based functional connectivity estimates from the temporal contour of the evoked responses of individual trials, only the average peak BOLD response was retained from each individual trial (the raw average of the 3rd and 4th timepoints of the task residual time series following the onset of each stimulus). For these analyses, data had undergone preprocessing with nuisance regression (described above), as well as having removed the condition-level mean responses during GLM regression analyses. This resulted in an "item series" of peak BOLD responses from correct trials in each individual voxel, with a maximum length of 100 OLD and 100 NEW trials and each divided approximately in half for Strong and Weak Primeability conditions (e.g., a maximum series of approximately 50 Strong Primeable, OLD trials). Functional connectivity analyses utilized a $2 \times 2 \times 2$ mixed effects design with Task (Overt, Covert) as a between-participant variable, Repetition (OLD, NEW) and Primeability (Strong, Weak) as within-participant variables and participant as a random variable. Employing a more continuous analysis of functional connectivity with priming strength was not possible within-participant, as only a single timepoint (the peak) was retained for each item; functional connectivity estimates therefore had to be calculated across items, with priming strength incorporated within-participant as a median split of items between Strong and Weak Primeability. For all analyses, two sets of tests were of primary interest: (1) a main effect of Repetition (OLD versus NEW) collapsing across Primeability, and (2) an interaction of Repetition × Primeability, given that this is the pattern observed in behavioral priming with larger differences between NEW and OLD response times for Strong Primeable compared to Weak Primeable

conditions (Fig. 4). Accordingly, the familywise alpha for multiple-comparisons correction was set at $P < 0.025$ (0.05/2) in order to correct for two sets of voxelwise (or ROI-level) tests, with the full FWE Type-I error rate controlled at $P < 0.05$. Our functional connectivity analyses employed the 3 steps used previously by Gotts and colleagues for whole-brain functional connectivity analyses: seed definition, target ROI selection, and region-to-region correlation analysis[34,71,72]. We undertook all 3 steps for both the Repetition main effect and the Repetition × Primeability interaction effect. The further interaction of these primary effects with Task was also evaluated to examine the possibility that any results could simply be due to the presence of speech artifacts (e.g., present in Overt but not Covert Naming).

Seeds were identified using whole-brain "connectedness"[34,73,74]. The average Pearson correlation of each voxel's "item series" (the array of peak BOLD responses to individual trials of the same type) with the item series in all voxels responding significantly above baseline in the task (identified for each individual participant, with $P < 0.0001$ needed to control FDR to $q < 0.05$) was calculated to create a 3D reduction of the 4D (3D + Time) dataset for each condition. In other words, the correlation of a particular voxel's responses was calculated with those of all task-responsive voxels, storing the average of those correlations back into the voxel. Since connectedness in this case reflects the average level of correlation with task-responsive voxels, it gives an indication of how intercorrelated a given voxel is with those voxels most engaged by the task. This approach, akin to centrality in graph theory, has been used previously in studies of both resting-state[34,66,71,73–77] and task-based functional connectivity[72,78,79]. LME models (using AFNI's 3dLME[80]) were constructed whose dependent variables were the voxel-wise connectedness maps in each experimental condition. Task, Repetition, Primeability, and their interactions were included as fixed effects. The global level of correlation among all brain voxels, GCOR[35,81], was included as a nuisance covariate in order to model any residual motion and/or breathing artifacts present after the nuisance regression (see[82] for discussion). Participant was treated as a random intercept. Cluster-size correction was used to control the Type I error rate. The average smoothness of the de-noised functional time series was estimated with AFNI's 3dFWHMx, using the empirical, spatial autocorrelation function (June 2016). Then, 3dClustSim (June, 2016) was used to run a Monte Carlo simulation with 10,000 iterations in a whole-brain mask in Talairach space within which the analyses were performed. Importantly, the smoothness estimates and noise simulations did not assume Gaussian distributions of activity, which has been shown to inflate the false positive rate in studies using more traditional cluster-size correction[69,70]. Clusters were selected at a cluster defining threshold of $P < 0.001$, familywise alpha of $P < 0.025$ (0.05/2, for two sets of tests), minimum cluster size $k = 25$ voxels.

The seed definition step was then followed with more typical seed-based correlation analyses. The item series within each seed region was averaged across voxels to form ROI-averaged item series, which were correlated with the item series for every voxel in the brain, separately for each experimental condition. These correlations were Fisher $z$-transformed and used as dependent variables in LME models with the same fixed and random effects as for the seed detection step. We tested for the main effect of Repetition and the Repetition × Primeability interaction at a voxel threshold of $P < 0.001$, with correction by cluster size for whole-brain comparisons as well as the number of seeds tested (i.e., FWE correction to $P < [0.025 / (\text{number of seeds})]$. Results were then further masked by voxels exhibiting above-baseline responses in both tasks individually ($P < 0.05$, $q < 0.05$), leaving clusters of voxels that both showed changes in functional connectivity with stimulus repetition and that were engaged in the task. Secondary target regions were then combined together with seed regions, as well as any regions exhibiting repetition suppression in both tasks, to arrive at a full set of ROIs.

Regions were sampled as 6-mm-radius spheres centered on the peak statistic used to identify each ROI ($F$-statistic from the LME on connectedness or seed-based correlation tests for ROIs showing changes in functional connectivity; the $t$-value of the OLD versus NEW comparison of beta coefficients for repetition suppression regions). Region-by-region matrix analyses were then conducted using the same LME approach applied to connectedness and the seed-based tests, allowing the examination of all inter-regional relationships. Multiple comparisons in the region-by-region analyses were controlled with FDR ($q < 0.05$).

**fMRI task-based effective connectivity analyses.** Single-trial responses (same data as for functional connectivity analyses) from the 7 regions identified in the whole-brain functional connectivity analyses (including repetition suppression regions) were submitted to effective connectivity analyses[54,83,84]. Of the various effective connectivity approaches (e.g., dynamic causal modeling - DCM, Granger causality, multivariate autoregressive modeling, etc.), structural equation modeling, or SEM[37,85,86], provided the best match for the current data characteristics, since it only requires covariance or correlation matrices as inputs. In particular, any approaches requiring the use of time series (e.g., Granger-based methods) were not appropriate, since the data consisted of a series of peak BOLD responses that had been notched out of the original time series in an event-related design with interleaved trial types. As with other exploratory effective connectivity approaches, we first performed a search for the model that best accounted for the pattern of covariance among the ROIs when pooling data across all conditions and participants (using AFNI's 1dSEM[37]). A search was performed with both tree growth and forest growth algorithms using the Akaike Information Criterion (AIC)[87], a

measure of out-of-sample prediction error, to choose among different SEM models and to guard against overfitting. The optimal model had 11 directional connections among the ROIs.

Following the model search step, the optimal model was parameterized for each participant's condition-specific datasets individually (AFNI's 1dSEMr[37]). The parameterization failed to converge for two participants' datasets (the NEW, Weak Primeable condition for one participant, the OLD, Strong Primeable condition for another; both Covert Naming participants), and these sets were excluded from further analyses. The results of the successful parameterizations were then submitted to an LME analysis that paralleled the one performed on the functional connectivity data, with the SEM parameters as the dependent variable and Task, Repetition, Primeability, and their interactions included as fixed effects. GCOR was included as a nuisance covariate, and Participant was treated as a random intercept. One out of the 11 SEM parameters (from the R TP to the Left Frontal ROIs) failed to differ from zero across participants overall or in any individual condition and was excluded from further analyses. The remaining 10 parameters were tested for overall effects of Repetition and Repetition × Primeability interactions, as with the functional connectivity analyses, with multiple comparisons corrected by FDR ($q < 0.05$). Further interactions with Task (Overt, Covert) were also evaluated. For connections exhibiting a significant Repetition × Primeability interaction, follow-up contrasts were conducted to clarify the nature of the interaction, comparing parameters to OLD and NEW objects separately for Strong versus Weak Primeable conditions, with multiple comparisons corrected by FDR ($q < 0.05$).

**Evaluating predictions of the Sharpening model with MVPA.** The Sharpening model predicts that OLD objects should have spatial patterns across repetition suppression ROIs that are less similar to one another than for NEW objects, consistent with the loss of neural overlap among the neural representations. To evaluate this, we applied MVPA[38,88–90] to the single-trial BOLD responses in each experimental condition (OLD, NEW × Strong, Weak Primeable). For these analyses, the data used were post-nuisance regression in preprocessing but prior to the removal of condition means in GLM analyses. Each single-trial response was calculated as the average of the 3rd and 4th TRs post-stimulus onset (the 4–6 s and 6–8 s TRs), i.e., the peak response, minus the average of the 1st TR in the trial (0–2 s) and the TR previous to it (−2 to 0 s), i.e., the baseline response. This resulted in a voxelwise pattern of BOLD responses within an ROI. For a given participant and ROI, the spatial correlation (Pearson) of all trials with all trials within the same condition (e.g., NEW, Strong Primeable) was calculated across voxels, storing the median correlation value to protect against skewing effects. The median correlations then served as the dependent variable in an LME analysis, including Task, Repetition, Primeability, and their interactions as fixed effects and Participant treated as a random intercept. Multiple comparisons were corrected by FDR ($q < 0.05$). Further interactions with Task (Overt, Covert) were also evaluated. For ROIs exhibiting a significant Repetition × Primeability interaction, follow-up contrasts were conducted to clarify the nature of the interaction, comparing median correlations to OLD and NEW objects separately for Strong versus Weak Primeable conditions, with multiple comparisons corrected by FDR ($q < 0.05$).

The use of individual trial-level and voxel-level BOLD responses could potentially suffer from poor SNRs when scanning at 3 Tesla, with "attenuation" of spatial correlation values due to noise[91–93]. This could be particularly problematic when the ROIs of interest have known differences in overall activity levels in different conditions, in this case due to repetition suppression—which is how the ROIs were defined. If OLD objects have BOLD responses that are closer to the baseline level than NEW objects, the spatial correlation values could be closer to 0 simply due to lower beta coefficients across the ROIs and poorer SNRs. In order to examine this issue, two nuisance covariates were included in a subsequent LME analysis of each ROI: (1) the average beta coefficient across the ROI, and (2) the estimated SNR of trials in each condition for each participant. The SNR, defined as the variance of the signal divided by the variance of the noise, was estimated for a given condition in the following manner: (1) for each trial, the difference between the peak and baseline TRs represented the single-trial BOLD response; the variance of this pattern across voxels served as the numerator in the SNR calculation, (2) a noise estimate for each trial was calculated by randomly selecting two baseline patterns from the same ROI and condition, subtracting one from the other, and then calculating the variance of this pattern across voxels, (3) dividing the variance in step 1 by the variance in step 2 yielded the SNR estimate for that trial, (4) the median SNR value across trials in each condition served as the second nuisance covariate in the new LME analysis. It is important to mention that the estimate of the peak responses in each trial is not noiseless in this case, so this estimate of SNR is expected to be an upper bound on the true SNR. A strong fit of the two nuisance covariates to the ROI data, combined with a change in significance levels from the LME analysis without the covariates, would indicate results consistent with correlation attenuation. These relationships were further evaluated by examining scatterplots of the nuisance variables on the $x$-axis and median spatial correlation level on the $y$-axis, overlapping results across the different experimental conditions (Supplementary Fig. 4).

**Evaluating timing predictions of the Facilitation model.** A central prediction of the Facilitation model is that neural responses should be temporally advanced for OLD compared to NEW objects in regions showing repetition suppression, with an

earlier peak response[22,23]. In order to examine this prediction for the evoked responses in fMRI, we fit a simple gamma variate function to the TR-specific beta regression coefficients calculated during the GLM analyses in the 4 repetition suppression ROIs, with a separate response function for each of the four experimental conditions [Repetition (OLD, NEW) × Primeability (Strong, Weak)]:

$$h(t) = h_0 + a \cdot t^b \cdot e^{-\frac{t}{c}} \qquad (1)$$

$h(t)$ represented the hemodynamic response as a function of time $t$, with 4 modifiable parameters: $h_0$, representing a baseline value of $h$; $a$, a scaling parameter on the height of the curve; $b$, an exponent on $t$ determining the rise time of the curve; and $c$, a time constant for the exponential decay of the curve. The time of the peak response ($t_{peak}$) then corresponded to the value of $t$ at which the derivative of $h(t)$ with respect to $t$ equaled zero:

$$t_{peak} = b \cdot c \qquad (2)$$

For each curve, the best-fitting parameter values for all 4 free parameters were determined initially by a course grid search, followed by fine-tuning through a gradient descent error minimization until stability had been reached (≤300 iterations). The estimates of $t_{peak}$ were then submitted to LME analyses as the dependent variables, including Task, Repetition, Primeability, and their interactions as fixed effects and Participant treated as a random intercept. Multiple comparisons were corrected by FDR ($q < 0.05$).

**Reporting summary**. Further information on research design is available in the Nature Research Reporting Summary linked to this article.

## Data availability

Raw MRI data are available via the XNAT platform. Users will need to request access through the XNAT system (https://central.xnat.org/app/template/Index.vm). This can be done by creating an XNAT user account and pressing the "request access" link. Processed, standard-space (Talairach) fMRI data (NIfTI format) and behavioral (response time) data are available from the authors (S.J.G.) on request. All data used to create the figures in the main text is available at Figshare (https://figshare.com/projects/Gotts_Milleville_Martin_2021_Communications_Biology_/99527).

## Code availability

fMRI preprocessing code used in the paper is available online at figshare.com: https://doi.org/10.6084/m9.figshare.14199896 and analysis code used in the paper is available online at figshare.com: https://doi.org/10.6084/m9.figshare.14199908 (https://figshare.com/projects/Gotts_Milleville_Martin_2021_Communications_Biology_/99527).

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

## Acknowledgements

We thank Ally Ossowski for assisting with data collection and analysis, Gang Chen and Daniel Glen for assisting with the structural equation modeling analyses, Vinai Roopchansingh for aid in designing the sagittal fMRI scanning sequences, and Adrian Gilmore, Avniel Ghuman, Michal Ramot, Andrew Persichetti, Jason Avery, Al Braun, Bob Cox, Peter Bandettini, Fernando Ramirez, Eli Merriam, and Chris Baker for helpful discussions. We would also like to thank Rik Henson and two anonymous reviewers for their numerous helpful and constructive comments on the manuscript. This study was supported by the National Institute of Mental Health, NIH, Division of Intramural Research (ZIAMH002920; ClinicalTrials.gov ID NCT00001360). The funders had no role in the study design, data collection and analysis, decision to publish, or preparation of the manuscript.

## Author contributions

S.J.G. and A.M. conceived and designed the experiments. S.J.G. and S.C.M. performed the experiments. S.J.G. and S.C.M. analyzed the data. S.J.G. and A.M. wrote the paper.

## Funding

## Competing interests

The authors declare no competing interests.
