## [Peer Review File · Communications Biology]

Reviewer #1:

Remarks to the Author:

This paper describes fMRI data on visual object naming from a relatively large sample of participants (collapsed across covert and overt naming tasks), various analyses of which are used to test the main theories about the neural basis of repetition suppression (RS), a brain phenomenon that might relate to the behavioural phenomenon of priming. The data are high quality and the analyses interesting, even if the results are not particularly conclusive about any theory. However, I have a few questions below.

1. Is there a problem of "regression to the mean" in the calculation of primeability? If one defines priming as the difference between NEW and OLD RTs, and the *same* NEW RTs are used to define primeability, then this is a serious problem, because the positive relationship in Figure 4B would simply be an artefact of regression to the mean. This is because if one draws RTs for NEW and OLD items from identical but independent distributions (i.e, with the same mean, i.e, with no true priming), one will still get a positive slope between NEW-OLD and NEW (in Matlab: "y = randn(100,2); figure, plot(y(:,1),y(:,1)-y(:,2),'o');"). However, if the trials are used to calculate NEW RTs for the x-axis (e.g, from pre-fMRI session) are independent from the trials used to define priming (e.g., NEW trials in the fMRI session), this is no longer a problem (in Matlab: "y = randn(100,3); figure, plot(y(:,1),y(:,2)-y(:,3),'o');"). I couldn't quite work out what the authors did, e.g, if different objects were used for different NEW conditions in different participants, but if there is any possibility of regression to the mean, then this should be scrapped.

2. Also, if primeability was a valid measure, I wasn't sure why a median split was used, rather than the (potentially more sensitive) use of the continuous measure of primeability for all the analyses in the paper?

3. I'm not sure the Predictive Coding model predicts a more negative coupling between regions after repetition. This model predicts that the top-down predictions will be more accurate after repetition, such that prediction error (assumed to relate to fMRI signal) is smaller. But I don't think this necessarily means that the trial-by-trial correlation between peaks of the fMRI responses in the two connected regions is negative, because it is not clear that more precise predictions (following synaptic change in the connections between regions) from the "higher" ROI necessarily means higher (or lower) peak responses in that ROI. In fact, I think you could argue the opposite, in the sense that when predictions are improved by priming, there is a stronger coupling between regions. Indeed, in two papers by Ewbank et al (2010, 2012) using Dynamic Causal Modelling (DCM), they found that repetition made top-down connections more positive (note that in DCM, positive connections mean that the rate-of-change of activity in the receiving ROI is positively related to the activity in the sending ROI). I think the authors need to spell out their precise reasoning here, which possibly includes some additional assumptions they are making about the Predictive Coding model, with which other readers might not agree.

4. Continuing Point 3, there are convincing arguments why DCM is superior to the SEM used by the authors for the purposes of determining effective connectivity in fMRI (arguments including the use of differential equations to properly infer directionality, the use of ROI-specific haemodynamic parameters, and the use of a Bayesian framework in order to more flexibly compare models, e.g, those with closed loops or non-nested). I appreciate that DCM is complex and difficult to use, and some people do not trust the results (e.g, the assumptions behind its use of Bayesian model evidence), but it would at least be interesting to know whether it provides converging results to those obtained by the authors' SEM (which is a static method based only on the covariance between ROI fMRI timeseries, rather than underlying neural timeseries).

5. Continuing Point 4, the results of the functional and effective (SEM) connectivity are somewhat unexpected, since they do not involve connections between the main, a priori ROIs (i.e., those showing RS, e.g, fusiform and inferior frontal). While it is theoretically interesting that ROIs can show repetition-related changes in connectivity without repetition-related changes in activity, the

role of the ROIs that show connectivity changes in priming is unclear (they are not typically associated with perceptual or semantic or verbal processing for example), and if the main purpose of the paper is to compare theoretical models of RS, then what is the relevance of ROIs that don't show RS? Moreover, since the actual goal of DCM is to explain condition-related changes in activity through changes in connectivity, a DCM network based on those ROIs that show RS (eg between inferior frontal and fusiform ROIs) would seem potentially much more informative about the authors' main question.

6. A minor quibble with the predictions stated for the Facilitation model in the Introduction: "responses in regions exhibiting RS should be temporally advanced for repeated objects, with earlier onset and peak". The Facilitation model assumes that repetition speeds up the (competitive) neural dynamics that resolve perception of an object (such as the dynamic reduction of prediction error). The simplest prediction is that the centre of mass of the firing rate (over a few hundred milliseconds after stimulus onset) occurs earlier for repetitions. However, this need not reflect a change in the onset of neural firing, or even the peak; it could reflect solely a reduction in the duration of neural firing. Moreover, when the authors talk about onset and peak, the authors should clarify whether they are talking about the neural or fMRI signal. After convolution with a hemodynamic response, any facilitation effects on the onset of the BOLD response would be difficult to detect – they are more likely to be seen in the latency (and magnitude) of the peak of the BOLD response (which is actually what the authors measured, so mention of the onset is confusing here).

7. In terms of access to the data and code, I don't think it's enough to simply say you need an XNAT server to get the data (without further information on data location) and to email authors for the code. Many papers provide these openly (e.g, for reviewers to assess), eg on OSF or other website.

Signed
Rik Henson

Reviewer #2:

Remarks to the Author:

This research by Gotts, Milleville & Martin uses fMRI to examine the neural basis of repetition priming. Examining four existing models of RP, the main focus of the paper was on changes in neural synchrony associated with stimulus repetition during a picture naming task.

While the experiment and results are noteworthy and have the potential to deepen our understanding of repetition priming, there are 2 major issues that would need to be addressed before moving forward. First, it is very unclear how well the study is able to adjudicate between the 4 theoretical perspectives. The introduction mentions in a single sentence that each model "makes unique predictions", but then goes on to focus the paper on something common between them, synchrony – "Common to most of these models is the prediction that measures of connectivity between brain regions should be affected by repetition." This leaves the reader with the sense that the work will not be able to falsify any of the existing models. However, the analysis then proceeds to test various aspects of select models posthoc – peak timing (predictive coding model) & similarity of voxel activation patterns between new and repeated items (sharpening model). Leaving aside whether fMRI is an appropriate methodology for testing differences in neural timing (addressed in the discussion) – this approach leaves the reader with a very muddled picture about what aspects of the 4 models were tested and what the results tell us.

The second major issue is the choice to divide objects into high and low "Primeability", based on the naming time for the items when first presented. Leaving aside the issue of whether it is ever a good idea to mean/median split a variable that shows continuous characteristics – (see -

MacCallum, R. C., Zhang, S., Preacher, K. J., & Rucker, D. D. (2002). On the practice of dichotomization of quantitative variables. *Psychological Methods*, 7(1), 19–40.), there is no reference for taking this approach, no work where it was previously done and therefore has a posthoc feel to it - we didn't find what we wanted when we included all items, so we divided them into 2 categories. If this is in fact the case, then this is an exploratory analysis and should not be the key approach to analyzing the question of interest. If this is not the case then a more convincing justification is needed that is based on the vast priming literature that doesn't seem to ever take this approach. In general, the priming literature has focused on the reliability of repetition priming rather than its magnitude. Slower initial items may show larger priming effects just due to mean regression around a primed mean.

In addition to these 2 major issues, there are some lesser items that could be addressed –

1 – alpha levels throughout seem arbitrary. These should be set up front and consistent across all major analysis.

2 – the approach to connectivity is not something common to the literature – “The response peak was then notched out of the time series, with connectivity calculated as the co-fluctuation (correlation) of peak responses across trials between pairs of regions/voxels” – could a reference be provided?

3 – The paper says - “failed to exhibit increased feed-forward coupling during the processing of OLD relative to NEW objects” – this is interpreting a null finding about a level of connectivity detail that fMRI is likely not suited for - is fMRI with a TR of 2 sec appropriate to determine “peak timing” differences of less than 500msec? Since the peak is determined by fitting to a standard model, there are many reasons the peak might change – number of trials fit, differences in the variability of response – something seen in the ERP literature. The authors themselves question the use of fMRI to examine subtle timing differences as well as extent of activation, yet these 2 approaches form critical tests of the competing theories.

4 – The paper says – “With respect to the Facilitation model, neural responses were indeed found to peak at earlier times for OLD compared to NEW objects in regions showing repetition suppression, although these effects failed to interact with primeability”. This seems an odd statement, given that the facilitation model has never made any predictions with respect to slower vs faster identified/named items.

5 – Finally, the changes in synchrony are examined in a circumscribed set of regions that showed univariate priming effects. Why not take a larger connectome approach to changes in connectivity? The current approach would hide whether priming results in increased/decreased connectivity in other regions that don't show repetition suppression effects. One might speculate that a sharpening model might reveal decreased connectivity in regions NOT critical to primed responses.

Reviewer #3:

Remarks to the Author:

This is the first review of 'Enhanced inter-regional coupling of neural responses and repetition suppression provide separate contributions to long-term behavioral priming' by Gotts, Milleville and Martin. The work looks at behavioral priming in naming times for visual stimuli using a slow event related fMRI design and several sophisticated analyses. The paper is quite dense given the number of complex analysis methods employed and the brief format and I had some difficulty keeping track of the methods and how they related to the abstract priming models discussed. As I note below, the impact of the work potentially suffers from the fact that the chosen method cannot possibly critically evaluate the relative merit of the four abstract priming mechanisms that are introduced at the beginning. The authors acknowledge this in the Discussion but until that point,

the paper is framed as though the data are capable of ruling out or clearly ranking the model competitors. Additionally, this undercuts the key aspect that is highlighted in the title, namely that the work may demonstrate that unique portions behavioral variance in priming may be accounted for by repetition suppression versus connectivity. Convincing a skeptical audience of this, and perhaps linking it different metrics of the materials would be quite exciting, but the latter isn't done at present. Below I elaborate on the main concerns that arose during the review, followed by a brief section outline more minor concerns.

MAJOR

What are the relative disadvantages of the fMRI method for the competing models? What I mean by this is that, depending upon the predictions of the four coarse models, some may fare better than others because of the unique characteristics of the current methods. For example, those models which predict temporal changes in activity are presumably hamstrung by the temporal filtering properties of the BOLD response and the chosen TR of 2000 ms. In contrast, the method may be more reliable for capturing interregional connectivity particularly since (I think) the authors are using residuals that are likely to depend upon lower frequency information. Ideally one would want to have similar power to detect the key predictions of each of the competitors. I discovered that the authors acknowledge this limitation in the Discussion section, but it undercuts the framing of the entire paper as a test among competing explanations of priming, which are themselves not mutually exclusive. Thus, the reader is left with two findings that appear to predict priming behavior, at least for a subset of the materials. A left PFC change in the univariate response that correlates with behavioral priming and replicates prior work, and a pattern of functional connectivity between left TPJ and ACC that also correlates with behavioral priming. This is interesting, but clearly insufficient as a basis for making claims about the validity of the abstract models of priming that the authors lay out in the Introduction. From my perspective, it'd be more interesting to know if in fact a strong case can be made that the RS in the left Frontal region and the connectivity between TPJ and ACC really do account for unique variance in the behavioral priming, and if so, is there something about the objects that might explain this distinction?

If I understood the outcome of the connectivity findings, it appears that the TPJ-ACC finding shows a positive relationship for strongly primeable items, but also shows a reliably negative relationship for weakly primeable items. I'm having a hard time understanding how the negative relationship fits into the same explanatory framework as the positive. For example, the authors speculate that the positive relationship may play a role in an expanded network that 'functions to retrieve prior stimulus-response associations encoded through cortico-striatal loops.' (page18). How would such a functional contribution demonstrate a negative association for half of the items?

It is not clear whether the median split primeability scores (based on solely on novel reaction times) was planned or exploratory.

Although they could not know of it at the time of the current submission, a paper by Davis et al. from the Cabeza lab (Davis et al. in press *Cerebral Cortex*) may be relevant as it examines visual and semantic similarity of objects and their potential impact on subsequent memory performance. I mention it because it seems like object similarity might also be an important consideration in naming times in the current report. That is, there may be a possibility that naming is slowed as a function of the physical similarity (or conceptual similarity) of say a current object, and objects previously seen with a different name. Their approach was based on scores derived from deep neural networks, discussed reviewed Kriegeskorte and Kievit (2013), however, there may be even simpler methods of scoring similarity than this (e.g., Zelinsky 2003 *Psychonomic Bulletin and Review*). Regardless, the authors seem to assume that the only thing that is affecting naming times is the number of encounters with the specific exemplar (and its primeability) whereas the relationship of the item with other members of the set may also be important.

MINOR.

In the methods it is stated that the ISI during scanning ranged from 6700 to 12700 ms at

multiples of the TR. However, the TR of the echoplanar sequence is later listed at 2000 ms.

I admit to becoming a little lost on the 'primeability' norming procedure as it appeared to mix times gathered across the pre-exposure, overt versus covert scanning response groups, and overt post scan responses of the covert group. In the end the researchers used the alternative of simply the RT during the first novel presentation of each item to produce a median split that they referred to as primeability. However, I thought that all subjects named half of the items aloud three times prior to entering the scanner, regardless of whether they were in the overt or covert groups. If so, then it is unclear why, given list counterbalancing, that these trials alone could not be used to rank order each stimulus in terms of its priming score. This would have the added benefit of being statistically independent of the reaction times actually observed during the later scanning, and it would seem less complicated than the procedure described in the manuscript.

December 04, 2020

We would like to thank the reviewers for their many insightful and helpful comments on our manuscript entitled "Enhanced inter-regional coupling of neural responses and repetition suppression provide separate contributions to long-term behavioral priming". We have provided point-by-point responses to all major and minor concerns below, and we have made a number of substantial changes to the manuscript. Of particular note, we appreciate the reviewers' concerns about the possibility of regression to the mean in the selection of strongly and weakly primeable items in the experiment and of the possibility of noise/selection bias in the results that may follow. In response, we have conducted a series of analyses to address this question directly which now appear in the supplementary materials (Supplementary Figures 5-7), and we can state confidently that any such effects are negligible and cannot explain our reported findings. We have also moved original Supplementary Figure S2 to a figure in the main text (new Figure 5) in order to highlight the novel within-participant association of priming magnitude (Primeability) relative to the more typical across-participant approach.

While recalculating results related to the analyses of regression to the mean, we identified and corrected 3 relatively minor mistakes present in the original manuscript: 1) the FDR correction used in the functional connectivity results was originally too stringent (correcting the individual Strong and Weak plots in Figure 6C for all comparisons rather than just those ROI combinations showing interactions); 2) one of the two participants whose data was supposed to be excluded from the effective connectivity analysis in the left panel of Figure 7D (due to the parameterization failing to converge) was originally included in the right panel of Figure 7D; after exclusion, the r - and P -values are slightly altered ($P < .0222$ compared to $P < .0393$ originally); and 3) the labels for ACC and R Fusiform were inadvertently flipped for analyses involving the full RS clusters (shown in Figure 3B), affecting Figures 5, 8 and Supplementary Figure 4. Corresponding corrections have been made to these figures.

We have further adjusted the manuscript in response to concerns that some of the proposed theories may not be testable with fMRI. Reviewer 3 notes that perhaps the strongest tests with fMRI would relate to predictions about connectivity, whereas tests of timing in neural activity would be potentially problematic due to the nature of the hemodynamic response, particularly in regard to predictions of the Facilitation and Predictive Coding models. We now more clearly highlight these issues in the introduction and discussion, focusing on the tests that prior work would suggest should be strengths of fMRI, namely tests related to connectivity (relevant to Synchrony, Predictive Coding and Sharpening models) and of the similarity of representations in neural activity relevant to the Sharpening model. We have nevertheless included the tests of timing, since there are actually significant effects that provide partial support for the Facilitation and Predictive Coding models, although with appropriate caveats.

We hope that with these changes the manuscript will now be acceptable for publication in Communications Biology.

Sincerely,
Stephen J. Gotts, PhD & Alex Martin, PhD

Reviewer #1 (Remarks to the Author):

1. Is there a problem of “regression to the mean” in the calculation of primeability? If one defines priming as the difference between NEW and OLD RTs, and the *same* NEW RTs are used to define primeability, then this is a serious problem, because the positive relationship in Figure 4B would simply be an artefact of regression to the mean. This is because if one draws RTs for NEW and OLD items from identical but independent distributions (i.e, with the same mean, i.e, with no true priming), one will still get a positive slope between NEW-OLD and NEW (in Matlab: “y = randn(100,2); figure, plot(y(:,1),y(:,1)-y(:,2),'o');”). However, if the trials are used to calculate NEW RTs for the x-axis (e.g, from pre-fMRI session) are independent from the trials used to define priming (e.g., NEW trials in the fMRI session), this is no longer a problem (in Matlab: “y = randn(100,3); figure, plot(y(:,1),y(:,2)-y(:,3),'o');”). I couldn't quite work out what the authors did, e.g, if different objects were used for different NEW conditions in different participants, but if there is any possibility of regression to the mean, then this should be scrapped.

Many thanks, Rik, for pointing out this issue. Our initial concern was to try to eliminate as much noise as possible in our estimation of naming times to NEW objects in order to minimize regression to the mean. This is why we tried to include as many naming trials as possible for each item – and included the during-fMRI naming responses. Since we only used overt naming responses for these norms, it is important to note that this potential issue has no impact on the analyses of the 28 Covert naming participants (since none of their behavioral data during fMRI was used in the norms). Using only the pre-fMRI naming sessions, each item's RT estimate was averaged over approximately 25 participants' correct naming trials (from the first of 3 pre-fMRI naming attempts). Including the overt naming data during fMRI, this number increases to an average of approximately 40 participants per item (adding about 15 data points per item to each of the two counterbalanced sets from the 32 Overt naming participants). What this ultimately means is that each Overt naming participant's fMRI trials contributed about 1/40th to the data on 100 of the 200 items (and nothing for the remaining 100 items). Our *a priori* expectation was that any selection bias that might result from the inclusion of each participant's data to these norms would be quite small and inconsequential, especially when calculating a simple median split in the average across all participants. We would also note that this issue is not exactly the same as would normally be considered in regression to the mean, since this usually has to do with selecting in one set and testing in a subsequent set (where noise in the first set leads to mis-estimation in the selection which is then unrelated to the noise in the second set). This is more like selecting trials of interest in the set itself based on a desired behavioral characteristic (e.g. only correct trials, or those trials later correctly versus incorrectly recognized as OLD in a subsequent recognition memory test).

In any case, we have now checked and quantified this issue in detail in response to your query (also noted by the other reviewers). We have presented these new analyses in Supplementary Figures 5-7 (also shown below for convenience). In order to eliminate the contribution of noise bias from any one participant, each participant's behavioral data during fMRI was excluded for the calculation of their Strong/Weak primeability selection – but the behavioral data of all other participants was included (in order to include as much data and eliminate as much noise in the average RTs as possible). This process resulted in a slightly different selection of Strong/Weak primeability for each Overt naming participant, and it retained the original selection for all of the Covert naming participants (whose fMRI data did not contribute to the original estimates). The tests of connectivity, both functional and effective

(Figure 6C, Figure 7A-D), and the within-participant Strong/Weak primeability RS analyses (Figure 5B) were then recalculated.

As shown in Figures R1-3 below (Supplementary Figures S5-S7 in the manuscript), the proportion agreement of Strong/Weak Primeability across Overt naming participants is quite high (median agreement = .99; Figure 1), with most participants differing from others by 1 item on the average (e.g. 1 item per 100 switches from Strong to Weak). These item switches also uniformly occurred for items near the median RT boundary, not for those at the extreme ends of the RT distribution. Accordingly, there are small quantitative changes to the priming effect size (Figure R1, right panel), beta weight (Figure R2A), and connectivity results (Figures R2B and R3), but the results remain qualitatively unchanged and virtually all of the original results continue to be significant and corrected for multiple comparisons. The two exceptions to this: 1) Weak versus Strong RS in the L Fusiform ROI, while still significant [$t(59) = 2.13, P=.038$], no longer remains FDR-corrected (Figure R2A), and 2) the effective connectivity Strong/Weak X OLD/NEW interaction from R STG -> R TP, while still significant [$F(1,171) = 6.094, P=.0291$], is no longer corrected for multiple comparisons by FDR (Figure R3).

These new analyses are now referred to on pp. 25-26 of the revised manuscript:

“Although the contribution of any particular participant’s response times during fMRI was expected to be minimal to the overall Strong/Weak Primeability norms, we also conducted a set of follow-up analyses in which each participant’s behavioral data was excluded from the calculation of the norms. This allowed selection of Strong/Weak Primeability to be completely independent of the analyses of the fMRI responses, with a unique selection of the Strong/Weak median split for each participant. As expected, there was no marked change in the pattern of results (see Supplementary Figures S5-S7).”

Figure R1. Left panel shows the proportion agreement among Overt Naming participants when the selection of Strong versus Weak Primeability is individualized by excluding each participant's data for their own Strong/Weak selection. Right panel shows that the priming magnitudes in Strong versus Weak conditions continue to be quite large when these individualized item selections are used (orange compared to blue bars).

A

B

Figure R2 (compare to Figure 5B and Figure 6C). (A) Weak versus Strong Primeability effects in RS are qualitatively similar with independent selection of Strong/Weak for all participants. Only the effects in L Fusiform fail to survive correction for multiple comparisons by FDR (results in Figure 5A, which were unaffected by Strong/Weak selection, also included in FDR calculation). (B) Functional connectivity effects are qualitatively unchanged and remain significant and corrected by FDR when Strong/Weak selection is independent for all participants.

Figure R3 (compare to Figure 7A-D in main text). Effective connectivity effects are qualitatively similar, with 2 of the 3 original Strong/Weak X OLD/NEW interactions remaining significant and corrected by FDR ($q < .05$). Brain-behavioral correlations involving the R TP->ACC connection and supporting the predictions of the Synchrony model also remain significant (shown in panel D).

2. Also, if primeability was a valid measure, I wasn't sure why a median split was used, rather than the (potentially more sensitive) use of the continuous measure of primeability for all the analyses in the paper?

This is an important question, and it highlights the fact that we must not have explained the measurement problem clearly enough. We are well aware of the statistical inefficiency of using a median split approach if a more continuous approach is possible (e.g. MacCallum et al., 2002, Psychological Methods 7, 19-40). If we were dealing with a method in which we had numerous measurements in time for each item (as one might have in EEG or ECoG at 1 per millisecond), this would indeed be possible. Connectivity could be calculated across timepoints between two ROIs within a trial. However, with only one time point per trial, it is actually impossible to estimate connectivity measures without binning trials in some way (calculating connectivity across trials rather than within individual trials, but still within-participant). In other words, with only one timepoint per trial, continuous correlations with behavior across trials are only possible for local, univariate measures that can be measured on individual trials (such as between single-trial BOLD amplitude and RT in individual voxels/ROIs) – but not for connectivity. We

could have included more than 2 levels (Strong/Weak), splitting the 100 OLD and 100 NEW trials into finer gradations (3, 4, or 5 levels). But even by 3 levels (25-30 trials per level), the stability and power in estimating Pearson correlation coefficients starts to degrade. We opted for the simplest version of within-participant priming strength, 2 levels, to maximize stability and reliability of these measures, allowing 40 or more trials per condition to contribute to the connectivity estimates. In response, we have added clarifying text to the methods on p.29:

“Employing a more continuous analysis of functional connectivity with priming strength was not possible within-participant, as only a single timepoint (the peak) was retained for each item; functional connectivity estimates therefore had to be calculated across items, with priming strength incorporated within-participant as a median split of items between Strong and Weak Primeability.”

3. I'm not sure the Predictive Coding model predicts a more negative coupling between regions after repetition. This model predicts that the top-down predictions will be more accurate after repetition, such that prediction error (assumed to relate to fMRI signal) is smaller. But I don't think this necessarily means that the trial-by-trial correlation between peaks of the fMRI responses in the two connected regions is negative, because it is not clear that more precise predictions (following synaptic change in the connections between regions) from the "higher" ROI necessarily means higher (or lower) peak responses in that ROI. In fact, I think you could argue the opposite, in the sense that when predictions are improved by priming, there is a stronger coupling between regions. Indeed, in two papers by Ewbank et al (2010, 2012) using Dynamic Causal Modelling (DCM), they found that repetition made top-down connections more positive (note that in DCM, positive connections mean that the rate-of-change of activity in the receiving ROI is positively related to the activity in the sending ROI). I think the authors need to spell out their precise reasoning here, which possibly includes some additional assumptions they are making about the Predictive Coding model, with which other readers might not agree.

This prediction was previously articulated by us in our 2012 discussion paper in Cognitive Neuroscience (Gotts et al., 2012, Cogn Neurosci 3, 227-37). The point that you are making here was also included in the response made in Ewbank & Henson (2012, Cogn Neurosci) in your related commentary on our paper (with the prediction again re-asserted in our response to the commentaries; see p. 251). Rather than being a cosmetic feature of the Predictive Coding model, we believe the claim (by Friston) of the top-down negative sign to be a central feature of the way that the algorithm works. We will briefly unpack our reasoning and present additional support here.

Friston formulated the Predictive Coding model in the context of David Mumford's previous work (e.g. Mumford, 1992; see also Rao & Ballard, 1999), arguing for top-down negative feedback, with the notion of residual positive feedforward activity (literally calculated as a subtraction) representing "prediction error". Other algorithms that similarly optimize model performance in response to data do not take this same tack, with prominent examples being Geoffrey Hinton's Boltzmann machine (which necessarily develops symmetrical feedforward and feedback connections) and recurrent and multi-layered connectionist networks utilizing the classic delta rule (with no sign constraint on the feedforward and feedback weights; see Gotts & Martin, 2014, Cogn Neurosci 5, 121-2, for further discussion). Predictive Coding and these other models actually calculate a similar algorithm overall by intrinsically contrasting two states (using subtraction), but they are implemented differently – and with different predictions for the neural

architecture and the dynamics of synaptic plasticity. Friston placed the model-based predictions and stimulus-driven activities in different pools of units so that they could be calculated simultaneously in the feedforward and feedback flow of activity, whereas Hinton's solution contrasts co-activities in two different phases that are typically implemented at different times (e.g. during active stimulation versus at rest, or sleep/wake, for which the Hebbian rule is sign-reversed between states). I.e. they roughly differ in implementing the critical contrast in space (different units) versus time (different temporal periods), respectively. For Friston's model, without the feedback connections being negative, we would assert that the algorithm will no longer calculate the desired gradient.

We would note that we are not the only ones to take this interpretation on Predictive Coding. Tai Sing Lee and David Mumford tested the negative feedback prediction directly in single-unit recordings in macaque V1 (Lee & Mumford, 2003, Journal of the Optical Society of America, 20, 1434-48), with the expectation that feedback from V2 onto V1 in matched receptive fields for an illusory contour should show a suppression of V1 activity (i.e. with no actual stimulation of the V1 cells from the retina). Instead, they observed excitation from V2 onto V1, leading them to formulate a revised Bayesian theory. As they state in their Conclusion section on p.1446:

"Central to our framework is the forward/backward mechanism that is embodied conceptually in many existing neural models.^{1,4,7,16,17} Here we attempt to reconcile a subtle, but important, difference between two competing schools of thought. In the adaptive resonance¹⁶ or interactive activation models,¹⁷ an active global concept will feed back to enhance the neural activities in the early areas that are consistent with the global percept. These ideas are supported by numerous neurophysiological experiments that show that higher-order information can enhance early visual responses.^{46,48,49} On the other hand, the efficient-coding¹ and the predictive-coding models⁷ emphasize that feedback serves to suppress the activities in the early areas as a way of "explaining away" the evidence in the earlier areas. This idea is supported particularly by some recent imaging experiments.^{7,72} In the latter class of models, only error residues are projected forward to the higher areas."

Importantly, with respect to Friston's Predictive Coding model, the feedback connections are defined to be reciprocally negative to the feedforward connections. From Friston (2005), p. 823 (bottom, 1st column) (see related Figure 2 from Friston, 2005, below):

"In the hierarchical scheme, the dynamics of representational units ϕ_{i+1} are subject to two, locally available, influences. A likelihood or recognition term mediated by forward afferents from the error units in the level below and an empirical prior conveyed by error units in the same level. Critically, the influences of the error units in both levels are mediated by linear connections with strengths that are exactly the same as the **(negative) reciprocal connections** from ϕ_{i+1} to ξ_i and ξ_{i+1} ." **[emphasis added]**

Figure 2 from Friston (2005). Feedforward interactions within the hierarchy are sent from the prediction error units (ξ_i) in superficial layers of level i to the units representing the conditional expectation of causes (ϕ_{i+1}) in level $i+1$. Feedback is sent in the reverse direction from ϕ_{i+1} to ξ_i with a negative sign [via term $-g(\phi_{i+1}, \theta_i)$].

Friston (2005) clarifies further in Endnote 2 that the feedback connections needn't be directly inhibitory as long as they are effectively inhibitory, perhaps mediated by inhibitory interneurons (p. 834, bottom first column):

² “Clearly, in the brain, backward connections are not inhibitory. However, after mediation by inhibitory interneurons, their effective influence could be thus rendered.”

Later in the paper, Friston directly addresses the phenomenon of repetition suppression. From Friston (2005), p. 829 **[emphasis added]**:

6. IMPLICATIONS FOR SENSORY LEARNING AND ERPs

In §5, we introduced the notion that evoked response components in sensory cortex encode a transient prediction error **that is rapidly suppressed by predictions mediated by backward connections**. *If the stimulus is novel or inconsistent with its context, then this suppression is compromised*. An example of this might be extra classical effects expressed 100 ms or so after stimulus onset. In the following, we consider responses to novel or deviant stimuli as measured with ERPs. This may be important for empirical studies because ERPs can be acquired noninvasively and can be used to study humans in both a basic and clinical context.

(a) Perceptual learning and long-latency responses

The E-step in our empirical Bayes scheme provides a model for the dynamics of evoked transients in terms of the responses of representational and error units. Representational learning in the M-step models plasticity in backward and lateral connections to enable more efficient inference using the same objective function. *This means that perceptual learning should progressively reduce free energy or prediction error on successive exposures to the same stimulus. For simple or elemental stimuli, this should be expressed fairly soon after stimulus onset; for high-order attributes of compound stimuli,*

*later components should be suppressed. **This suppression of responses to repeated stimuli is exactly what one observes empirically and is referred to as repetition suppression** (Desimone 1996). This phenomenon is ubiquitous and can be observed using many different sorts of measurements.*

In other words, a main source of RS in this model is an incremental reduction of prediction error (activity of units in superficial layers) mediated by enhanced top-down suppression of activity (via the M-step). For novel items, there's relatively little top-down suppression. For repeated items, there is more. Hence, the prediction of greater negative top-down feedback and RS with repetition.

However, we would concede that this is not the only potential source of RS in this model. As you mention, enhanced precision leading to higher gain of the connections could also contribute by leading to earlier onset and termination of activity – potentially observed in BOLD fMRI as an overall reduced level of activity. So, while we feel that the above facts merit the stated prediction about negative top-down connections and RS magnitude, we are happy to soften our conclusions by mentioning the potential additional role of increased precision – which fMRI admittedly does not have the temporal resolution to evaluate well. We have added new text outlining this qualification:

pp.5-6: “The Facilitation model is less clear regarding predictions about connectivity but predicts that neural responses in regions exhibiting repetition suppression should peak and terminate earlier for repeated objects, **with a similar prediction made by the Predictive Coding model under conditions of high model precision²¹.**”

p.15: “**Similarly, under conditions of high model precision and enhanced processing gain, the Predictive Coding model explains priming through a speeding up of neural activity, with this speeding possibly generating an additional component of repetition suppression²¹.**”

4. Continuing Point 3, there are convincing arguments why DCM is superior to the SEM used by the authors for the purposes of determining effective connectivity in fMRI (arguments including the use of differential equations to properly infer directionality, the use of ROI-specific haemodynamic parameters, and the use of a Bayesian framework in order to more flexibly compare models, e.g, those with closed loops or non-nested). I appreciate that DCM is complex and difficult to use, and some people do not trust the results (e.g, the assumptions behind its use of Bayesian model evidence), but it would at least be interesting to know whether it provides converging results to those obtained by the authors' SEM (which is a static method based only on the covariance between ROI fMRI timeseries, rather than underlying neural timeseries).

Our initial plan in this project was to use DCM, since we thought that it would be the most persuasive to proponents of Predictive Coding. However, after reading through the SPM manual sections on it, it became clear that it is not a “stand-alone” analysis and requires expertise with SPM more generally. We were not confident in our ability to implement DCM correctly for this set of data, and we were unable to find local collaborators (in the AFNI group and broader community) with the needed expertise. We therefore opted to use SEM, the effective connectivity method employed by the AFNI group (advising us extensively on its use) and that seemed most appropriate for the type of data that we had. We also noted the recent

paper in Nature Neuroscience by Reid et al. (2019) (with Michael Cole, Russ Poldrack, Vince Calhoun, Bharat Biswal, Steve Hanson and others) advocating the use of SEM more broadly in contexts such as ours.

We agree that it would be very interesting to compare the current results to those with DCM, and we would be quite open to a future collaboration with researchers with the right expertise to look at that. However, for the current paper, we don't have any reason to doubt the appropriateness of SEM, and it has plenty of proponents in the field.

5. Continuing Point 4, the results of the functional and effective (SEM) connectivity are somewhat unexpected, since they do not involve connections between the main, a priori ROIs (i.e., those showing RS, e.g, fusiform and inferior frontal). While it is theoretically interesting that ROIs can show repetition-related changes in connectivity without repetition-related changes in activity, the role of the ROIs that show connectivity changes in priming is unclear (they are not typically associated with perceptual or semantic or verbal processing for example), and if the main purpose of the paper is to compare theoretical models of RS, then what is the relevance of ROIs that don't show RS? Moreover, since the actual goal of DCM is to explain condition-related changes in activity through changes in connectivity, a DCM network based on those ROIs that show RS (eg between inferior frontal and fusiform ROIs) would seem potentially much more informative about the authors' main question.

Our top priority in this project was to conduct a whole-brain search for brain regions showing repetition-related effects on connectivity that were also related to priming, given the central relationship of connectivity to ideas about RS and priming. Accordingly, we adapted the degree-centrality-based methods ("connectedness" followed by seed-based post-hoc tests) that we have used in numerous other experimental contexts (e.g. Gotts et al., 2012, Brain; Song et al., 2015, J Cogn Neurosci; Berman et al., 2016, Brain; Jasmin et al., 2019, Brain). We fully expected that this would involve regions showing RS. While some of the regions showing RS are involved in these changes (e.g. R Fusiform and ACC), many are not. In fact, if you restrict yourself only to those ROIs showing RS (which we have done for both functional and effective connectivity a la SEM), there are no significant functional or effective connectivity effects. For SEM, since effective connectivity is modeling what is seen in functional connectivity – and given that there are no functional connectivity effects – this is perhaps not surprising. But it is surprising that such large RS effects can be seen simultaneously with virtually no functional connectivity changes amongst the RS ROIs.

Combined with the observation of significant correlations between effective connectivity and priming magnitude between one region showing RS (ACC) and one showing no hint of RS (R TP) (see Supplementary Figure S3), not to mention that this variance is virtually unrelated to that seen in RS-priming correlations (via the partial correlation analyses that we conducted), we think this suggests the need to look beyond just those regions showing RS to understand the full picture here. Indeed, to the extent that functional connectivity effects are observed among non-RS regions, how valid would effective connectivity models be that exclude them? What made the most sense to us was to include comprehensively all regions showing either functional connectivity or RS effects and allow the effective connectivity modeling to sort out which connections were most responsible for the functional connectivity changes.

We would also note that this is not the only example of which we are aware showing a dissociation of beta weight and connectivity effects with respect to behavior. A recent paper from our lab examining the neural bases of recognition memory for faces has reported a qualitatively similar dissociation, with connections among the well-known face patches showing little correlation with face memory performance despite correlations with the beta weights in these regions (Ramot, Walsh, & Martin, 2019, *J Neurosci* 39, 4976-85). Instead, the connections of the face patches with systems outside the traditional face processing network (with medial temporal lobe structures and regions associated with social processing) were most associated with face memory performance. We have now included a reference to this study in the discussion section on p.18:

“In any case, a central finding of the current study is that repetition suppression and coupling each account for unique portions of the variance in priming and are expressed in only partially overlapping brain regions. This is reminiscent of recent findings from another study, in which beta-weight correlations with face recognition memory were found locally to traditional face processing regions, whereas connectivity-based correlations with face memory lay mainly in relationships with regions outside of the traditional face-processing network.⁴¹”

6. A minor quibble with the predictions stated for the Facilitation model in the Introduction: “responses in regions exhibiting RS should be temporally advanced for repeated objects, with earlier onset and peak”. The Facilitation model assumes that repetition speeds up the (competitive) neural dynamics that resolve perception of an object (such as the dynamic reduction of prediction error). The simplest prediction is that the centre of mass of the firing rate (over a few hundred milliseconds after stimulus onset) occurs earlier for repetitions. However, this need not reflect a change in the onset of neural firing, or even the peak; it could reflect solely a reduction in the duration of neural firing. Moreover, when the authors talk about onset and peak, the authors should clarify whether they are talking about the neural or fMRI signal. After convolution with a hemodynamic response, any facilitation effects on the onset of the BOLD response would be difficult to detect – they are more likely to be seen in the latency (and magnitude) of the peak of the BOLD response (which is actually what the authors measured, so mention of the onset is confusing here).

The James et al. (2000) and James and Gauthier (2006) articulations of the Facilitation model actually do focus more on the timing of peak neural response. We agree with you, though, that this is not strictly necessary. The time of the peak firing rate response is not likely to be a meaningful measure with respect to the responses in downstream brain regions, as cells spike continuously during the presentation of a stimulus, not just after some peak firing rate response is achieved in a sending region. All that is necessary to explain reduced BOLD responses is less overall area under the firing rate curve, which could be manifest multiple ways – including having an identical onset with earlier termination of activity.

In response to this comment and in response to concerns of reviewer 3 and the editor, we have qualified these analyses. As you point out, the underlying neural response timing and the timing of the BOLD response here are necessarily ambiguous with respect to one another. In the revised manuscript, we have placed more weight on the predictions for which fMRI is best suited. We have nevertheless included the tests of timing, since there are significant effects observable using the BOLD response, but we add the following new text to the manuscript:

p.6: “A strength of BOLD fMRI is its potential for examining changes in connectivity at the scale of the whole brain, as well as its ability to evaluate local changes in representational similarity, germane to predictions of the Synchrony, Predictive Coding, and Sharpening models. The ability of BOLD fMRI to adjudicate subtle changes in the timing of neural activity is less clear, but we nevertheless examine the predicted timing changes of the Facilitation and Predictive Coding models in the event that relevant timing information is detectable. What this means is that it will not be possible to test all four models on equal terms, but the results will have the potential to favor certain models over others, as well as to clarify the relationships between changes in connectivity, repetition suppression, and representational similarity.”

p.19: “In our view, it will take multiple studies across a range of methodologies to fully rule out any of these models, and the primary contribution of the current study is its whole-brain assessment of changes in connectivity and their relationship to repetition suppression and priming magnitude, with more moderate progress on model evaluation.”

7. In terms of access to the data and code, I don't think it's enough to simply say you need an XNAT server to get the data (without further information on data location) and to email authors for the code. Many papers provide these openly (e.g, for reviewers to assess), eg on OSF or other website.

We are happy to make the code freely available, but these data were unfortunately collected several years ago under an older protocol that did not make allowances for public data sharing. All of our current experimental protocols include data sharing consent, but our IRB has made clear to us that we would need to re-consent all of the subjects in the study in order to share them without constraint in a public database. This intermediate solution of sharing on XNAT is what our IRB has allowed us to do. We have now added the link to the XNAT database to the Data Availability statement, and the code will be posted to SG's ResearchGate website: https://www.researchgate.net/profile/Stephen_Gotts/publications.

Reviewer #2 (Remarks to the Author):

While the experiment and results are noteworthy and have the potential to deepen our understanding of repetition priming, there are 2 major issues that would need to be addressed before moving forward. First, it is very unclear how well the study is able to adjudicate between the 4 theoretical perspectives. The introduction mentions in a single sentence that each model “makes unique predictions”, but then goes on to focus the paper on something common between them, synchrony – “Common to most of these models is the prediction that measures of connectivity between brain regions should be affected by repetition.” This leaves the reader with the sense that the work will not be able to falsify any of the existing models. However, the analysis then proceeds to test various aspects of select models posthoc – peak timing (predictive coding model) & similarity of voxel activation patterns between new and repeated items (sharpening model). Leaving aside whether fMRI is an appropriate methodology for testing differences in neural timing (addressed in the discussion) – this approach leaves the

reader with a very muddled picture about what aspects of the 4 models were tested and what the results tell us.

We apologize for the lack of clarity. It is true that 3 of 4 main models make predictions about connectivity, but they are not the same predictions. We would also emphasize that they are not post-hoc in the sense that they were articulated in our 2012 discussion paper in Cognitive Neuroscience that was published prior to the collection of these data (Gotts, Chow, & Martin, 2012, Cogn Neurosci 3, 227-37). The central features of these models have not changed since then, and the predictions follow from their central features. In response, we have altered the introduction text in order to make the distinctions between the models clearer, as well as to make clearer which aspects are better versus more poorly suited to testing by fMRI. (pp. 5-6):

“While these models are not necessarily mutually exclusive, each makes unique predictions that can be tested. The Synchrony, Predictive Coding, and Sharpening models differ from one another with regard to their predictions about the effect of repetition on connectivity between brain regions. For example, the Synchrony model predicts increased positive inter-regional coupling with repetition. The Predictive Coding model, in contrast, predicts that top-down connections should have stronger negative coupling with repetition and that this coupling should correlate with the magnitude of repetition suppression in the regions receiving this input. The Sharpening model, like the Synchrony model, predicts an increase in positive coupling, but it further requires the sending region to exhibit repetition suppression, accompanied by a reduction in the similarity of neural responses in the sending region due to reduced overlap among neural representations. The Facilitation model is less clear regarding predictions about connectivity but predicts that neural responses in regions exhibiting repetition suppression should peak and terminate earlier for repeated objects, with a similar prediction made by the Predictive Coding model under conditions of high model precision²¹. Finally, each model predicts that behavioral priming magnitude should be related to its core changes, with repetition suppression serving as an indirect marker of the operation of its critical underlying mechanisms. **A strength of BOLD fMRI is its potential for examining changes in connectivity at the scale of the whole brain, as well as its ability to evaluate local changes in representational similarity, germane to predictions of the Synchrony, Predictive Coding, and Sharpening models. The ability of BOLD fMRI to adjudicate subtle changes in the timing of neural activity are less clear, but we nevertheless examine the predicted timing changes of the Facilitation and Predictive Coding models in the event that relevant timing information is detectable. What this means is that it will not be possible to test all four models on equal terms, but the results will have the potential to favor certain models over others, as well as to clarify the relationships between changes in connectivity, repetition suppression, and representational similarity.**”

The second major issue is the choice to divide objects into high and low “Primeability”, based on the naming time for the items when first presented. Leaving aside the issue of whether it is ever a good idea to mean/median split a variable that shows continuous characteristics – (see - MacCallum, R. C., Zhang, S., Preacher, K. J., & Rucker, D. D. (2002). On the practice of dichotomization of quantitative variables. Psychological Methods, 7(1), 19–40.), there is no reference for taking this approach, no work where it was previously done and therefore has a posthoc feel to it - we didn’t find what we wanted when we included all items, so we divided them into 2 categories. If this is in fact

the case, then this is an exploratory analysis and should not be the key approach to analyzing the question of interest. If this is not the case then a more convincing justification is needed that is based on the vast priming literature that doesn't seem to ever take this approach.

We fully appreciate the point that the reviewer is making about median splits, and we are already familiar with the MacCallum et al. paper and agree with the statistical points. Reviewer 1 (Rik Henson) made the same point above. The comment highlights the fact that we must not have explained the measurement problem clearly enough. If we were dealing with a method in which we had numerous measurements in time for each item (as one might have in EEG or ECoG at 1 per millisecond), a more continuous approach would indeed be possible. Connectivity could be calculated across timepoints between two ROIs within a single trial. However, with only one time point per trial, it is actually impossible to estimate connectivity measures without binning trials in some way (calculating connectivity across trials rather than within individual trials, but still within-participant). In other words, with only one timepoint per trial, continuous correlations with behavior across trials are only possible for local, univariate measures that can be measured on individual trials (such as between single-trial BOLD amplitude and RT in individual voxels/ROIs) – but not for connectivity. We could have included more than 2 levels (Strong/Weak), splitting the 100 OLD and 100 NEW trials into finer gradations (3, 4, or 5 levels). But even by 3 levels (25-30 trials per level), the stability and power in estimating Pearson correlation coefficients starts to degrade. We opted for the simplest version of within-participant priming strength, 2 levels, to maximize stability and reliability of these measures, allowing 40 or more trials per condition to contribute to the connectivity estimates. In response, we have added clarifying text to the methods:

p.29: “Employing a more continuous analysis of functional connectivity with priming strength was not possible within-participant, as only a single timepoint (the peak) was retained for each item; functional connectivity estimates therefore had to be calculated across items, with priming strength incorporated within-participant as a median split of items between Strong and Weak Primeability.”

It is true that this approach differs from previous studies. In fMRI, a big part of that has been the inability to separate out individual trial responses when using a typical rapid event-related design. We are attempting to afford this kind of analysis with our use of a slow event-related design and with more extensive de-noising akin to resting-state preprocessing than is typically done in task-based fMRI studies. A common criticism of task-based functional connectivity is that the evoked response is simply aliased into the connectivity measure by virtue of the shape of the evoked response (which is nigh impossible to remove in something like a block design, since most of the stimulus-to-stimulus variability within a block is unmodeled by the block regressor). This led to the use of single-trial peak BOLD responses here.

Furthermore, many priming studies are solely behavioral, with no need to engage in brain-behavior correlations. Those that have performed brain-behavior correlations have commonly used inter-participant variability in a summary measure of priming strength for the single participant, along with inter-participant variability in RS or some other measure (as in our Figure 5A). That's not necessarily because it's the best thing to do but because it's the easiest and most straightforward thing to do (as discussed at length in the paper by Lebreton et al., 2019, *Nat Hum Behav* 3, 897-905, that we cite in the manuscript). We've been wanting to add a within-participant measure of priming magnitude for some time in order to check the assumptions implicitly made by the inter-participant variability approach. How can we be sure that we are really measuring correlations of priming and not of some other non-specific factor

that varies across individuals (such as processing speed or attentional ability)? This concern also underlies the common practice in word processing research of using both item and subject as the random variable in ANOVA. If an effect is only significant by subject and not by item, it's not taken as seriously.

In the current study, with the limit of one fMRI measurement per trial and the need to evaluate connectivity for the predictions of interest, we had to adopt an approach along these lines in order to include a within-participant estimate of priming and connectivity strength. In the case of the highlighted finding (effective connectivity from R TP->ACC), it is noteworthy that the within-participant and across-participant correspondences between priming strength and connectivity strength both agree (Figure 7C and 7D) (as well as the results shown in Figures 5A and 5B).

In general, the priming literature has focused on the reliability of repetition priming rather than its magnitude. Slower initial items may show larger priming effects just due to mean regression around a primed mean.

The reviewer raises the issue of regression to the mean when selecting Strong/Weak priming based on naming times to novel pictures (also mentioned by Reviewer 1). This is quite possible if one were to select Strong/Weak priming conditions based on a single participant's response times (given the high level of noise present in the responses, as shown in Supplementary Figure 1). However, it is much less likely when averaging responses across many participants. If enough data were used to make the median split effectively noiseless (i.e. the average RTs become so stable and precise that adding more data no longer changes the rank ordering of the items), then regression to the mean due to the selection could be completely eliminated.

One easy way to see whether Strong/Weak primeability is due to regression to the mean (based on the initial RT) is to apply the selection to an independent dataset. That was essentially already presented in the previous manuscript in one example. The Strong/Weak selection did not include any data from the fMRI sessions of the Covert Naming participants, yet the results in Figure 3C show that they hold when applied to the button press times provided by the Covert Naming participants during fMRI. We have also applied the Strong/Weak selection determined from data in the current experiment to the picture naming conditions from another dataset that we recently published (Gilmore et al., 2019, Identifying task-general effects of stimulus familiarity in the parietal memory network. *Neuropsychologia* 124, 31-43). The impact of amount of data averaging (either just using the pre-fMRI trials or all Overt Naming trials, including fMRI) is shown for both the Gilmore et al. (2019) and Covert Naming fMRI trials (current study) in **Figure R4**. Approximately 25 naming trials are averaged per item for the pre-fMRI only selection, whereas approximately 40 naming trials are averaged per item for all Overt Naming trials (original selection).

Figure R4. Selecting Strong/Weak priming from behavioral data in the current experiment replicates in independent data (left panel: data from Gilmore et al., 2019; right panel: Covert Naming button press times during fMRI in the current experiment). Strong>Weak priming effects are significant when selecting based on approximately 40 naming responses per item (Original), and even if only based on approximately 25 naming responses per item (pre-fMRI data only). However, these effects are stronger, with less regression to the mean, when selecting based on more data – presumably because of greater noise cancellation across participants/trials (Original).

We have also examined the impact of excluding each participant's own behavioral data during fMRI (which would eliminate any noise bias from their own data) on the magnitude of the Strong vs Weak priming effects. In order to examine this, we individualized the Strong/Weak selection for each Overt Naming participant such that their fMRI naming trials were excluded from the group-average naming times, recalculating the median split of Strong/Weak. This led to slightly different selections of Strong/Weak for each Overt Naming participant (the original selections are used for the Covert Naming participants as none of their fMRI data contributed to the norms). As shown in **Figure R5** below, the proportion agreement of Strong/Weak Primeability across Overt naming participants is quite high (median agreement = .99), with most participants differing from others by 1 item on the average (e.g. 1 item per 100 switches from Strong to Weak). These item switches also uniformly occurred for items near the median RT boundary, not for those at the extreme ends of the RT distribution. The Strong vs Weak priming effects remain highly significant. We have also checked that none of the main results reported in the manuscript are lost when calculating Strong/Weak selections in this way (see Supplementary Figures S5-S7; repeated above as Figures R2-R3 in response to Reviewer 1). We are therefore confident that our results are not due to regression to the mean.

Agreement of Strong/Weak Primeability Selection Among Overt Naming Participants

Mean Agreement = 0.9867
 Median Agreement = 0.99
 (min = 0.97, max = 1.0)

Priming Effects in Overt Naming Remain Large when Strong/Weak Selection is Independent

Figure R5 (repeated above to Reviewer 1 as Figure R1). Left panel shows the proportion agreement among Overt Naming participants when the selection of Strong versus Weak Primeability is individualized by excluding each participant's data for their own Strong/Weak selection. Right panel shows that the priming magnitudes in Strong versus Weak conditions continue to be quite large when these individualized item selections are used (orange compared to blue bars).

In addition to these 2 major issues, there are some lesser items that could be addressed

1 – alpha levels throughout seem arbitrary. These should be set up front and consistent across all major analysis.

We have clarified the choice of alpha levels for the analyses in this study in a new methods section on p.26 entitled “Choice of Statistical Thresholds for fMRI Analyses”. There are really 3 sets of whole-brain analyses, for which alpha choice is particularly critical: set 1) tests of OLD versus NEW to determine Repetition Suppression (or Enhancement) in Overt and Covert Naming (Figure 3B), set 2) tests of stimulus versus baseline in Overt and Covert Naming (Figure 3A), and set 3) tests of OLD/NEW and OLD/NEW X Strong/Weak in seed-detection and seed testing. From pp.26-27:

Choice of Statistical Thresholds for fMRI Analyses. The overall analysis strategy was to first identify brain regions exhibiting repetition suppression using typical GLM contrasts of OLD versus NEW. The voxelwise alpha levels on these contrasts were chosen such that False Discovery Rate (FDR⁶⁴) indicated that fewer than 1 voxel in each task (Overt Naming, Covert Naming) could be due to chance, with the interpretation that surviving voxels replicate across tasks (in this case, a voxelwise threshold of $P < .00001$, with FDR correction to $q < .00006$). This high thresholding also broke apart large clusters and afforded detection of isolated regions of interest (ROIs). We also required that any such voxels exhibited above baseline levels of activity at a minimum level of significance ($P < .05$, $q < .05$). Whole-brain searches for repetition-related changes in functional connectivity were conducted on the same data sample used for the identification of repetition suppression ROIs. For whole-brain searches of functional connectivity using

cluster-size correction, we chose a voxelwise alpha of $P < .001$, correcting for multiple comparisons to a familywise Type-I error rate of $P < .05$, as this has been shown to control for false positive rates at 5% or less when using an empirical autocorrelation function estimation for cluster distribution^{65,66}. All subsequent analyses utilized FDR to correct for multiple comparisons, with the FDR controlled at $q < .05$.

Given the goals of the study, we think it less important to have a fixed voxelwise alpha for all analyses, since it would prevent the isolation of ROIs if we were to use $P < .001$ (most of the ROIs would remain connected in a very large cluster given the magnitudes of the RS effects). It is also important to point out that the more critical alpha – that related to the familywise error rather than the voxelwise Type I error rate, is set consistently at $P < .05$. FDR thresholds are also set to be consistent at $q < .05$ throughout the study (with the exception of the ROI isolation procedure).

2 – the approach to connectivity is not something common to the literature – “The response peak was then notched out of the time series, with connectivity calculated as the co-fluctuation (correlation) of peak responses across trials between pairs of regions/voxels” – could a reference be provided?

The approach to task-based (as opposed to resting state) connectivity is novel to this paper. It is designed to eliminate the contour of the BOLD response from the task-based connectivity estimates. Figure 2 and its caption are dedicated to explaining this choice, along with text from the Online Methods on p. 28:

“In order to avoid contamination of task-based functional connectivity estimates from the temporal contour of the evoked responses of individual trials, only the average peak BOLD response was retained from each individual trial (the raw average of the 3rd and 4th timepoints of the task residual timeseries following the onset of each stimulus).”

Since we are employing Pearson correlation for the functional connectivity measures, which is an average of the products of individual time points taken from the z-transformed time series, eliminating TRs from the average is not necessarily expected to result in large qualitative differences from using the full time series for the calculation (with the Pearson correlation when using the full timeseries being roughly comparable to the weighted average of the Pearson correlations when using the peak TRs and the correlation when using the non-peak TRs). The approach has the advantage of preventing the shape of the HRF (or changes in the shape of the HRF) from becoming aliased into the connectivity measures (because the HRF has been reduced to a single data point). One precedent that involves excluding TRs when calculating Pearson correlations in functional connectivity is motion scrubbing (Power et al., 2012), although the TR removal probably happens to a larger degree in the current method than for motion scrubbing in participants with small to moderate motion.

Aside from the notching out of peak TRs, the overall connectivity approach in terms of connectedness and seed-based testing have numerous references, and these are already included in the relevant section of the Online Methods (*fMRI Task-Based Functional Connectivity Analyses*) on pp. 28-31.

3 – The paper says - “failed to exhibit increased feed-forward coupling during the processing of OLD relative to NEW objects” – this is interpreting a null finding about a

level of connectivity detail that fMRI is likely not suited for - is fMRI with a TR of 2 sec appropriate to determine “peak timing” differences of less than 500msec? Since the peak is determined by fitting to a standard model, there are many reasons the peak might change – number of trials fit, differences in the variability of response – something seen in the ERP literature. The authors themselves question the use of fMRI to examine subtle timing differences as well as extent of activation, yet these 2 approaches form critical tests of the competing theories.

It is true that when we did a comprehensive search of regions showing connectivity, we failed to find support for certain models. In some cases, these are null effects. If that’s all that we had to say, then we agree that this is not very interesting. However, we did observe new findings, and those findings happened to support one of the theories more than the others, based on the more detailed predictions that each model has for connectivity and RS. We also observed some contrary evidence (such as significant positive rather than negative connections related to the Predictive Coding model), and through partial correlation analyses, we established that the connectivity effects that we did observe were at least partially independent of the RS effects. This may be a more subtle verdict on the theories than desired, but it’s also hard to imagine it being otherwise, unless we observed significant effects in connectivity in the incorrect direction for some of the theories. The bottom line is that we have new findings that are more consistent with certain theories than they are with others. It would be too strong to say that we have ruled out any of these theories, but those theories are also not necessarily accounting for the findings currently on the table either, which means that they’ll need modification. In our view, that’s good progress from a single study.

If one thinks about opportunities afforded by other methods, there are actually significant downsides to all choices. If you choose ECoG, you will not get full brain coverage, but you will have an appropriate measure of RS (high frequency broadband power decreases) and high temporal resolution. However, you probably won’t have the subject numbers needed to do across-subject correlations with priming, and you’ll need an approach similar to our current within-participant approach (Strong/Weak). If you choose EEG/MEG, there are real questions about what the corresponding choice of measure is for RS compared to RS measured in firing rate or BOLD fMRI (discussed in Gilbert et al., 2011, Front Hum Neurosci). EEG and MEG measure temporal fluctuations in current/magnetic field, not necessarily amplitude of activity (these can potentially index non-overlapping aspects of activity). You will also have to choose a form of source estimation, which introduces additional questions/assumptions. In our view, only with a mixture of all available approaches will we be able to test all of the various features of the existing models. We are currently attempting to bridge the gaps between some of these methods by testing predictions in ECoG (with Kareem Zaghloul at NINDS/NIH), as well as in simultaneous fMRI/EEG. However, fMRI is capable of testing quite a range of the predictions well, if not so well for subtle differences in timing of the responses related to the Facilitation model – although there are indeed relevant observable effects in fMRI that match the expectations of this model.

As discussed above, we have sought to be clearer in explaining the limits of fMRI in evaluating the 4 models:

p.6: “A strength of BOLD fMRI is its potential for examining changes in connectivity at the scale of the whole brain, as well as its ability to evaluate local changes in representational similarity, germane to predictions of the Synchrony, Predictive Coding, and Sharpening models. The ability of BOLD fMRI to adjudicate subtle changes in the timing of neural activity is less clear, but we nevertheless examine the predicted timing

changes of the Facilitation and Predictive Coding models in the event that relevant timing information is detectable. What this means is that it will not be possible to test all four models on equal terms, but the results will have the potential to favor certain models over others, as well as to clarify the relationships between changes in connectivity, repetition suppression, and representational similarity.”

p.15: “As discussed above, fMRI is not the optimal technique to evaluate this prediction, but we nevertheless examined changes in the timing of the BOLD response in the event that such changes were detectable.”

p.19: “In our view, it will take multiple studies across a range of methodologies to fully rule out any of these models, and the primary contribution of the current study is its whole-brain assessment of changes in connectivity and their relationship to repetition suppression and priming magnitude, with more moderate progress on model evaluation.”

4 – The paper says – “With respect to the Facilitation model, neural responses were indeed found to peak at earlier times for OLD compared to NEW objects in regions showing repetition suppression, although these effects failed to interact with primeability”. This seems an odd statement, given that the facilitation model has never made any predictions with respect to slower vs faster identified/named items.

It is true that no proponent of the facilitation model has made claims about slower versus faster identified items. However, if it is to serve as a unitary account for RS and priming, one would hope that it would track the differential magnitudes of priming in the two conditions. We have clarified now on p.17 that this was not necessarily directly claimed by authors such as James, Gauthier and colleagues:

p.17: “On this point, it is important to mention that none of the original proponents of these models have made explicit claims about differential timing for different types of items (e.g. greater change in the peak timing for more strongly primed items that are identified initially more slowly). Nevertheless, we believe that this prediction should follow if these models are to serve as a unifying explanation of repetition suppression and priming.”

5 – Finally, the changes in synchrony are examined in a circumscribed set of regions that showed univariate priming effects. Why not take a larger connectome approach to changes in connectivity? The current approach would hide whether priming results in increased/decreased connectivity in other regions that don’t show repetition suppression effects. One might speculate that a sharpening model might reveal decreased connectivity in regions NOT critical to primed responses.

We perhaps did a poor job explaining this, but the “connectedness” approach that we have applied here is intended to do what the reviewer is asking for. If one breaks the brain up into a set of discrete voxels or ROIs, a whole-brain search for connectivity effects has to correct somehow for $N*(N-1)/2$ comparisons where N equals the number of voxels/ROIs (with the expectation that one will almost always fail to find effects when facing Bonferroni or even often FDR correction). If one compresses the voxelwise connectivity into an average (“connectedness”), where each voxel represents the average Pearson correlations with all of the

other voxels, then there are only N comparisons to correct for. If done at the voxel level, this can be done with cluster-size correction, just like a typical whole-brain search for beta weights. This is outlined in a series of papers that we published utilizing resting-state fMRI to compare connectivity in clinical and control groups (e.g. Gotts et al., 2012; Meoded et al., 2015; Berman et al., 2016; Jasmin et al., 2019; Stoddard et al., 2016). We have simply applied this approach to the current context. We then formed a union of all ROIs showing any connectivity effects and those showing RS effects.

Reviewer #3 (Remarks to the Author):

This is the first review of ‘Enhanced inter-regional coupling of neural responses and repetition suppression provide separate contributions to long-term behavioral priming’ by Gotts, Milleville and Martin. The work looks at behavioral priming in naming times for visual stimuli using a slow event related fMRI design and several sophisticated analyses. The paper is quite dense given the number of complex analysis methods employed and the brief format and I had some difficulty keeping track of the methods and how they related to the abstract priming models discussed. As I note below, the impact of the work potentially suffers from the fact that the chosen method cannot possibly critically evaluate the relative merit of the four abstract priming mechanisms that are introduced at the beginning. The authors acknowledge this in the Discussion but until that point, the paper is framed as though the data are capable of ruling out or clearly ranking the model competitors.

Additionally, this undercuts the key aspect that is highlighted in the title, namely that the work may demonstrate that unique portions behavioral variance in priming may be accounted for by repetition suppression versus connectivity. Convincing a skeptical audience of this, and perhaps linking it different metrics of the materials would be quite exciting, but the latter isn’t done at present. Below I elaborate on the main concerns that arose during the review, followed by a brief section outline more minor concerns.

We thank the reviewer for his/her many helpful and insightful comments. We respond to each in turn below.

MAJOR

What are the relative disadvantages of the fMRI method for the competing models? What I mean by this is that, depending upon the predictions of the four coarse models, some may fare better than others because of the unique characteristics of the current methods. For example, those models which predict temporal changes in activity are presumably hamstrung by the temporal filtering properties of the BOLD response and the chosen TR of 2000 ms. In contrast, the method may be more reliable for capturing interregional connectivity particularly since (I think) the authors are using residuals that are likely to depend upon lower frequency information. Ideally one would want to have similar power to detect the key predictions of each of the competitors. I discovered that the authors acknowledge this limitation in the Discussion section, but it undercuts the framing of the entire paper as a test among competing explanations of priming, which are themselves not mutually exclusive. [..]

This is an excellent point and was also made by the editor and other reviewers. We agree that fMRI is not the best choice *a priori* to test predictions about subtle differences in latency.

However, as the reviewer notes, a strength of fMRI should be its ability to examine predictions about changes in connectivity and how these relate to RS. Based on the rather large fMRI MVPA (multi-voxel/ivariate-pattern analysis) literature, one might also expect fMRI to have a strength in evaluating the prediction of the Sharpening model regarding decreased representational similarity. There is a fine line between framing the paper as an attempt to test all four models on equal footing (not that we actually claimed equal footing) versus a paper that applies a data-driven approach to examining connectivity changes that allow us to examine certain predictions of 3 out of the 4 models (with a weaker basis to examine predictions related to the 4th). It's much fairer to state that we're doing the latter rather than the former, but it's still the case that we can evaluate predictions of all 4 models that have not thus far been examined to date. Even though the basis for evaluating the timing predictions of the Facilitation (and partly the Predictive Coding model) is weaker, we still nevertheless observed significant effects in these tests that were consistent with the model predictions; we felt that it would be a disservice and unfair not to mention those results for proponents of the Facilitation model. Nevertheless, in the revised manuscript, we now explicitly weight our expectations along the lines discussed above:

pp. 5-6: "While these models are not necessarily mutually exclusive, each makes unique predictions that can be tested. The Synchrony, Predictive Coding, and Sharpening models differ from one another with regard to their predictions about the effect of repetition on connectivity between brain regions. For example, the Synchrony model predicts increased positive inter-regional coupling with repetition. The Predictive Coding model, in contrast, predicts that top-down connections should have stronger negative coupling with repetition and that this coupling should correlate with the magnitude of repetition suppression in the regions receiving this input. The Sharpening model, like the Synchrony model, predicts an increase in positive coupling, but it further requires the sending region to exhibit repetition suppression, accompanied by a reduction in the similarity of neural responses in the sending region due to reduced overlap among neural representations. The Facilitation model is less clear regarding predictions about connectivity but predicts that neural responses in regions exhibiting repetition suppression should peak and terminate earlier for repeated objects, with a similar prediction made by the Predictive Coding model under conditions of high model precision²¹. Finally, each model predicts that behavioral priming magnitude should be related to its core changes, with repetition suppression serving as an indirect marker of the operation of its critical underlying mechanisms. **A strength of BOLD fMRI is its potential for examining changes in connectivity at the scale of the whole brain, as well as its ability to evaluate local changes in representational similarity, germane to predictions of the Synchrony, Predictive Coding, and Sharpening models. The ability of BOLD fMRI to adjudicate subtle changes in the timing of neural activity is less clear, but we nevertheless examine the predicted timing changes of the Facilitation and Predictive Coding models in the event that relevant timing information is detectable. What this means is that it will not be possible to test all four models on equal terms, but the results will have the potential to favor certain models over others, as well as to clarify the relationships between changes in connectivity, repetition suppression, and representational similarity.**"

p.15: "As discussed above, fMRI is not the optimal technique to evaluate this prediction, but we nevertheless examined changes in the timing of the BOLD response in the event that such changes were detectable."

p.19: "In our view, it will take multiple studies across a range of methodologies to fully rule out any of these models, and the primary contribution of the current study is its whole-brain assessment of changes in connectivity and their relationship to repetition suppression and priming magnitude, with more moderate progress on model evaluation."

[.] Thus, the reader is left with two findings that appear to predict priming behavior, at least for a subset of the materials. A left PFC change in the univariate response that correlates with behavioral priming and replicates prior work, and a pattern of functional connectivity between left TPJ and ACC that also correlates with behavioral priming. This is interesting, but clearly insufficient as a basis for making claims about the validity of the abstract models of priming that the authors lay out in the Introduction. From my perspective, it'd be more interesting to know if in fact a strong case can be made that the RS in the left Frontal region and the connectivity between TPJ and ACC really do account for unique variance in the behavioral priming, and if so, is there something about the objects that might explain this distinction?

Perhaps the critical analysis was lost in the rest of the results, but we did conduct a partial correlation analysis, removing the variance shared from the univariate RS effect in L Frontal cortex from the R TPJ->ACC effective connectivity effect, showing that the effective connectivity contribution to priming was indeed partly independent:

pp.13-14: "These results, taken together with the lack of modulation by stimulus repetition of connections into other ROIs exhibiting repetition suppression (e.g. left frontal and left fusiform ROIs), suggest that changes in coupling due to repetition are largely independent of local repetition suppression magnitudes. In order to establish this relationship more clearly for the one connection correlated with behavioral priming magnitude, R TP to ACC, we used partial correlation to remove the magnitude of repetition suppression exhibited in the ACC from the correlation between priming effect size and SEM parameter contrasts in the Strong versus Weak Primeable conditions. This partial correlation remained at approximately the same level as without the partialling [partial $r(55) = 0.334$, $P < .0112$]. We further asked whether the connection from R TP to ACC remained correlated with priming after partialing overall RS observed in the left frontal ROI, which was previously shown to correlate with priming (Figure 3C). The partial correlation again remained virtually unchanged [partial $r(55) = 0.329$, $P < .0125$]. Taken together, these results establish that changes in coupling between R TP and ACC are correlated with priming magnitude in a manner largely independent of the magnitude of repetition suppression, with each accounting for unique portions of the priming variance."

At present, we cannot say definitively which types of objects are most involved in the two different types of effects. The changes in connectivity from R TP->ACC are numerically larger for Strongly Primeable objects (which are objects named more slowly when novel) [$r(56) = .3002$, $P < .0222$, versus $r(56) = .0700$, $P > .6$, for Weakly Primeable objects]. In contrast, RS is numerically more correlated with priming for Weakly Primeable objects [$r(58) = .3432$, $P < .008$, versus $r(58) = .1712$, $P > .19$, for Strongly Primeable objects]. These differences are not statistically significant at these sample sizes. However, they may provide clues that could aid future investigations.

If I understood the outcome of the connectivity findings, it appears that the TPJ-ACC finding shows a positive relationship for strongly primeable items, but also shows a reliably negative relationship for weakly primeable items. I'm having a hard time understanding how the negative relationship fits into the same explanatory framework as the positive. For example, the authors speculate that the positive relationship may play a role in an expanded network that 'functions to retrieve prior stimulus-response associations encoded through cortico-striatal loops.' (page18). How would such a functional contribution demonstrate a negative association for half of the items?

This is a great question. The negative relationship present in the functional connectivity effects for weakly primeable items was unexpected. In the effective connectivity results, the OLD>NEW effects for Strongly primeable items becomes more separated from the NEW>OLD effects for Weakly primeable items (with only the first being significant for the R TP->ACC connection and only the second being significant for the ACC->R Fusiform and R STG->R TP connections). It's not clear at this point exactly what might be behind the reversed effects for Weakly primeable items, but we did wonder whether it might have something to do with mechanisms underlying more explicit memory effects. Recognition memory is not controlled or measured in this experiment; knowing that something is OLD does not help you name it (especially when there are many possible items). There have been some reports showing inverse relationships between priming and recognition memory. For example, Wagner, Maril, and Schacter (2000, *J Cogn Neurosci*, 12 Suppl 2, 52-60) found that words repeated after short lags (within-session) exhibited greater priming than when repeated after long lags (from previous day). However, recognition memory for the words 2 days later was greater for the words previously showing weaker priming. While we did not measure recognition memory performance in the current study, we used the same 200 object pictures in Gilmore et al. (2019, *Neuropsychologia*, 124, 31-43). In that study, we used half of the items to assess recognition memory and half to assess priming, and it allowed us to calculate normative statistics over 40 participants for recognition memory performance. Using a continuous measure of recognition memory strength over all 200 items (RT in recognition memory judgements when an object was OLD versus NEW, averaged across participants, normalized by the quantity OLD+NEW for each item), we do find a negative correlation between this measure and primeability by item (using the norms from the current experiment: $(NEW-OLD)/(NEW+OLD)$) (see Figure R6).

Negative Correlation between Priming and Recognition Memory Strength

Figure R6. A negative correlation is observed between priming strength and recognition memory strength across the current experiment (Priming) and Gilmore et al. (2019) (Recognition Memory) when using the same 200 picture stimuli. Differences between RTs to OLD versus NEW pictures during recognition memory are more pronounced for Weakly Primeable objects.

This is obviously post-hoc speculation at this point. But this is one candidate to explain the reversed effects seen for the Weakly Primeable condition. As mentioned in the Discussion in the original manuscript, activity has been found to be modulated in the R TP region in explicit memory tasks (e.g. Cabeza, Ciaramelli, & Moscovitch, 2012, Trends Cogn Sci 16, 338-52). We think it makes sense that processes involved in recognition memory could contribute simultaneously in these regions, perhaps being superimposed in neural measures of priming effects. We'll need to perform subsequent experiments in which we measure both priming and recognition memory on the same items for the same participants.

Given the speculative and post-hoc nature of this response, we'd prefer to hold off on putting it in print until we have more direct evidence.

It is not clear whether the median split primeability scores (based on solely on novel reaction times) was planned or exploratory.

In the planning phases of the study, we knew that we wanted to have a within-participant measure of priming strength. Previous studies had all used correlations between summary BOLD measures and priming magnitudes across participants, and we wondered whether this might be emphasizing factors such as individual variability in attention or other non-specific

factors that are less related to priming, per se (see also Lebreton et al., 2019, *Nat Hum Behav*, 3, 897-905, for discussion).

We had two basic ideas about how to measure within-participant priming magnitude by item: 1) a continuous measure over items, perhaps based on that participant's own response times inside the scanner, and 2) a discretized measure like that presented in the current paper (in which items are binned by priming strength). For 1), we had the idea of trying to extend a method that we developed in a paper relating structural covariance to behavior in ASD (Eisenberg et al., 2015, *Mol Autism*, Oct 1;6:54); it is a similar measurement problem in that each participant contributes one measure of volume or thickness per voxel/vertex, and one needs to figure out how to examine how covariation of those measures relates to behavior. The method involves correlating individual products of volumes/thicknesses per participant (appropriately normalized) with individual behavioral measures across participants. The idea was to apply this method to individual items within-participant. However, in the end, this method added an even deeper layer of complexity, making it harder to communicate and understand – and individual trial response times turned out to be so noisy (see Supplementary Figure 1) that we abandoned the approach in favor of the 2nd approach, which is what is presented in the paper.

As mentioned in responses to Reviewers 1 and 2, a median split was not chosen for statistical reasons when a more continuous approach was available. It was chosen because it's difficult to estimate "connectivity" when each item only has one measurement per trial (in this case, the peak BOLD response per trial). We could have chosen more than two divisions of Primeability (Strong, Weak), but this would ultimately reduce the number of trials used to estimate connectivity; by 3 divisions (low, medium, and high), we'd be estimating Pearson correlations with ~25 trials rather than ~ 40-45 trials with two divisions. We chose to stick with the simplest version that still permitted within-participant estimates of priming strength (a median split).

We also could have chosen to perform a median split more directly on average priming strength rather than on RTs to novel items (i.e. the difference in the mean RT to an item when repeated versus when novel). The test-retest reliability of average priming strength across Overt and Covert naming participants is $r(198)=0.436$ ($P<2.0\times 10^{-10}$), whereas the same test-retest reliability of average novel RTs is $r(198)=0.704$ ($P<5.0\times 10^{-31}$). The difference in reliability is the main reason that we chose to use novel RT as the basis for Strong/Weak Primeability. One also observes robust correlations of average novel RT and average priming magnitude across Overt/Covert naming groups [average novel RT in Covert Naming participants with average priming magnitude in Overt Naming participants: $r(198) = 0.454$, $P<2.0\times 10^{-11}$; average novel RT in Overt Naming participants with average priming magnitude in Covert Naming participants, calculated using the pre- and post-fMRI naming sessions: $r(198) = 0.508$, $P<2.0\times 10^{-14}$).

The following text passages are intended to clarify the choice of a median split of items regarding Primeability:

p.9: "This fact, combined with the need to pool trials into larger groups anyway in order to calculate connectivity estimates across trials (with only a single datapoint contributed by each trial), led us to adopt a median split of trials into strongly and weakly primed objects (see Online Methods for details)."

p.29: "Employing a more continuous analysis of functional connectivity with priming strength was not possible within-participant, as only a single timepoint (the peak) was retained for each item; functional connectivity estimates therefore had to be calculated

across items, with priming strength incorporated within-participant as a median split of items between Strong and Weak Primeability.”

Although they could not know of it at the time of the current submission, a paper by Davis et al. from the Cabeza lab (Davis et al. in press Cerebral Cortex) may be relevant as it examines visual and semantic similarity of objects and their potential impact on subsequent memory performance. I mention it because it seems like object similarity might also be an important consideration in naming times in the current report. That is, there may be a possibility that naming is slowed as a function of the physical similarity (or conceptual similarity) of say a current object, and objects previously seen with a different name. Their approach was based on scores derived from deep neural networks, discussed reviewed Kriegeskorte and Kievit (2013), however, there may be even simpler methods of scoring similarity than this (e.g., Zelinsky 2003 Psychonomic Bulletin and Review). Regardless, the authors seem to assume that the only thing that is affecting naming times is the number of encounters with the specific exemplar (and its primeability) whereas the relationship of the item with other members of the set may also be important.

We agree with the reviewer that many factors can affect naming RTs. To the extent that different representational overlap in certain collections of objects lead repetition to have a differential effect, then perhaps this is part of the underlying explanation as to why certain items are more strongly primeable than others. However, we would also note that counterbalancing in the current experiment should help to rule out this kind of factor as a design flaw, in the sense that item properties cannot explain group-level effects of NEW/OLD. Items cannot be counterbalanced in terms of being Strongly and Weakly primeable, and we view item properties as the de facto explanation as to why these conditions differ. We suspect that there are numerous additional factors to frequency of exposure, visual and semantic similarity, as well. We have now clarified that we believe multiple factors to be responsible for differences in Primeability and not just number of encounters (citing the Davis et al., Cerebral Cortex, paper):

p.25: “We fully expect item primeability to be a complex function of a variety of factors including frequency of exposure, the level of visual detail and color present in the picture, as well as its “neighborhood” of visual and conceptual relationships with other objects⁶³. The use of response time to NEW pictures here is a practical convenience.”

MINOR.

In the methods it is stated that the ISI during scanning ranged from 6700 to 12700 ms at multiples of the TR. However, the TR of the echoplanar sequence is later listed at 2000 ms.

We apologize for the confusion. The total trial length occurs in multiples of the 2000 ms TR (adding the fixation period, stimulus duration, and ISI). The sentence on p. 22 has been changed to the following:

p.22: “In both tasks, a naming trial consisted of a central black fixation cross presented for 1000 ms, followed by the picture to be named for 300 ms, followed by a blank screen for a period ranging from 6700 to 12700 ms at multiples of the TR (total trial lengths of

8000, 10000, 12000, 14000 ms) and sampled with a uniform distribution (Figure 2; see ⁵⁴ for discussion of optimal ISIs in slow event-related fMRI designs).”

I admit to becoming a little lost on the ‘primeability’ norming procedure as it appeared to mix times gathered across the pre-exposure, overt versus covert scanning response groups, and overt post scan responses of the covert group. In the end the researchers used the alternative of simply the RT during the first novel presentation of each item to produce a median split that they referred to as primeability. However, I thought that all subjects named half of the items aloud three times prior to entering the scanner, regardless of whether they were in the overt or covert groups. If so, then it is unclear why, given list counterbalancing, that these trials alone could not be used to rank order each stimulus in terms of its priming score. This would have the added benefit of being statistically independent of the reaction times actually observed during the later scanning, and it would seem less complicated than the procedure described in the manuscript.

We apologize for the lack of clarity in the norming procedures. After re-checking the norming calculation, the post-fMRI RTs were in fact not used. The average includes the first time named in the pre-fMRI session or the fMRI session for the Overt Naming participants. The description has been simplified on pp. 24-25:

“A within-participant estimation of “primeability” (Strong versus Weak) in object naming was determined from normative analyses of response times across all 60 participants. Response times on correct naming trials were averaged across participants when encountered for the first time as NEW objects (i.e. the first presentation in the pre-fMRI session or the first presentation during the fMRI session), as well as for OLD objects (i.e. presentation during fMRI of objects seen during the pre-fMRI session). Only response times in overt naming sessions were used for this purpose, since button press response times during Covert Naming were systematically faster than for overt naming sessions by as much as 200-300 ms per participant. Each item’s resulting average NEW response time included the correct naming responses from approximately 40 participants.

While each item is named 3 times in pseudorandom order prior to fMRI, only the very first naming instance is used (since this is the only real “novel” instance). Each participant names 100 out of the total 200 items in this way. The remaining 100 items are overtly named as novel when presented during fMRI. When including pre-fMRI sessions from both Overt and Covert naming participants, each item is exposed to ~ 30 participants as Novel (1/2 of the Overt and 1/2 of the Covert naming participants). After factoring in correctness, each item is averaged across ~ 25 participants’ naming responses. If one includes the fMRI naming responses, this number increases to ~ 40 naming responses per item. Our goal was to minimize any later regression to the mean by including as much data in the averaging as possible. One can compare how well the Strong/Weak assignment generalizes to other datasets by examining priming magnitudes in independent data. Using only pre-fMRI naming responses, Strong/Weak Primeability effects are indeed observed in independent data (see Figure R7 below), for both the naming responses in Gilmore et al. (2019, *Neuropsychologia*) and in the fMRI Covert Naming button press times (which were not used in the norming). However, effects are more robust when using the greater amount of data afforded by the fMRI naming responses (Original).

Figure R7. Selecting Strong/Weak priming from behavioral data in the current experiment replicates in independent data (left panel: data from Gilmore et al., 2019; right panel: Covert Naming button press times during fMRI in the current experiment). Strong>Weak priming effects are significant when selecting based on approximately 40 naming responses per item (Original), and even if only based on approximately 25 naming responses per item (pre-fMRI data only). However, these effects are stronger, with less regression to the mean, when selecting based on more data – presumably because of greater noise cancellation (Original).

For the independence issue that the reviewer mentions (also raised by Reviewer 1), we have conducted a series of control analyses in which Strong/Weak calculations for a given participant eliminated each participant's own fMRI naming responses from the average, still allowing the fMRI naming responses from all other participants. This leads to individualized selection of Strong/Weak primeability for each participant. After recalculating all of the main results in the paper (shown in new Supplementary Figures 5-7), we can confidently state that the results are not explained by this issue.

Reviewers' comments:

Reviewer #1 (Remarks to the Author):

I am happy with the authors responses to my and the other reviewers' comments. I have one remaining request, again to do with the predictions of the predictive coding account of Friston. I still think that, while the general prediction of negative backward connectivity for primed stimuli is correct, testing this is not so simple with fMRI. The main reason is that given in the Ewbank & Henson (2012) paper the authors mention in their reply: the full predictive coding model of Friston includes different neuronal populations in different cortical layers, and while backward predictions might reduce prediction errors in superficial cortical layers, they could increase activity in the "state" neurons in deeper layers, such that when BOLD averages over all layers (in the absence of laminar-resolution), there may no longer be a negative correlation between BOLD signal in those two layers (see also Egner et al, 2010, J Neuro). This is where EEG/MEG might be able to help, on the assumption that they chiefly detect synchronous activity in the superficial neurons. Anyway, while I appreciate this sounds like a "get out clause", I think it would be fair for the authors to at least admit that there are other opinions (to this critical prediction of their paper), by modifying this sentence:

"The Predictive Coding model, in contrast, WOULD SEEM TO predict that top-down connections should have stronger negative coupling with repetition, and that this coupling should correlate with the magnitude of repetition suppression in the regions receiving this input (THOUGH SEE EWBANK & HENSON, 2012)."

I'd also like to note that I think it might be possible to examine a continuous (eg linear) effect of primeability on connectivity. Imagine one puts the peak responses for 100 trials into a vector y for one ROI, into vector x for a second ROI, and also has a vector p of 100 primeability scores. One could split x into two vectors according to whether p is above or below the median, put these into a GLM design matrix X , and evaluate a $[1 \ -1]$ contrast to test whether the relationship between x and y differs by whether p is above or below the median. This is equivalent to running a $[1 \ 0]$ contrast on a GLM in which the first regressor is the original x values multiplied by 1 or -1 (depending on whether p above median) and a second constant regressor of 1s (ie just a rotation of the original design matrix). From this, one could imagine replacing the first vector with the element-wise-product of the original peak values in x with a mean-corrected version of the continuous vector p , and testing the same $[1 \ 0]$ contrast. I think this would test whether the dependency between y and x (connectivity) depended on primeability. I am not suggesting the authors do this – for their factorial analysis and desire for symmetric connectivity, the median split is easier – and it is possible I am missing something about the design – but just in case of interest.

Also, for the record, I think the authors overstate the robustness of their functional connectivity using two TRs around the peak of responses in a slow event-related design. I agree that a long SOA design minimises assumptions of linearity necessary for deconvolving short SOA designs, but I think the use of two TRs to summarise a trial's response is also sensitive to HRF shape to some extent, and in fact may have lower SNR than the more typical parameter estimate of single gamma, two-gamma or independently-defined HRF (as in the original "Beta-Series Regression" of Rissman et al 2004, Neuroimage). This is because the author's approach can also be thought of as a weighting function across TRs that is 0 except for the two peak TRs where it is 1, whereas an HRF can be thought as a more continuous (smooth) weighting across multiple TRs, but where those values are small either size of the peak (eg gamma function). If the true HRF has a different delay in the two regions, that could still affect the mean activity over the two TRs. This could also affect connectivity, assuming that delay also has some random element across trials, in the same way that such a delay would affect the parameter estimate from a model HRF. Moreover, even if a model HRF is slightly different from the true HRF, it is likely to have a higher SNR, by virtue of being closer to the true HRF shape than a "box-car" function of two TRs around the peak. But I'm happy to be proved wrong!

Reviewer #2 (Remarks to the Author):

The authors have done a very commendable job of responding to the reviewers previous comments. While the responses are thoughtful and professional, this reviewer didn't come away with a clear sense of a strengthened manuscript. One large issue remains from before - the decision to split items by "primeability". This approach equates magnitude of the repetition effect with the construct of interest and by the authors own review of the literature, that is a problematic approach. As pointed out - repetition priming can reflect a mix of processes - perceptual facilitation and novelty/episodic memory. Also as pointed out in the authors review of previous literature, magnitude of priming likely reflects this.

A second issue, minor I guess, was the response to reviewer 1 pointing out that DCM may be the better approach to examining connectivity. The authors response that, yes perhaps but we couldn't figure out the software is pretty unsatisfying. Perhaps since reviewer 1 is named, they could work together on this issue.

Reviewer #3 (Remarks to the Author):

This is the second review of 'Enhanced inter-regional coupling of neural responses and repetition suppression provide separate contributions to long-term behavioral priming' by Gotts, Mileville, and Martin. The authors were responsive to the previous concerns and the various analyses are now clearer. That said, there is quite a lot being attempted within this short format, and indeed, the description of the methods and statistical procedures is almost as long as the paper itself. Moreover, given the authors responses and my re-reading, it does appear that the primeability scoring and connectivity analyses are best described as exploratory. Thus while I think the finding that connectivity between temporoparietal and striatal regions explains unique variance in individual differences in behavioral priming compared to univariate findings is important, it would seem wise to find an independent confirmation of this given the number of choice points in the analysis that uncovered the finding. Moreover, I remain skeptical that the data are capable of adjudicating among the various models of priming discussed, primarily because these models are insufficiently detailed as to be falsifiable by these data in isolation. For example, as I note below, while the synchrony model anticipates that connectivity findings (somewhere) will be linked to priming behavior, it says nothing about univariate findings and widespread repetition suppression. I outline my remaining concerns below, followed by a brief summary paragraph.

I continue to feel that the paper implies that the broad/coarse models it discusses are sufficiently specified to be meaningfully adjudicated by the fMRI data and analysis. However, to my mind the novel finding, that interregional connectivity is systematically correlated to behavioral priming scores, while important cannot possibly weigh against models specified at this level of detail. For example, the authors contend that the Synchrony model 'holds that as neural activity decreases, cells become more synchronized in their firing, leading to a larger impact on downstream targets and earlier, more reliable propagation of individual spikes, supporting earlier motor responses.' But I am unclear which 'cells' this model is referring to in the sense that it is not clear that model is linked to pre-specified perceptual, associative, or motor related regions. If I read between the lines it seems as though the authors are suggesting that changes in connectivity are linked to aspects of response/decision selection or execution and hence are not perceptual in nature. This would be a neat idea and potentially fractionate priming behavior into different motor (connectivity linked) and representational (univariate) components captured by different aspects of the BOLD response. However, at present the connectivity model simply suggests some regions, somewhere in the brain, will show increased functional connectivity when behavioral priming is present. Without tying this to particular processes, with clearer boundary conditions, the model is arguably just a re-description of the findings. Related to this potential lack of specificity, the Figure 1 caption of

the Synchrony model illustrates 'Local Field Activity' which does not seem specific to predicting an increase in effective connectivity during fMRI for unspecified, but distant cortical regions.

On page 9 the authors indicate that their findings using primeability scores 'suggest that a within-participant measure of priming may indeed add sensitivity, as a significant across-participant correlation with priming was only observed in the left frontal region (compare Figure 5A to 5B).' However, this assertion seems odd. First, the two halves of the figure are not testing the same relationships and so differences in the p-values are even less meaningful than usual. Moreover, it seems hardly surprising (even if regression to the mean is ruled out) to find that items that yield weaker behavioral priming effects yield weaker repetition suppression effects. Thus it is not clear that Figure 5b provides any information not already clear in Figure 5a, particularly if one doesn't treat .05 as a categorical bright line. Thus while it may be good standard practice to eliminate items that perform poorly (by some independent measure) when considering brain behavior relationships, the authors could have presumably done the same thing by removing low primeability items before conducting the analysis in Figure 5A.

I didn't quite understand the rationale for carrying forward all the regions examined in the functional connectivity analysis to the effective connectivity analysis. The authors indicate that SEM carries more information than basic functional connectivity, but keeping the regions that failed to show functional connectivity effects and then conducting a completely exploratory SEM would seem to run a high risk of overfitting.

In summary, the authors provide initial evidence that connectivity and repetition suppression BOLD effects may make separable contributions to behavioral priming. This is quite interesting, but the connectivity findings seem fairly exploratory and so this possibility is fairly tentative. Moreover, the notion that the data weigh heavily in favor of any of the coarse models discussed seems strained to me. While I understand that some of the discussed models do not anticipate connectivity findings in regions that do not show repetition suppression, the synchrony model does not explain why repetition suppression, which is the most robust effect present in priming studies, occurs at all. Finally, there is the broader issue that it is becoming increasingly common for authors to report both significant multivariate and univariate findings in fMRI data, and I have little doubt that if older priming data sets relying strictly upon univariate analysis were re-analyzed, there would often be reliable multivariate effects as well. In this light, finding both effects in any given data set would not seem particularly strong evidence for or against any specific functional theory unless for some reason that theory forbade one class of analysis (univariate vs. multivariate) from rendering significant effects. Thus, while I found the connectivity findings quite interesting and a good target for future research, they don't strike me as dispositive for any particular model of priming. Finally, while the primeability scoring seems like a good methodological practice if it can be conducted independently, the main contribution is equivalent to recommending that researchers conduct item analyses in an MLM framework or otherwise control for the fact that some items simply perform poorly on tests.

** See the Nature Portfolio author and referees' website at www.nature.com/authors for information about policies, services and author benefits

Communications Biology is committed to improving transparency in authorship. As part of our efforts in this direction, we are now requesting that all authors identified as 'corresponding author' create and link their Open Researcher and Contributor Identifier (ORCID) with their account on the Manuscript Tracking System prior to acceptance. ORCID helps the scientific community achieve unambiguous attribution of all scholarly contributions. You can create and link your ORCID from the home page of the Manuscript Tracking System by clicking on 'Modify my Springer Nature account' and following the instructions in the link below. Please also inform all co-authors that they can add their ORCIDs to their accounts and that they must do so prior to acceptance.
<https://www.springernature.com/gp/researchers/orcid/orcid-for-nature-research>

If you experience problems in linking your ORCID, please contact the Platform Support Helpdesk.

February 12, 2021

We would like to thank the reviewers again for their detailed and helpful comments on our revised manuscript entitled "Enhanced inter-regional coupling of neural responses and repetition suppression provide separate contributions to long-term behavioral priming". We have provided point-by-point responses below to their remaining concerns, and we have made corresponding changes to the manuscript. Major changes include the addition of a Limitations section to the Discussion, as well as a brief Conclusions section.

We hope that with these additional changes the manuscript will now be acceptable for publication in *Communications Biology*.

Sincerely,
Stephen J. Gotts, PhD & Alex Martin, PhD

Reviewers' comments:

Reviewer #1 (Remarks to the Author):

I am happy with the authors responses to my and the other reviewers' comments. I have one remaining request, again to do with the predictions of the predictive coding account of Friston. I still think that, while the general prediction of negative backward connectivity for primed stimuli is correct, testing this is not so simple with fMRI. The main reason is that given in the Ewbank & Henson (2012) paper the authors mention in their reply: the full predictive coding model of Friston includes different neuronal populations in different cortical layers, and while backward predictions might reduce prediction errors in superficial cortical layers, they could increase activity in the "state" neurons in deeper layers, such that when BOLD averages over all layers (in the absence of laminar-resolution), there may no longer be a negative correlation between BOLD signal in those two layers (see also Egner et al, 2010, J Neuro). This is where EEG/MEG might be able to help, on the assumption that they chiefly detect synchronous activity in the superficial neurons. Anyway, while I appreciate this sounds like a "get out clause", I think it would be fair for the authors to at least admit that there are other opinions (to this critical prediction of their paper), by modifying this sentence:

"The Predictive Coding model, in contrast, WOULD SEEM TO predict that top-down connections should have stronger negative coupling with repetition, and that this coupling should correlate with the magnitude of repetition suppression in the regions receiving this input (THOUGH SEE EW BANK & HENSON, 2012)."

This is a reasonable request, and we are happy to edit our text in this way (included on p.5 in the main text).

While, in general, the main job of effective connectivity is to unmix what are otherwise mixed and superimposed signals from underlying sources, we agree that it may not accomplish this perfectly. Recent experiments in our lab using VASO and layer fMRI (e.g. Persichetti et al., 2019 in *Current Biology*, doi:10.2139/ssrn.3482808) suggest that this technique may be able to accomplish the needed "unmixing" through measurement rather than solely through effective

connectivity estimation. In that context, we should be able to revisit this issue. We have further included this point in the new **Limitations of the Current Study** section on pp. 20-21.

pp. 20-21: “Future work should examine alternatives that employ different underlying statistical approaches using the full fMRI timeseries (e.g. Dynamic Causal Modeling⁵⁵), as well as different measurement strategies that can potentially isolate feedforward/feedback cortical signals, such as layer-specific fMRI (see ⁵⁶ for one recent example).”

I'd also like to note that I think it might be possible to examine a continuous (eg linear) effect of primeability on connectivity. Imagine one puts the peak responses for 100 trials into a vector y for one ROI, into vector x for a second ROI, and also has a vector p of 100 primeability scores. One could split x into two vectors according to whether p is above or below the median, put these into a GLM design matrix X , and evaluate a $[1 -1]$ contrast to test whether the relationship between x and y differs by whether p is above or below the median. This is equivalent to running a $[1 0]$ contrast on a GLM in which the first regressor is the original x values multiplied by 1 or -1 (depending on whether p above median) and a second constant regressor of 1s (ie just a rotation of the original design matrix). From this, one could imagine replacing the first vector with the element-wise-product of the original peak values in x with a mean-corrected version of the continuous vector p , and testing the same $[1 0]$ contrast. I think this would test whether the dependency between y and x (connectivity) depended on primeability. I am not suggesting the authors do this – for their factorial analysis and desire for symmetric connectivity, the median split is easier – and it is possible I am missing something about the design – but just in case of interest.

We agree that something like this might accomplish a continuous form of the analysis. Indeed, we proposed something like this previously to associate covariation in cortical thickness/volume measures with behavior, where the problem of introducing connectivity is similar (one measure per ROI per person, using a product of standardized thickness/volume to stand in for a full correlation) (Eisenberg et al., 2015, in *Molecular Autism*; doi://10.1186/s13229-015-0047-7). In the current context, with a data-driven search for changes in connectivity, it would somehow have to be integrated within the concept of “connectedness” (perhaps by calculating the trial-level products of standardized activity with each voxel and primeability value, then averaging the products back into each corresponding “seed” voxel), and it would have the additional problem of being even more difficult to explain than what we've currently done. However, we agree that something like this might be possible. It is probably more straightforward for an ROI-level analysis, where the ROIs are already known/selected.

Also, for the record, I think the authors overstate the robustness of their functional connectivity using two TRs around the peak of responses in a slow event-related design. I agree that a long SOA design minimises assumptions of linearity necessary for deconvolving short SOA designs, but I think the use of two TRs to summarise a trial's response is also sensitive to HRF shape to some extent, and in fact may have lower SNR than the more typical parameter estimate of single gamma, two-gamma or independently-defined HRF (as in the original “Beta-Series Regression” of Rissman et al 2004, *Neuroimage*). This is because the author's approach can also be thought of as a weighting function across TRs that is 0 except for the two peak TRs where it is 1,

whereas an HRF can be thought as a more continuous (smooth) weighting across multiple TRs, but where those values are small either size of the peak (eg gamma function). If the true HRF has a different delay in the two regions, that could still affect the mean activity over the two TRs. This could also affect connectivity, assuming that delay also has some random element across trials, in the same way that such a delay would affect the parameter estimate from a model HRF. Moreover, even if a model HRF is slightly different from the true HRF, it is likely to have a higher SNR, by virtue of being closer to the true HRF shape than a “box-car” function of two TRs around the peak. But I’m happy to be proved wrong!

It is important to point out that there was no further model fitting after the GLM and prior to the functional connectivity analyses (i.e. there was no fitting of a box-car function with two 1's at the peak and 0's elsewhere). The functional connectivity analyses were conducted on the residuals of the nuisance regression+GLM. And the GLM had a separate regressor for each TR of each condition type (the “TENT” function: 1 at the TR of interest and 0's elsewhere), so there is no obvious opportunity for a tradeoff in the estimate of the peak and the non-peak values. Our measure here was to simply average the residuals in the period of 4-8 seconds on each correct trial. The approach is intended to guard against aliasing the shared temporal contour of the task responses (off-then-on-then-off) into the connectivity measure. Here, the amplitudes of the responses at the peak of each trial have to covary in order to have a non-zero functional connectivity value. Indeed, if there were further model fitting (either a standard HRF or a box-car function), then while it may help SNR (allowing noise in the peak and non-peak TRs to effectively cancel), it could re-introduce an element of the stimulus contour.

We agree, though, that this is not to say that the HRF shape has zero contribution to the estimates of the peak and, possibly, to the estimates of connectivity. Choosing the wrong peak period (e.g. at half height) would be expected to affect the estimates of the peak value – as well as the connectivity estimates if the SNR is different than at the correct peak (lower SNR would lead to lower connectivity estimates). It's simply to say that it is minimizing one known artifactual contribution to the connectivity measures (the gross temporal contour of the stimulus response, which given a non-zero stimulus response will definitely otherwise contribute). We did confirm that for the ROIs involved in the main connectivity analyses, the empirically determined peak times (mean across trials) did fall in the 4-8 second window for every subject. Obviously, if the trial-wise peaks can vary dramatically in their temporal onset, this will also affect estimates of the peak and connectivity. However, this would be true if one were to fit a single HRF to each trial's response, as well. Our practice of eliminating trials with unusually long RTs (> 2000 ms) should hopefully minimize one foreseeable source of this type of phenomenon (see Methods section on *Behavioral Priming Analyses* on pp. 28-29).

In response, though, we have softened the language used to describe this approach, replacing words like “eliminate” and “avoid” with “minimize”:

p.6: “Given the importance of between-region connectivity to the predictions being tested, a novel form of fMRI task-based connectivity was designed to **minimize** contamination by the temporal contour of the task-evoked response to functional and effective connectivity measures.”

p.31: “In order to **minimize** contamination of task-based functional connectivity estimates from the temporal contour of the evoked responses of individual trials, only the average peak BOLD response was retained from each individual trial (the raw average of the 3rd and 4th timepoints of the task residual timeseries following the onset of each stimulus).”

p.39 (caption of Figure 2): “For connectivity analyses, the peak BOLD response was calculated for each trial and in each fMRI voxel by averaging the 2 timepoints (4-6 and 6-8 seconds) adjacent to the expected peak of the hemodynamic response function, with a single peak value saved for each trial in order to minimize the contribution of the temporal contour of the evoked responses from functional and effective connectivity estimates.”

Reviewer #2 (Remarks to the Author):

The authors have done a very commendable job of responding to the reviewers previous comments. While the responses are thoughtful and professional, this reviewer didn't come away with a clear sense of a strengthened manuscript. Once large issue remains from before - the decision to split items by "primeability". This approach equates magnitude of the repetition effect with the construct of interest and by the authors own review of the literature, that is a problematic approach. As pointed out - repetition priming can reflect a mix of processes - perceptual facilitation and novelty/episodic memory. Also as pointed out in the authors review of previous literature, magnitude of priming likely reflects this.

We appreciate the reviewer's point that it is intrinsically difficult to disentangle priming from more explicit forms of memory. However, that same issue is no less applicable to a continuous measure (i.e. it is not the median split by primeability, per se, that is really at issue). It is not obvious to us that the behavioral measure of priming itself is strongly collinear with the operation of explicit memory processes. For example, remembering that a picture of a tiger was presented previously does not itself help to produce the label “tiger” as opposed to “lion” or “car” when both of those stimuli were also previously presented (each participant has named 100 pictures previously, so matching to previous responses in working memory is infeasible). In contrast, the brain activity that is measured simultaneously will likely reflect the operation of both implicit and explicit memory processes.

For the current stimulus set (200 items), we have estimates of recognition memory performance from a separate group of participants by item (N=40 in Gilmore et al., 2019, *Neuropsychologia*). Below is the correlation matrix by item (198 dfs) of two recognition memory measures (corrected hit rate: hits – false alarms; NEW-OLD RTs normalized by NEW) with priming magnitude (also normalized by NEW) and average naming RT:

	Priming (NEW-OLD)/NEW RT	Naming RT
Corrected Hit Rate (H-FA)	0.081892676	-0.0927011
(NEW-OLD)/NEW RT	-0.140953017	0.00874853

As you can see, the correlations between behavioral measures of recognition memory strength and the measure we used to split primeability (average naming RT) are not strong ($r=-.093$ for corrected hit rate, $P<.2$; $r=.0087$ for an RT-based measure, $P<.91$), sharing less than 1% of the variance. Therefore, while recognition memory processes may explain some of the variation in the neural measures in the current experiment, the lack of a strong correlation with the critical behavioral measures used here (priming magnitude and naming RT) suggest that the particular

brain-behavior correlations reported here are unlikely to be strongly undermined by the presence of explicit memory processes.

Nevertheless, we think that it will be important for future studies to jointly assess contributions of explicit memory processes to these phenomena, perhaps by employing both implicit and explicit memory tasks in the same participants, permitting the partialling of one type of measure from the other by participant. We have mentioned this issue as an important limitation of the current study on p. 21 (**Limitations of the Current Study** section).

p. 21: "In the current study, repetition of stimuli was implicit to the task being performed. However, participants were likely aware that repeated stimuli had been presented previously, and markers of explicit recollection could be present, particularly in the neural repetition effects. Future studies should extend the current work by measuring both priming and measures of explicit memory (e.g. recognition memory) simultaneously in the same participants, which may permit the partialling and/or isolation of the contributions of explicit recollection."

We also have reported the item-level correlations of naming RT, priming magnitude, and recognition memory strength from the Gilmore et al. (2019) study in Supplementary Table 2.

A second issue, minor I guess, was the response to reviewer 1 pointing out that DCM may be the better approach to examining connectivity. The authors response that, yes perhaps but we couldn't figure out the software is pretty unsatisfying. Perhaps since reviewer 1 is named, they could work together on this issue.

Rik Henson wasn't exaggerating that DCM is complicated to implement and interpret. It requires a level of expertise in SPM that we know we don't possess (and would take real time to gain under skilled supervision). We do plan to ask for his help with this issue, and we've also mentioned the need for further examination of effective connectivity approaches in the Limitations section on pp. 20-21.

Reviewer #3 (Remarks to the Author):

This is the second review of 'Enhanced inter-regional coupling of neural responses and repetition suppression provide separate contributions to long-term behavioral priming' by Gotts, Mileville, and Martin. The authors were responsive to the previous concerns and the various analyses are now clearer. That said, there is quite a lot being attempted within this short format, and indeed, the description of the methods and statistical procedures is almost as long as the paper itself. Moreover, given the authors responses and my re-reading, it does appear that the primeability scoring and connectivity analyses are best described as exploratory. Thus while I think the finding that connectivity between temporoparietal and striatal regions explains unique variance in individual differences in behavioral priming compared to univariate findings is important, it would seem wise to find an independent confirmation of this given the number of choice points in the analysis that uncovered the finding. Moreover, I remain skeptical that the data are capable of adjudicating among the various models of priming discussed, primarily because these models are insufficiently detailed as to be falsifiable by these data in isolation. For example, as I note below, while the synchrony model anticipates that

connectivity findings (somewhere) will be linked to priming behavior, it says nothing about univariate findings and widespread repetition suppression. I outline my remaining concerns below, followed by a brief summary paragraph.

We thank the reviewer again for his/her thorough and insightful comments and criticisms. We agree with the reviewer that our data-driven analyses are inherently exploratory, and we are happy to clarify and highlight this as a limitation. We do not agree with the reviewer that the theories are not sufficiently detailed to make contact with the current data. We have made specific predictions and tested them, with some directly contradictory findings for the Predictive Coding, Sharpening and Synchrony models, and we have highlighted the need for revision in all of these models. Even the null findings can be useful to the field, as eventually across many studies and many different methods, if a theory says that something should happen and it continues to fail to be observed, one's confidence and interest in that theory will progressively wane. That said, we agree that these points could be better emphasized, and we have added a more forceful paragraph along these lines to the new Limitations section in the Discussion, highlighting the failures of the Synchrony model more specifically (pp. 21-22). We have also softened the language used to describe the agreement of the results with the Synchrony model throughout the text (in the Abstract, Highlights, Results, and Discussion), replacing language such as "provides critical support for" with "is most consistent with". We respond in more detail to each point below.

I continue to feel that the paper implies that the broad/coarse models it discusses are sufficiently specified to be meaningfully adjudicated by the fMRI data and analysis. However, to my mind the novel finding, that interregional connectivity is systematically correlated to behavioral priming scores, while important cannot possibly weigh against models specified at this level of detail. For example, the authors contend that the Synchrony model 'holds that as neural activity decreases, cells become more synchronized in their firing, leading to a larger impact on downstream targets and earlier, more reliable propagation of individual spikes, supporting earlier motor responses.' But I am unclear which 'cells' this model is referring to in the sense that it is not clear that model is linked to pre-specified perceptual, associative, or motor related regions. [..]

The reviewer is suggesting that the models discussed in the current paper are not specified in sufficient detail to make specific predictions. However, a lot has been written and tested about these models already at a variety of different levels (ranging from fMRI to MEG to ECoG and single-cell recording studies in monkeys), and we have referred to more detailed reviews that provide a great deal more context than is possible to provide in an article of this length (see for example, Grill-Spector et al., 2006, TICS; Gotts et al., 2012, Cogn Neurosci; Gotts, 2016, Psychon Bull Rev). We have then simply stated predictions derived from the models (and articulated previously) rather than elaborating on all of the previous evidence collected that led to them. Perhaps this is contributing to the reviewer's unease with the current articulation of the models and their predictions.

However, the models at their current level of detail do make falsifiable predictions – and in this specific experimental context (and the number of other previous studies associated with these models further attest to that). As previously stated, the Predictive Coding model places the cause of repetition suppression as increases in top-down negative connectivity. The Sharpening model places the cause of repetition suppression as decreased overlap of the underlying neural representations (leading them to become more stimulus-specific), which in turn is taken to cause an increase in connectivity (positive) with downstream regions. We

found: 1) positive rather than negative connectivity in all connections, 2) no relationship between top-down connectivity and the magnitude of RS, and 3) no increase in connectivity from a region showing RS (nor a relationship to “sharpening” measured in the MVPA analyses). The Synchrony model predicts mainly that repetition should lead to increased coupling among task-engaged regions and that this coupling should correlate with the magnitude of priming. Because Predictive Coding and Sharpening have combined RS and connectivity as reflecting operations of the same underlying mechanisms, they do not allow for the orthogonal separation of these phenomena. Therefore, as currently articulated, they have no way to explain the connectivity finding, whereas the Synchrony model can. However, the Synchrony model requires revision, too, since it has no way to explain a correlation between RS and priming without corresponding connectivity changes. To our minds, this shows that the novel findings do indeed weigh against the models specified at the current level of detail. How otherwise could revisions be required? Nevertheless, we have added a new paragraph in the **Limitations of the Current Study** section detailing these points with a particular emphasis on the limits of the Synchrony model:

pp. 21-22: “Finally, the four theoretical models themselves are specified in somewhat general terms that make joint, head-to-head testing difficult. For example, they do not make direct claims about the involvement and role of particular brain regions in priming, but rather apply generally to task-engaged regions and particularly those that exhibit repetition suppression. They also do not necessarily have equivalent levels of complexity or address the same exact family of possible experimental observations. As previous proponents of the Synchrony model¹⁶, we feel that it’s important to disclose that while the connectivity pattern observed between the right temporoparietal and ACC regions fits with the general expectation of this model, the separation of repetition suppression from connectivity in the current experiment is also problematic. The Synchrony model (along with the Predictive Coding and Sharpening models) has no clear way to explain the correlation between repetition suppression and priming that is not mediated by connectivity changes. Furthermore, we cannot measure spiking activity directly with fMRI, so even the aspects of the current results that provide supporting evidence are necessarily indirect. The Synchrony model, at least as previously articulated¹⁶, is no less guilty of assuming that connectivity changes would be observed among regions showing repetition suppression, although it left open the possibility that regions not showing repetition suppression could also contribute. These failures will require further examination of the underlying mechanistic bases of repetition suppression using methods with higher temporal resolution (e.g. ECoG, simultaneous fMRI/EEG), and they will require rethinking of all of the models, the Synchrony model included.”

[..] If I read between the lines it seems as though the authors are suggesting that changes in connectivity are linked to aspects of response/decision selection or execution and hence are not perceptual in nature. This would be a neat idea and potentially fractionate priming behavior into different motor (connectivity linked) and representational (univariate) components captured by different aspects of the BOLD response. However, at present the connectivity model simply suggests some regions, somewhere in the brain, will show increased functional connectivity when behavioral priming is present. Without tying this to particular processes, with clearer boundary conditions, the model is arguably just a re-description of the findings. Related to this potential lack of specificity, the Figure 1 caption of the Synchrony model illustrates

'Local Field Activity' which does not seem specific to predicting an increase in effective connectivity during fMRI for unspecified, but distant cortical regions.

We do not think that predictions have to be specific to certain brain regions in order to be interesting or useful. As an example, Sharpening has been proposed to apply in explaining RS effects throughout the brain, with the content of what cells represent changing from region to region (e.g. sharpening among decision representations in lateral frontal cortex versus sharpening among visual object representations in the fusiform gyrus; discussed in Gotts, 2016, *Psychon Bull Rev*). Similarly, Friston and colleagues have argued for Predictive Coding as a general rubric for the function of neocortical networks throughout the brain. One could say the same thing about the Synchrony model and the Facilitation model. These are models intending to specify mechanistic detail about *how* processing occurs rather than *where*, with a general assumption that similar cortical computations can apply throughout neocortex. They are similar to the Spreading Activation model and Connectionist Networks in Cognitive Psychology or the Biased Competition model of Desimone/Duncan in Neuroscience in this way. This is not to say that different "processes" like perception, conceptual processing, and response/decision selection don't exist. It is to say that they can be represented with the same set of underlying mechanistic parts. One could, of course, argue that completely different underlying mechanisms apply in different cortical regions. But that is not the default position of these models. The Predictive Coding, Sharpening, and Facilitation models are posited to occur in (and among) locations where RS is observed, which ends up being quite a lot of cortex.

Empirically, it would be possible to probe whether the current novel findings are limited to one portion of the sensory-to-motor transformation by manipulating different aspects of the stimuli and the tasks participants are performing (for example, see studies by Dobbins et al., 2004, *Nature*; Horner & Henson, 2008, *Neuropsychologia*; Race et al., 2009, *J Cogn Neurosci*). In a naming task, though, there is a limit on the number of items that one could assign to a large number of factorial conditions (in developing the current stimulus set, we found that only around 250 objects are high enough in name frequency and agreement to be effectively useable without lots of non-responses and lost trials). It's a good idea for future studies that could employ alternative forced choice tasks, though (for which the stimulus set could be much larger). Previous studies that have performed such manipulations have found that RS can be eliminated in certain regions under certain circumstances, but this has still left the *how* questions unanswered in regions continuing to show RS (with all 4 models potentially still explaining RS/priming there).

On page 9 the authors indicate that their findings using primeability scores 'suggest that a within-participant measure of priming may indeed add sensitivity, as a significant across-participant correlation with priming was only observed in the left frontal region (compare Figure 5A to 5B).' However, this assertion seems odd. First, the two halves of the figure are not testing the same relationships and so differences in the p-values are even less meaningful than usual. Moreover, it seems hardly surprising (even if regression to the mean is ruled out) to find that items that yield weaker behavioral priming effects yield weaker repetition suppression effects. Thus it is not clear that Figure 5b provides any information not already clear in Figure 5a, particularly if one doesn't treat .05 as a categorical bright line. Thus while it may be good standard practice to eliminate items that perform poorly (by some independent measure) when considering brain behavior relationships, the authors could have presumably done the same thing by

removing low primeability items before conducting the analysis in Figure 5A.

Given the many instances of non-significant correlations between RS and priming in the literature (and in several of our previous studies), we did not take the within-participant demonstration of greater RS for greater primeability as a given. Interestingly, if one does what the reviewer asks (including only strongly primeable items and performing a correlation of RS and priming across participants), one fails to find any significant correlations between RS and priming in any of the 4 RS regions [the largest correlation is for the R Fusiform: $r(58) = 0.253$, $P < .06$, uncorrected, a non-significant trend; the result for L Frontal is $r(58) = 0.171$, $P < .2$, uncorrected]. At least in the case of the L Frontal ROI, the across-participant relationship appears to be more driven by the weakly primeable items: $r(58) = 0.343$, $P < .008$ (uncorrected). We think that this highlights the need to look at these data in multiple ways, and indeed, the across-participant and within-participant results may differ substantively. We would be more confident that the traditional across-participant results aren't contaminated by effects of less interest (e.g. how distractable some participants might be) if the results were to cohere with the scenario suggested by the reviewer. For these reasons, we'd like to continue to include both Figures 5A and 5B. However, we agree with the reviewer that comparing p-values in *r*-tests versus *t*-tests is at best inconclusive, and we have removed the language about added sensitivity on pp.8-9:

p.9: "These results provide evidence that RS and priming magnitude are related in regions outside of left frontal cortex, including the left fusiform gyrus and the ACC (Figures 5A and 5B)."

I didn't quite understand the rationale for carrying forward all the regions examined in the functional connectivity analysis to the effective connectivity analysis. The authors indicate that SEM carries more information than basic functional connectivity, but keeping the regions that failed to show functional connectivity effects and then conducting a completely exploratory SEM would seem to run a high risk of overfitting.

We apologize for the lack of clarity on this point. In general, it is important in an effective connectivity model to include all regions that are strongly modulated by a task and that are expected to interact, as leaving out one important region can affect the connectivity estimates among all other regions (for discussion, see Smith et al., 2012, *Front Syst Neurosci*; Roebroeck et al., 2011, *NeuroImage*). In this particular case, it was also important to include the RS regions so that parameter estimates could be derived for the testing of the Predictive Coding and Sharpening models (without including the RS regions, this would not be possible). The reviewer is correct that the more complex a model becomes, the more opportunity there is for overfitting noise. This is the reason that the Akaike Information Criterion (AIC) is typically used during the model search process; it is a measure of out-of-sample prediction error. As a model starts small and gains more parameters, it fits the in-sample data better and the AIC measure is simultaneously reduced (the model search process involves training on part of the data and reserving some of the data to assess the degree of model generalization). However, as more and more parameters are added to improve the fit, the AIC eventually starts to increase again as overfitting occurs. This is the reason that the optimal model chosen is the one that significantly fits the in-sample data but that also minimizes the AIC (which is the process that we have followed here). We have now mentioned the role of the AIC in addressing overfitting on p.34.

The reviewer is also correct that the effective connectivity modeling was exploratory. In response, we have highlighted the exploratory nature of this analysis in the Limitations section on pp. 20-21 (also included to address comments of Reviewer 2):

pp. 20-21: “As an additional example, the effective connectivity analyses in the current study utilized exploratory Structural Equation Modeling, in which a search was conducted for the optimal model that explained the data while minimizing out-of-sample error. We made efforts to include the most relevant brain regions in the current task, pooling both regions exhibiting repetition suppression and those identified as exhibiting any priming-related connectivity changes. However, there is a risk that effective connectivity relationships can be model- and method-dependent⁵⁴, and it is inherently challenging for a statistical model to induce the correct underlying feedforward and feedback influences among highly interconnected cortical regions. Future work should examine alternatives that employ different underlying statistical approaches using the full fMRI timeseries (e.g. Dynamic Causal Modeling⁵⁵), as well as different measurement strategies that can potentially isolate feedforward/feedback cortical signals, such as layer-specific fMRI (see ⁵⁶ for one recent example).”

In summary, the authors provide initial evidence that connectivity and repetition suppression BOLD effects may make separable contributions to behavioral priming. This is quite interesting, but the connectivity findings seem fairly exploratory and so this possibility is fairly tentative. Moreover, the notion that the data weigh heavily in favor of any of the coarse models discussed seems strained to me. While I understand that some of the discussed models do not anticipate connectivity findings in regions that do not show repetition suppression, the synchrony model does not explain why repetition suppression, which is the most robust effect present in priming studies, occurs at all. [..]

We agree with the reviewer that our analyses are exploratory in terms of location – both for connectivity and for local effects of repetition suppression. That is true of all data-driven analyses. A critical aspect of such analyses is that they appropriately correct for the degree of searching that is done, and we have done that here. We have clarified in the Limitations section that the data-driven, whole-brain analyses are necessarily exploratory, and we have emphasized that they should be followed up in future studies with hypothesis-driven tests regarding the role of the right temporoparietal cortex (and the ventral attention network, more generally) in priming (pp. 19-21). We have also clearly stated the limitations of the models and the Synchrony model, in particular (pp. 21-22, quoted above).

Finally, there is the broader issue that it is becoming increasingly common for authors to report both significant multivariate and univariate findings in fMRI data, and I have little doubt that if older priming data sets relying strictly upon univariate analysis were re-analyzed, there would often be reliable multivariate effects as well. In this light, finding both effects in any given data set would not seem particularly strong evidence for or against any specific functional theory unless for some reason that theory forbade one class of analysis (univariate vs. multivariate) from rendering significant effects. Thus, while I found the connectivity findings quite interesting and a good target for future research, they don't strike me as dispositive for any particular model of priming. Finally, while the primeability scoring seems like a good methodological practice if it can be

conducted independently, the main contribution is equivalent to recommending that researchers conduct item analyses in an MLM framework or otherwise control for the fact that some items simply perform poorly on tests.

We disagree with the reviewer that previous fMRI datasets could be equally useful in examining multivariate (connectivity) and univariate (RS) simultaneously. Most studies have used rapid event related designs, which would not permit clean separation of OLD and NEW items, much less further subdivisions of these items. As previously discussed, the details of how RS and connectivity interrelate are not the same across theories, and these can indeed help to force revisions in the models. There are numerous novel empirical details in the current paper, and we think that proponents of the various models will find these details important as they consider revisions of their ideas. Subsequent work using ECoG and simultaneous fMRI/EEG will hopefully provide further clarity regarding the more subtle timing predictions of the Facilitation and Predictive Coding models (data which are, at present, not available for the longer-term repetitions studied here).

REVIEWERS' COMMENTS:

Reviewer #1 (Remarks to the Author):

I am happy with the authors' responses.